# Start Smart: Leveraging Gradients for enhancing Mask-based XAI Methods

**Buelent Uendes, Shujian Yu & Mark Hoogendoorn**
Vrije Universiteit Amsterdam
The Netherlands
`{b.uendes}@vu.nl`

## Abstract

Mask-based explanation methods offer a powerful framework for interpreting deep learning model predictions across diverse data modalities, such as images and time series, in which the central idea is to identify an instance-dependent mask that minimizes the performance drop from the resulting masked input. Different objectives for learning such masks have been proposed, all of which, in our view, can be unified under an information-theoretic framework that balances performance degradation of the masked input with the complexity of the resulting masked representation. Typically, these methods initialize the masks either uniformly or as all-ones. In this paper, we argue that an effective mask initialization strategy is as important as the development of novel learning objectives, particularly in light of the significant computational costs associated with existing mask-based explanation methods. To this end, we introduce a new gradient-based initialization technique called StartGrad, which is the first initialization method specifically designed for mask-based post-hoc explainability methods. Compared to commonly used strategies, StartGrad is provably superior at initialization in striking the aforementioned trade-off. Despite its simplicity, our experiments demonstrate that StartGrad enhances the optimization process of various state-of-the-art mask-explanation methods by reaching target metrics faster and, in some cases, boosting their overall performance.

## 1 Introduction

As machine learning models become deeply integrated into critical areas such as healthcare or medicine, the need for transparency and interpretability grows ever more urgent (Vellido, 2020). Explainable AI (XAI) research has responded by developing various XAI frameworks, with saliency methods—also referred to as feature attribution methods—at the forefront. These methods seek to highlight the most relevant inputs that drive a model's prediction, offering insights into how complex models, often regarded as black boxes, make decisions.

Mask-based explanation methods, a particularly powerful subset of feature attribution techniques, aim to learn sparse masks that highlight the key inputs driving a prediction. These methods are highly flexible: the objective function can be designed to capture specific notions of relevance, while constraints like sparsity or smoothness can be added to meet criteria for a 'good' explanation. This is often formalized through an information-theoretic framework, such as the information bottleneck (IB) principle (Tishby et al., 1999) and the rate-distortion function (Thomas & Joy, 2006). The carefully designed objective function in mask-based methods allows them to adapt effectively to different data modalities, resulting in superior performance compared to approaches such as Saliency maps (Simonyan et al., 2014), Integrated Gradients (Sundararajan et al., 2017), SmoothGrad (Smilkov et al., 2017), or surrogate-based methods like LIME (Ribeiro et al., 2016) and SHAP (Lundberg & Lee, 2017). These approaches have been also shown to produce explanations that are fragile and noisy, or even ones that can be manipulated (Adebayo et al., 2018; Ghorbani et al., 2019; Slack et al., 2020).

Despite their effectiveness, state-of-the-art mask-based explanation methods come with heavy computational costs due to their prolonged optimization process with hundreds of epochs of iterations,

especially when compared to gradient-based saliency techniques like SmoothGrad (Smilkov et al., 2017) or Integrated Gradients (Sundararajan et al., 2017). For instance, the most advanced mask-based methods in the vision domain such as the recently proposed WaveletX (Kolek et al., 2023) and ShearletX (Kolek et al., 2023), require orders of magnitude more execution time to generate explanations than their gradient-based counterparts.

This trade-off between performance and speed presents a major limitation for mask-based methods, especially in time-sensitive applications where rapid decisions are critical. As a result, users are often forced to choose between the superior faithfulness of mask-based explanations and the faster, but potentially less reliable, gradient-based alternatives. This tension between explanation accuracy and real-time applicability remains a key challenge in making mask-based methods more widely applicable in high-stakes environments such as healthcare.

To address the challenge of balancing performance and computational efficiency in mask-based methods, we draw on two key insights: First, while the way how to initialize masks in existing mask-based XAI methods is usually neglected, it plays a crucial role in optimization in terms of both running time and the final achievable maximum or minimum value. Second, although gradient-based saliency methods are not explicitly designed to meet desiderata of high-quality explanations, they do provide valuable signals about the model's decision-making process with minimal computational overhead. By combining these two insights, we propose StartGrad, a novel gradient-based mask initialization technique specifically designed for post-hoc explanation methods. StartGrad leverages gradient signals to provide provably superior initialization masks in terms of minimal distortion and sparsity—two essential criteria for mask-based explanation methods—compared to commonly used strategies. By doing so, StartGrad harnesses the strengths of gradient-based approaches to enhance existing mask-based explainability techniques.

We summarize our contributions as below:

- We introduce StartGrad, a novel gradient-based mask initialization algorithm grounded in the rate distortion explanation (RDE) framework (Macdonald et al., 2019), representing the first initialization technique explicitly designed to enhance the performance of mask-based explanation methods.

- We prove that StartGrad is superior at initialization compared to other initialization strategies in reducing distortion and improving sparsity—two essential criteria for effective mask-based explanations.

- Extensive experiments on vision and time-series tasks demonstrate that StartGrad enables state-of-the-art methods like ShearletX (Kolek et al., 2023) and ExtremalMask (Enguehard, 2023) to reach target metrics faster while also improving overall performance in some cases.

- To the best of our knowledge, this work presents the first comprehensive theoretical and empirical analysis of mask initialization techniques across both vision and time-series domains, providing critical insights for improving mask-based explanation methods.

## 2  RELATED WORK

Feature attribution methods can be divided into white-box, gray-box, and black-box approaches, depending on the amount of information required to generate an explanation (Muzellec et al., 2024). Mask-based explanation methods are considered black-box attribution techniques, as they do not require access to the model's internal architecture and rely only on model predictions. Recently, these methods gained attraction in the community and have been subsequently applied to different domains such as vision (Fong & Vedaldi, 2017; Petsiuk et al., 2018; Kolek et al., 2022; 2023) or time series (Crabbé & Van Der Schaar, 2021; Enguehard, 2023; Liu et al., 2024; Zichuan Liu, 2024). A key advantage of these techniques is their explicit optimization of an objective function that formalizes desiderata for "good" explanations, such as parsimony and fidelity to the model and often formalized via the IB principle (Tishby et al., 1999) or the RDE framework (Macdonald et al., 2019) which formalize the trade-off between explanation complexity and predictive accuracy.

Despite their strong performance, mask-based approaches are often computationally expensive due to iterative optimization, limiting their practicality in real-world applications. However, much of

the previous work has mainly investigated designing novel objective functions suited for either the data modality at hand (Ying et al., 2019; Crabbé & Van Der Schaar, 2021), proposing new architectural design choices such as learning the perturbation function(Enguehard, 2023; Liu et al., 2024), or even learning a trainable masking model that can produce masks after training (Dabkowski & Gal, 2017). To the best of our knowledge, no previous work has investigated the role of mask initialization scheme itself, which is surprising given the role initialization schemes can have on the overall performance of the optimization result (Glorot & Bengio, 2010; He et al., 2015). In fact, we noticed that across data modalities, different initialization techniques are used. For instance, mask-based XAI methods in the time-series domain initialize their masks uniformly at start (Crabbé & Van Der Schaar, 2021; Enguehard, 2023; Liu et al., 2024)[1], whereas vision explanation models use an all-ones initialization scheme (Kolek et al., 2022; 2023).

## 3 BACKGROUND

Different objectives have been developed to learn an "optimal" mask $\mathbf{m}$. Based on our observation, these objectives can be unified under the same umbrella, formalizing the trade-off between the complexity of the masked input and its predictive performance from an information-theoretic perspective. At their core, all methods revolve around these two fundamental concepts, with additional constraints, such as smoothness, often incorporated to further refine the explanations.

Given a pre-trained predictive model $\Phi_c : \mathbb{R}^d \to [0,1]^c$ for a total of $c$ classes and an input instance $\mathbf{x} \in \mathbb{R}^d$, the goal of the mask-based post-hoc explanation to a black-box output $\Phi_c(\mathbf{x})$ is to identify a sparse mask $\mathbf{m} \in \{0,1\}^d$ over $\mathbf{x}$, such that the resulting perturbed input $\tilde{\mathbf{x}}$ leads to the minimum performance drop in the model's prediction (Fong & Vedaldi, 2017; Macdonald et al., 2019). Usually, $\tilde{\mathbf{x}} := \mathbf{m} \odot \mathbf{x} + (1 - \mathbf{m}) \odot \mathbf{u}$, where $\mathbf{u} \in \mathbb{R}^d$ is random noise following a predefined probability distribution $V$, such as a Gaussian distribution.

### 3.1 THE RATE-DISTORTION EXPLANATION (RDE) FRAMEWORK

Formally, the objective of the RDE framework can be defined as the following constrained optimization problem (Macdonald et al., 2019; Kolek et al., 2022):

$$\min_{\mathbf{m}\in\{0,1\}^d} E_{\mathbf{u}\sim V}\left[\mathcal{D}\left(\Phi_c(\tilde{\mathbf{x}}), \Phi_c(\mathbf{x})\right)\right] \quad \text{s.t.} \ \|\mathbf{m}\|_0 \leq s, \tag{1}$$

where $\mathcal{D} : [0,1]^c \times [0,1]^c \to \mathbb{R}_+$ quantifies the performance difference between the classifier's prediction for the original input $\mathbf{x}$ and the perturbed input $\tilde{\mathbf{x}}$, a term which is also referred to as the "distortion". $s \in \{1, ..., d\}$ controls the sparsity of the mask $\mathbf{m}$. The mean-squared error loss function is often used as the distortion measure $\mathcal{D}$ [2].

In practice, the RDE optimization framework in equation 1 is relaxed using both a continuous mask $\mathbf{m} \in [0,1]^d$ and a $\|\cdot\|_1$ norm:

$$\min_{\mathbf{m}\in[0,1]^d} E_{\mathbf{u}\sim V}\left[\mathcal{D}\left(\Phi_c(\tilde{\mathbf{x}}), \Phi_c(\mathbf{x})\right)\right] + \lambda\|\mathbf{m}\|_1, \tag{2}$$

which can be optimized using gradient-based methods. Moreover, instead of applying the RDE framework to the original input domain, one can first transform $\mathbf{x}$ using an invertible function $\mathcal{F} : \mathbb{R}^d \to \mathbb{R}^k$, such as Wavelet transform (Kolek et al., 2022) and Shearlet transform (Kolek et al., 2023), and enforce sparsity in the frequency domain. In this scenario, the resulting objective becomes:

$$\min_{\mathbf{m}\in[0,1]^k} E_{\mathbf{u}\sim V}\left[\mathcal{D}\left(\Phi_c(\mathcal{F}^{-1}(\mathbf{m}\odot\mathcal{F}(\mathbf{x}) + (1-\mathbf{m})\odot\mathbf{u})), \Phi_c(\mathbf{x})\right)\right] + \lambda\|\mathbf{m}\|_1. \tag{3}$$

### 3.2 THE INFORMATION BOTTLENECK (IB) EXPLANATION FRAMEWORK

An alternative, yet increasingly popular, objective for learning the mask $\mathbf{m}$ is based on the IB principle (Tishby et al., 1999; Gilad-Bachrach et al., 2003):

$$\min_{\mathbf{m}\in[0,1]^d} -I(\hat{y}; \tilde{\mathbf{x}}) + \beta I(\mathbf{x}; \tilde{\mathbf{x}}), \tag{4}$$

---

[1]Strictly speaking, these methods start with a constant value of 0.5 which is the expected value of an uniform distribution $\mathcal{U}(0, 1)$.

[2]In fact, Kolek et al. (2022) report that the choice of the distortion function has limited effect on the performance of their proposed mask-based vision explanation methods.

where $I(\cdot;\cdot)$ represents the mutual information between two random variables, $\hat{y} = \Phi_c(\mathbf{x})$ is the black-box output, $\beta > 0$ is a Lagrange multiplier. Maximizing $I(\hat{y};\tilde{\mathbf{x}})$ ensures that the perturbed input $\tilde{\mathbf{x}}$ retains sufficient information to approximate or explain the black-box output $\hat{y}$ (Bang et al., 2021), while minimizing $I(\mathbf{x};\tilde{\mathbf{x}})$ encourages compression such that $\tilde{\mathbf{x}}$ does not capture any redundant information in $\mathbf{x}$ that is irrelevant to explain $\hat{y}$. In other words, the compression terms essentially encourages the conditional independence between $\tilde{\mathbf{x}}$ and $\mathbf{x}$, conditioning on $\hat{y}$ (Fischer, 2020).

Same to the RDE framework, the mask can be applied in a latent space (Schulz et al., 2020; Demir et al., 2021). In this case, the compression terms becomes $I(\mathbf{z}, \mathbf{z} \odot \mathbf{m} + (1 - \mathbf{m}) \odot \mathbf{u})$, in which $\mathbf{z} = f_l(\mathbf{x})$ refers to the $l$-th layer output of an instance $\mathbf{x}$.

Recently, the general idea of IB has been extended to the design of built-in interpretable deep learning architectures for image data (Choi et al., 2024) and graphs (Yu et al., 2021), where $\hat{y}$ is replaced by the ground-truth label $y$. However, our paper remains focused on post-hoc explanations.

From our perspective, there is a clear resemblance between equation 4 and the RDE objective in equation 2, as illustrated in Proposition 1 and Remark 1.

**Proposition 1** *Maximizing $I(\hat{y};\tilde{\mathbf{x}})$ is equivalent to minimizing the expected Kullback–Leibler (KL) divergence $\mathbb{E}\left(D_{KL}(\Phi_c(\tilde{\mathbf{x}});\Phi_c(\mathbf{x}))\right)$, under the choice of KL divergence as the distortion measure in the RDE framework. Furthermore, the expected KL divergence provides an upper bound for the $\ell_1$-loss, i.e., $\mathbb{E}\left(D_{KL}(\Phi_c(\tilde{\mathbf{x}});\Phi_c(\mathbf{x}))\right) \geq \frac{1}{2\log 2}\mathbb{E}\left(\|\Phi_c(\tilde{\mathbf{x}}) - \Phi_c(\mathbf{x})\|_1^2\right)$.*

**Remark 1** *In the IB implementation, the compression term $I(\mathbf{x};\tilde{\mathbf{x}})$ can simply be replaced by an entropy term $H(\tilde{\mathbf{x}})$[3] (Strouse & Schwab, 2017; Kirsch et al., 2020). In fact, the simplest way to compress $H(\tilde{\mathbf{x}})$ is by encouraging the number of explanatory variables to be small, i.e., encouraging $m$ to be sparse (Bang et al., 2021; Tao et al., 2020). This regularization can be expressed as $\|m\|_0$ or approximated with $\|m\|_1$.*

*Proof.* All proofs can be found in Appendix A.

## 4 METHOD

In this section, we present StartGrad, our novel gradient-based mask initialization algorithm. Unlike traditional approaches, StartGrad leverages gradient information to provide tailored initialization of masks based on the specific characteristic of the sample and model. This gradient-informed initialization can not only provide a more effective starting point at *initialization* in light of the previously discussed RDE and IB framework, but can also significantly boost the performance of subsequent mask-based explanation algorithms as we will show in section 5.

### 4.1 APPROXIMATING THE DISTORTION LOCALLY

Given a differentiable classifier $\Phi_c$, we can use a first-order Taylor expansion to approximate the prediction for the distorted input $\tilde{\mathbf{x}}$:

$$\Phi_c(\tilde{\mathbf{x}}) \approx \Phi_c(\mathbf{x}) + \nabla\mathbf{x}\Phi_c(\mathbf{x}) \cdot \Delta\mathbf{x}, \tag{5}$$

where $\nabla_{\mathbf{x}}\Phi_c(\mathbf{x})$ represents the gradient of the model's prediction with respect to the original input feature $\mathbf{x}$ and $\Delta\mathbf{x} = \tilde{\mathbf{x}} - \mathbf{x}$ denotes the distortion induced due to the mask $\mathbf{m}$. To ensure that this approximation is accurate, we require that $\Phi_c$ is locally linear in a small neighborhood around $\mathbf{x}$. Formally, this neighborhood is defined as the open ball $B_\epsilon(\boldsymbol{x}) = \{\tilde{\boldsymbol{x}} \in \mathbb{R}^d : \|\tilde{\boldsymbol{x}} - \boldsymbol{x}\| < \epsilon\}$, where $\epsilon > 0$ ensures that higher-order terms in the Taylor expansion of $\Phi_c$ are negligible.

Using a $\|\cdot\|_p$ norm as a choice for the distortion function $\mathcal{D}$, we can leverage equation 5 to approximate the distortion:

$$\mathcal{D}\left(\Phi_c(\tilde{\mathbf{x}}), \Phi_c(\mathbf{x})\right) = \|\Phi_c(\tilde{\mathbf{x}}) - \Phi_c(\mathbf{x}))\|_p \approx \|\nabla_{\mathbf{x}}\Phi_c(\mathbf{x}) \cdot \Delta\mathbf{x}\|_p = \left(\sum_{i=1}^{d} |\nabla_{x_i}\Phi_c(\mathbf{x}) \cdot \Delta x_i|^p\right)^{\frac{1}{p}}. \tag{6}$$

---

[3]If the mapping from $\mathbf{x}$ to $\tilde{\mathbf{x}}$ is deterministic, we always have this equivalence; however, this may not hold true in practice, such as when unmasked regions of an image are substituted with random Gaussian noise.

From the approximation in equation 6, it follows that minimizing the distortion $\mathcal{D}\left(\Phi_c(\tilde{\mathbf{x}}), \Phi_c(\mathbf{x})\right)$ requires controlling the term $\|\nabla_{\mathbf{x}}\Phi_c(\mathbf{x}) \cdot \Delta\mathbf{x}\|_p$. In particular, we can leverage the gradient information to the minimize the distortion.[4] Features with small gradient magnitudes $|\nabla_{x_i}\Phi_c(\mathbf{x})|$ have less impact on the distortion term and can be therefore masked without significantly increasing $\mathcal{D}$. This insight leads to the following heuristic:

$$M_i = \begin{cases} 1 & \text{if } |\nabla_{x_i}\Phi_c(\mathbf{x})| \text{ is "large"}, \\ 0 & \text{if } |\nabla_{x_i}\Phi_c(\mathbf{x})| \text{ is "small"}. \end{cases} \tag{7}$$

Since $\nabla_{x_i}\Phi_c(\mathbf{x})$ indicates the sensitivity of the model's prediction to changes in the corresponding feature, masking features with "small" gradients ensures that any induced changes minimally impact the model prediction. This approach is crucial in maintaining the predictive performance while modifying or explaining the input, effectively balancing sparsity and distortion.

## 4.2 GRADIENT-BASED MASK INITIALIZATION

A key challenge in our approach is setting appropriate thresholds to distinguish between "large" and "small" gradients when defining the mask in equation 7. As discussed in section 3.1, we avoid a binary threshold by leveraging continuous masks. Specifically, we define the absolute gradient of the model's prediction with respect to the class of interest as:

$$\mathcal{S}_c(\mathbf{x}) = \left|\frac{\partial \Phi_c(\mathbf{x})}{\partial \mathbf{x}}\right|. \tag{8}$$

For multi-channel inputs such as RGB images, we take the maximum value across channels. We then apply a transformation function $\mathcal{T} : \mathbb{R}_+^d \to [0,1]^d$, which normalizes the gradient signal $\mathcal{S}_c(\mathbf{x})$ into the range $[0,1]^d$. The primary challenge lies in selecting a suitable transformation function that preserves the relative importance of each feature.

Empirically, we observe that the absolute gradient values follow a highly skewed distribution, similar to a Laplace distribution[5] (see Appendix D). Given this skewness, we employ the Quantile Transformation Function (QTF), which maps the gradient values to a uniform distribution $\mathcal{U}(0,1)$. The QTF has two key advantages: (1) it handles the skewness in the data effectively, and (2) it preserves the relative feature importance based on the absolute gradient values. Furthermore, using the QTF facilitates direct comparison with the standard uniform mask initialization scheme, thereby simplifying the evaluation of our proposed framework. Algorithm 1 gives pseudocode for our gradient-based initialization method which we call StartGrad.[6]

---

**Algorithm 1:** Gradient-based Mask Initialization (StartGrad)

---

| **Input** | : Pre-trained classifier $\Phi_c$, input samples $\{\boldsymbol{x}_i\}_{i=1}^N$ with $\boldsymbol{x}_i \in \mathbb{R}^d$, quantile transformation $\mathcal{T} : \mathbb{R}_+^d \to [0,1]^d$ |

**Hyperparameters:** Output distribution $\mathcal{U}(0,1)$, number of quantiles (bins) for $\mathcal{T}$

**Output** : Masks $\mathbf{M} \in [0,1]^{N \times d}$

1 Initialize mask list $\mathbf{M} := []$ and quantile transformer $\mathcal{T}$;
2 **for** $i \leftarrow 1$ **to** $N$ **do**
3 $\quad$ $c_i \leftarrow \arg\max(\Phi_c(\boldsymbol{x}_i))$ // Class prediction
4 $\quad$ $\mathbf{S}_i \leftarrow |\nabla_{\boldsymbol{x}_i}\Phi_c(\boldsymbol{x}_i)|_c$ // Gradient magnitudes
5 $\quad$ $\mathbf{m}_i \leftarrow \mathcal{T}(\mathbf{S}_i)$ // Quantile transform
6 $\quad$ Append $\mathbf{m}_i$ to $\mathbf{M}$;
7 **end for**
8 **return** $\mathbf{M}$

---

[4]An alternative strategy is to trivially set $\Delta\mathbf{x} = \mathbf{0}_d$, i.e. $\mathbf{m} = \mathbf{1}_d$, thereby keeping all features and eliminating distortion entirely. However, this solution fails to satisfy the sparsity objective of the RDE framework and is thus suboptimal for the overall task.

[5]This behavior has been previously observed in other domains as well, such as time series (Ismail et al., 2020) and graph-based models (Xie et al., 2022)

[6]For simplicity, we assume for the pseudocode in algorithm 1 that we operate in the same input space $\mathcal{X}$. In Appendix B, a more general version is provided where we we use an invertible and differentiable function $\mathcal{F}$. A visualization of StartGrad is also provided in the Appendix, as it was moved there due to space constraints.

### 4.3 Balancing sparsity and distortion under the RDE framework

As outlined in section 3.1, the primary objective of mask-based explanation algorithms is typically to identify a sparse mask that minimally distorts the model's predictions.[7] While our previous discussion in 4.1 has centered on minimizing distortion at initialization, we now turn our attention to sparsity, comparing the standard initialization strategies of all-ones and uniform to our newly introduced algorithm StartGrad. Specifically, we provide formal proofs that StartGrad is *provably superior at initialization* in balancing the trade-off between distortion and sparsity, as formalized in the RDE framework, when compared to other mask initialization strategies. To the best of our knowledge, this represents the first formal comparison of different mask initialization methods.

**Proposition 2** *Let $\Phi_c(\boldsymbol{x})$ with $\Phi_c : \mathbb{R}^d \to [0,1]^c$ be a classifier that is differentiable and locally linear in the neighborhood of an input $\mathbf{x} \in \mathbb{R}^d$. Formally, this neighborhood is defined as the open ball $B_\epsilon(\boldsymbol{x}) = \{\tilde{\boldsymbol{x}} \in \mathbb{R}^d : \|\tilde{\boldsymbol{x}} - \boldsymbol{x}\| < \epsilon\}$, where $\epsilon > 0$ ensures that higher-order terms in the Taylor expansion of $\Phi_c$ are negligible. Additionally, let $\mathbf{m}_{unif}$ represent the uniformly-initialized mask and $\mathbf{m}_{grad}$ the mask initialized based on the gradient of $\Phi_c(\mathbf{x})$ with respect to $\mathbf{x}$, transformed using a quantile function so that $\mathbf{m}_{grad} \in [0,1]^d$. Then, given a norm $\|\cdot\|_p$ with $p \geq 1$ as a choice for the distortion function $\mathcal{D}$, the following inequality holds at initialization:*

$$\mathbb{E}_{\mathbf{m}_{grad},\mathbf{u}\sim V}\left[D\left(\Phi_c(\tilde{\mathbf{x}}_{grad}),\Phi_c(\mathbf{x})\right) + \lambda\|\mathbf{m}_{grad}\|_1\right] \leq \mathbb{E}_{\mathbf{m}_{unif},\mathbf{u}\sim V}\left[D\left(\Phi_c(\tilde{\mathbf{x}}_{unif}),\Phi_c(\mathbf{x})\right) + \lambda\|\mathbf{m}_{unif}\|_1\right]$$

*with $\tilde{\mathbf{x}}_{unif} = \mathbf{m}_{unif} \odot \mathbf{x} + (1 - \mathbf{m}_{unif}) \odot \mathbf{u}$ and $\tilde{\mathbf{x}}_{grad} = \mathbf{m}_{grad} \odot \mathbf{x} + (1 - \mathbf{m}_{grad}) \odot \mathbf{u}$, where $\mathbf{u} \in \mathbb{R}^d$ is a random perturbation drawn from a predefined distribution $V$ and $\lambda$ is a hyperparameter encouraging sparsity in the masks.*

**Proposition 3** *Let $\Phi_c(\boldsymbol{x})$ with $\Phi_c : \mathbb{R}^d \to [0,1]^c$ be a classifier and let $\boldsymbol{m}_{ones}$ denote the mask initialized with all ones, i.e. $\boldsymbol{m}_{ones} = \mathbf{1}_d$ where $\mathbf{1}_d$ is a vector of ones with dimension $d$ and $\mathbf{m}_{grad}$ denote the mask initialized based on the gradient of $\Phi_c(\mathbf{x})$ with respect to $\mathbf{x}$, transformed using a quantile function so that $\mathbf{m}_{grad} \in [0,1]^d$. Given a norm $\|\cdot\|_p$ with $p \geq 1$ as a choice for the distortion function $\mathcal{D}$, the following inequality holds at initialization:*

$$\mathbb{E}_{\mathbf{m}_{grad},\mathbf{u}\sim V}\left[D\left(\Phi_c(\tilde{\mathbf{x}}_{grad}),\Phi_c(\mathbf{x})\right) + \lambda\|\mathbf{m}_{grad}\|_1\right] \leq \mathbb{E}_{\mathbf{m}_{ones},\mathbf{u}\sim V}\left[D\left(\Phi_c(\tilde{\mathbf{x}}_{ones}),\Phi_c(\mathbf{x})\right) + \lambda\|\mathbf{m}_{ones}\|_1\right]$$

*for*

$$\frac{2\left(1 + (2-1)\mathbb{1}_{c\geq 2}\right)^{\frac{1}{p}}}{d} \leq \lambda$$

*with $\tilde{\mathbf{x}}_{ones} = \mathbf{m}_{ones} \odot \mathbf{x} + (1 - \mathbf{m}_{ones}) \odot \mathbf{u}$ and $\tilde{\mathbf{x}}_{grad} = \mathbf{m}_{grad} \odot \mathbf{x} + (1 - \mathbf{m}_{grad}) \odot \mathbf{u}$, where $\mathbf{u} \in \mathbb{R}^d$ is a random perturbation drawn from a predefined distribution $V$ and $\lambda$ is a hyperparameter encouraging sparsity in the masks.*

*Proof.* All proofs can be found in Appendix A.

**Remark 2** *In case we want to explain the class prediction of a classifier $\Phi_c(\boldsymbol{x})$ for a specific category, i.e. $c = 1$, Proposition 3 reduces to*

$$\frac{2^{\frac{1}{p}}}{d} \leq \lambda$$

**Remark 3** *Proposition 3 holds in the same manner for the comparison between $\mathbf{m}_{unif}$ and $\mathbf{m}_{ones}$.*

Proposition 2 and 3 together provide a principled approach for selecting an initialization strategy within the RDE framework. Our proposed gradient-based initialization algorithm yields an expected loss in terms of distortion and sparsity that is less than or equal to that of uniform initialization. Additionally, StartGrad outperforms all-ones initialization in high-dimensional settings (where $d$ is large) which is a typical setting for deep learning models, and consequently, mask-based explanation approaches.

---

[7]In addition to sparsity and distortion, some mask-based algorithms also regularize the smoothness of the mask (Fong & Vedaldi, 2017). Furthermore, rather than minimizing distortion, certain algorithms aim to find the smallest mask that induces the maximum distortion (Enguehard, 2023).

## 5 EXPERIMENTS

In this section, we evaluate our gradient-based initialization scheme using state-of-the-art mask-based explanation methods across two challenging data modalities: vision and time series.[8] Vision is a domain where mask-based explanation methods are well-established, while time-series data, though less studied in the XAI community, is critical in fields like healthcare and finance, gaining more attention in recent years (Crabbé & Van Der Schaar, 2021; Enguehard, 2023; Liu et al., 2024; Queen et al., 2023; Zichuan Liu, 2024). These domains present distinct challenges for gradient estimation—vision involves spatial complexity, while time-series data introduces temporal dependencies (Ismail et al., 2020). This cross-domain evaluation demonstrates how StartGrad adapts to varied data characteristics and model architectures, underscoring the robustness and potential of our proposed initialization algorithm.

### 5.1 VISION

We evaluate StartGrad using two state-of-the-art mask-based explanation methods: WaveletX, ShearletX (Kolek et al., 2023) and a well-established third method, i.e. PixelMask (Fong & Vedaldi, 2017). These vision methods were chosen as they apply masking in different input spaces and vary in their objective formulations, creating a diverse and challenging testbed for our proposed initialization method.

For quantitative evaluation, we use the conciseness-preciseness (CP) Pixel and L1 scores introduced by Kolek et al. (2023), calculated on 500 random samples from the ImageNet validation dataset (Deng et al., 2009). These information-theoretic metrics are designed to reward masks that preserve classification accuracy while minimizing the amount of information used, making them well-suited for mask-based methods that prioritize sparsity and compressed explanations. Unlike faithfulness metrics (e.g., insertion and deletion), which rely on an inherent ranking of feature importance, CP scores are specifically designed for methods like ours that optimize binary masks without ordering feature relevance (Kolek et al., 2023). Additionally, using CP scores ensures consistency and comparability with the baseline methods in Kolek et al. (2023).[9]

As in Kolek et al. (2023), we use a pretrained ResNet18 model (He et al., 2016) and VGG16 (Simonyan & Zisserman, 2014) for classification. We also evaluate StartGrad on vision transformers (Dosovitskiy et al., 2021) and swin transformers (Liu et al., 2022). Due to space constraints, only results for the ResNet18 model are shown in the main body of the paper, with results for other architectures provided in Appendix F.1, F.4, and F.3. The qualitative findings discussed here are vastly consistent across all tested models. For all mask-based vision models, we use the default parameter settings as reported in the respective papers to facilitate comparisons. Further details on the explanation models and hyperparameters are in Appendix C.1. Additional results and ablation studies are provided in section D.3 and F. In Appendix F.18, F.19, F.20 we also provide visual results.

**Results.** Table 1 shows the median performance difference between StartGrad and both all-ones and uniform initialization strategies. Across all mask-based explanation methods, StartGrad significantly improves performance compared to the uniform initialization which shares the underlying initialization distribution. This underscores the advantage of using gradient signals to guide the mask initialization process.

For PixelMask (Fong & Vedaldi, 2017) and WaveletX (Kolek et al., 2023), StartGrad particularly enhances performance at early iteration steps, whereas at the final iteration step (300) no statistical differences can be observed for these two mask-based explanation methods. However, initializing ShearletX (Kolek et al., 2023) with our proposed StartGrad algorithm leads to a substantial and statistically significant performance improvement at all iteration steps.

In addition to measuring performance at a given iteration step, we examine how StartGrad reduces runtime (as measured in iteration steps) when aiming for a specific target metric. However, defining a universally "good" target score is inherently challenging due to the absence of a clear threshold. To tackle this, we define the per image target score as the highest attainable score across all three initializations and measure subsequently the iterations required to this specific normalized score.

---

[8]Our code is publicly available at `https://github.com/BuelentUendes/StartGrad`

[9]For a more detailed justification of using the CP scores in place of faithfulness, see Appendix C.2.2.

Table 1: Median pairwise performance difference between StartGrad and alternative initialization methods across iteration steps for 500 ImageNet (Deng et al., 2009) samples using a ResNet18 (He et al., 2016) classifier. Baseline refers to the initialization scheme originally used for each mask explanation method. Statistical significance is marked as ***, **, and * for 1%, 5%, and 10% levels, respectively, using a one-sided Wilcoxon signed-rank test with Bonferroni correction. For PixelMask (Fong & Vedaldi, 2017), CP-Pixel and CP-L1 scores are identical as it operates only in the pixel space.

| | | △ CP-Pixel ↑ | | | △ CP-L1 ↑ | | |
|---|---|---|---|---|---|---|---|
| | | Iteration steps | | | Iteration steps | | |
| Method | Reference initialization | 50 | 100 | 300 | 50 | 100 | 300 |
| PixelMask | All-ones (baseline) | 2.09*** | 0.72 | 0.33 | 2.09*** | 0.72 | 0.33 |
| | Uniform | 1.11*** | 1.26*** | 1.12*** | 1.11*** | 1.26*** | 1.12*** |
| WaveletX | All-ones (baseline) | 2.55*** | 0.73** | 0.11 | 0.71*** | 0.37*** | 0.14 |
| | Uniform | 2.35*** | 3.39*** | 5.20*** | 0.60*** | 0.88*** | 1.04*** |
| ShearletX | All-ones (baseline) | 52.91*** | 135.28*** | 236.37*** | 2.89*** | 20.67*** | 51.82*** |
| | Uniform | 58.65*** | 130.71*** | 208.74*** | 3.09*** | 33.42*** | 65.57*** |

As shown in Figure 1, StartGrad outperforms the other two initializations strategies for PixelMask (Fong & Vedaldi, 2017) and WaveletX (Kolek et al., 2023) for medium target scores and for ShearletX (Kolek et al., 2023) beginning from a score of 0.3.

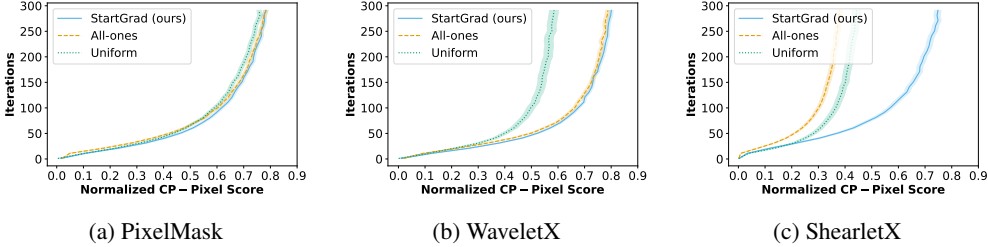

(a) PixelMask         (b) WaveletX         (c) ShearletX

Figure 1: Normalized CP-Pixel score vs. iteration steps for the PixelMask (Fong & Vedaldi, 2017) (a), WaveletX (Kolek et al., 2023) (b), and ShearletX (Kolek et al., 2023) (c) models, comparing three initialization methods: StartGrad (ours) (solid line), All-ones (dashed line), and Uniform (dotted line). The curves represent the average number of iteration steps needed to reach a normalized target CP-Pixel score. This target score is defined as the highest CP-Pixel value achieved across all three initialization methods for each model, then normalized by the overall maximum score observed. The shaded regions represent the standard error across 500 randomly selected ImageNet (Deng et al., 2009) validation samples, evaluated using a pretrained ResNet18 (He et al., 2016) classifier.

Given the strong performance of the StartGrad initialization method for the ShearletX (Kolek et al., 2023) model, we seek to evaluate how quickly a mask-based algorithm initialized with StartGrad can reach the final performance levels of the all-ones and uniform initialization strategies. As it can be seen from table 2, we achieve large speedups up to 6.73, further illustrating the potential of StartGrad and designing mask-based initialization strategies in general. To further understand the dynamics of the optimization process and the effectiveness of StartGrad, we analyze its role in guiding mask-based explanation methods. As shown in Appendix F.6, F.7, F.8, initializing the mask using gradient signals, as implemented in StartGrad, provides a crucial early advantage by guiding the optimization process toward regions of minimal distortion, consistent with theoretical predictions. This early advantage is critical, as uniform initialization fails to catch up in terms of distortion reduction throughout the optimization process which underlines the importance of an effective initialization.

Table 2: Average iteration steps for the ShearletX (Kolek et al., 2023) method initialized with Start-Grad to match the performance of the all-ones or uniform initializations. Reported are the iteration steps to match the interquartile mean (IQM) and median of the CP-Pixel and CP-L1 scores, along with speedup calculated as the ratio of reference initialization to StartGrad iteration steps. All experiments use 500 randomly selected ImageNet (Deng et al., 2009) samples with a pretrained ResNet18 (He et al., 2016) classifier, and metrics are computed over 300 iterations. Average and standard errors are estimated via bootstrapping (250 resamples). For each metric, ↑ denotes higher is better and ↓ lower is better. Values that are two standard errors away from the baseline (300 iterations, 1 for speedup) are highlighted in bold.

| Reference | Target metric (at 300 iterations) | CP-Pixel | | CP-L1 | |
|---|---|---|---|---|---|
| | | Iterations ↓ | Speedup ↑ | Iterations ↓ | Speedup ↑ |
| All-ones (baseline) | IQM | **44.57 ± 2.36** | **6.73 ± 0.35** | **71.09 ± 6.47** | **4.24 ± 0.38** |
| | Median | **49.60 ± 3.39** | **6.06 ± 0.41** | **87.70 ± 9.98** | **3.45 ± 0.38** |
| Uniform | IQM | **45.50 ± 3.18** | **6.60 ± 0.45** | **72.92 ± 8.68** | **4.16 ± 0.47** |
| | Median | **50.58 ± 4.26** | **5.95 ± 0.49** | **94.44 ± 15.10** | **3.24 ± 0.47** |

## 5.2 TIME SERIES

Following Tonekaboni et al. (2020), Crabbé & Van Der Schaar (2021) and Enguehard (2023), we test StartGrad on two commonly used synthetic benchmark datasets, i.e. state and switch-feature data both of which use a hidden Markov model (HMM) to generate the data. In line with Crabbé & Van Der Schaar (2021); Enguehard (2023) we train a one-layer GRU (Cho et al., 2014) for each of the experiments. As a baseline mask-based explanation method we use the recently introduced Ex-tremalMask (Enguehard, 2023), which showed state-of-the-art performance on the aforementioned datasets. As a perturbation model, we use a bidirectional one-layer GRU (Cho et al., 2014) model. We evaluate StartGrad using the following four metrics: area under recall (AUR), area under precision (AUP), information (I) and mask entropy (E), where the last two were introduced by Crabbé & Van Der Schaar (2021). For both synthetic datasets, we use the same experimental setup as in previous studies (Crabbé & Van Der Schaar, 2021; Enguehard, 2023; Liu et al., 2024), i.e. we generate 1,000 samples and train the classifier on 800 training examples, and evaluate the performance on the remaining 200 samples while reporting results across five folds. We use the default hyperparameter settings as well as the same random seed as used by Enguehard (2023) to facilitate fair comparisons.

A complete description of the datasets, models and hyperparameters used in this study can be found in the Appendix C.3. Due to space constraints, we only report the results on the dwitch-feature dataset for the preservation game version of the ExtremalMask (Enguehard, 2023) model. All remaining results are provided in section E.

**Results.** As illustrated in Table 3, initializing ExtremalMask (Enguehard, 2023) with StartGrad yields strong performance improvements especially in the early iteration steps. At iteration 50, StartGrad outperforms the baseline (uniform) in all metrics, with notably performance improvements. Compared to the all-ones initialization, StartGrad provides performance improvements in 3 out 4 metrics, with the exception of AUP which is slightly lower. However, the all-ones mask initialization has a low corresponding AUR score, indicating that ExtremalMask (Enguehard, 2023) initialized with all-ones method misses a lot of important features. This advantage is maintained through iteration 100. By iteration 500, all methods converge to similar final performance, demonstrating that StartGrad provides a strong initial boost without compromising the overall outcome for ExtremalMask (Enguehard, 2023). The findings are consistent across datasets and objective formulations as shown in Appendix E.

## 6 LIMITATIONS AND FUTURE RESEARCH

Despite the promising results of the StartGrad initialization algorithm, few limitations require attention:

Table 3: Average performance for the ExtremalMask (Enguehard, 2023) explanation method (preservation game objective) across iteration steps for the switch-feature dataset when initialized with StartGrad, all-ones and uniformly. The reported numbers are the mean and standard deviation across five folds. For each metric, ↑ indicates that higher is better, and ↓ that lower is better. Baseline refers to the initialization scheme originally used for the respective mask explanation method. Outcomes that are one standard deviation away from the second-best ones are highlighted in bold.

| Metric | Initialization | Iteration steps | | | |
| | | 50 | 100 | 300 | 500 |
|---|---|---|---|---|---|
| AUP ↑ | Uniform (baseline) | $0.849 \pm 0.049$ | $0.942 \pm 0.039$ | $0.982 \pm 0.008$ | $0.986 \pm 0.004$ |
| | All-ones | $\mathbf{0.952 \pm 0.003}$ | $0.967 \pm 0.004$ | $0.986 \pm 0.003$ | $0.986 \pm 0.003$ |
| | StartGrad (ours) | $0.922 \pm 0.004$ | $\mathbf{0.984 \pm 0.003}$ | $0.987 \pm 0.003$ | $0.987 \pm 0.003$ |
| AUR ↑ | Uniform (baseline) | $0.746 \pm 0.029$ | $0.737 \pm 0.023$ | $0.747 \pm 0.020$ | $0.748 \pm 0.020$ |
| | All-ones | $0.696 \pm 0.015$ | $\mathbf{0.769 \pm 0.017}$ | $0.751 \pm 0.020$ | $0.750 \pm 0.020$ |
| | StartGrad (ours) | $\mathbf{0.805 \pm 0.014}$ | $0.758 \pm 0.020$ | $0.751 \pm 0.020$ | $0.751 \pm 0.021$ |
| I $[10^5]$ ↑ | Uniform (baseline) | $1.069 \pm 0.085$ | $1.876 \pm 0.229$ | $2.513 \pm 0.079$ | $2.533 \pm 0.078$ |
| | All-ones | $1.905 \pm 0.061$ | $2.446 \pm 0.091$ | $2.541 \pm 0.079$ | $2.539 \pm 0.072$ |
| | StartGrad (ours) | $\mathbf{2.498 \pm 0.105}$ | $\mathbf{2.533 \pm 0.083}$ | $2.544 \pm 0.082$ | $2.540 \pm 0.082$ |
| E $[10^4]$ ↓ | Uniform (baseline) | $2.325 \pm 0.151$ | $1.659 \pm 0.197$ | $1.281 \pm 0.071$ | $1.267 \pm 0.072$ |
| | All-ones | $1.984 \pm 0.066$ | $1.582 \pm 0.085$ | $1.277 \pm 0.070$ | $1.276 \pm 0.072$ |
| | StartGrad (ours) | $\mathbf{1.688 \pm 0.103}$ | $\mathbf{1.370 \pm 0.074}$ | $1.276 \pm 0.069$ | $1.278 \pm 0.069$ |

First, StartGrad's effectiveness relies on accurate gradient estimation, making it potentially sensitive to noise. In Appendix F.5, we assess the impact of noisy gradients in the vision domain by adding Gaussian noise to the gradient signal. Appendix F.13 examines uninformative gradients by shuffling values and explores an adversarial setting where gradient-mask associations are inverted. Finally, Appendix F.10 evaluates sensitivity to gradient estimation methods using SmoothGrad (Smilkov et al., 2017). Our findings show StartGrad remains robust to noise and alternative estimation techniques, matching uniform initialization in uninformative scenarios. However, its failure to recover from adversarially initialized masks highlights the importance of proper initialization.

Second, the choice of the transformation function $\mathcal{T}$ that maps the gradients into a continuous mask is crucial for StartGrad as illustrated in Appendix F.11. Future work could investigate alternative transformations functions other than the QTF. Another promising direction for future work is filtering high-frequency artifacts in gradient signals before applying the QTF transform, as this has been shown to improve the quality of the resulting gradient signal (Muzellec et al., 2024).

Third, compared to other initialization techniques, StartGrad involves some computational overhead. Yet, as illustrated in Appendix D.2, the additional computational costs are negligible, especially in light of the presented performance gains and speedups.

## 7 CONCLUSION

In this work, we present the first comprehensive theoretical and empirical analysis of mask initialization techniques and introduce StartGrad, a novel gradient-based mask initialization method. Specifically designed for post-hoc mask-based explanation methods, we demonstrate that StartGrad is provably superior in terms of distortion and sparsity at initialization compared to other techniques. Despite its simplicity, our experiments demonstrate that StartGrad enhances the optimization process of various state-of-the-art mask-based explanation methods, particularly by improving performance early on.

Given its effectiveness, StartGrad can serve as a strong baseline initialization method moving forward. We also hope that this work sparks further interest in exploring strategies for effective initialization techniques in mask-based explanation methods in an attempt to improve their practicality in real-world applications.

ACKNOWLEDGMENTS

This work is funded by Stress in Action. The research project 'Stress in Action': www.stress-in-action.nl is financially supported by the Dutch Research Council and the Dutch Ministry of Education, Culture and Science (NWO gravitation grant number 024.005.010).

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

# A PROOFS

## A.1 RESEMBLANCE BETWEEN IB OBJECTIVE AND RDE OBJECTIVE

The IB optimization trade-off can be considered as a generalized rate-distortion problem (Tishby et al., 1999; Piran et al., 2020) with the distortion function between an instance $\mathbf{x}$ and a (compressed) representation $\tilde{\mathbf{x}}$ taken as the KL-divergence between their predictions of the black-box output $\hat{y}$:

$$
\begin{aligned}
d_{\text{IB}}(\mathbf{x}; \tilde{\mathbf{x}}) &= D_{\text{KL}}[p(\hat{y}|\mathbf{x}); p(\hat{y}|\tilde{\mathbf{x}})] \\
&= \sum_{\hat{y}} p(\hat{y}|\mathbf{x}) \log \left( \frac{p(\hat{y}|\mathbf{x})}{p(\hat{y}|\tilde{\mathbf{x}})} \right).
\end{aligned}
\tag{9}
$$

The expected distortion $\mathbb{E}[d_{\text{IB}}(\mathbf{x}; \tilde{\mathbf{x}})]$ simply reduces to $I(\mathbf{x}; \hat{y}) - I(\tilde{\mathbf{x}}; \hat{y})$. This is because:

$$
\begin{aligned}
\mathbb{E}_{p(\mathbf{x}, \tilde{\mathbf{x}})}[d_{\text{IB}}(\mathbf{x}; \tilde{\mathbf{x}})] &= \mathbb{E}[\sum_{\hat{y}} p(\hat{y}|\mathbf{x}) \log p(\hat{y}|\mathbf{x})] - \mathbb{E}[\sum_{\hat{y}} p(\hat{y}|\mathbf{x}) \log p(\hat{y}|\tilde{\mathbf{x}})] \\
&= \sum_{\mathbf{x}} \sum_{\tilde{\mathbf{x}}} \sum_{\hat{y}} p(\mathbf{x}, \tilde{\mathbf{x}}) p(\hat{y}|\mathbf{x}) \log p(\hat{y}|\mathbf{x}) - \sum_{\mathbf{x}} \sum_{\tilde{\mathbf{x}}} \sum_{\hat{y}} p(\mathbf{x}, \tilde{\mathbf{x}}) p(\hat{y}|\mathbf{x}) \log p(\hat{y}|\tilde{\mathbf{x}}) \\
&= \sum_{\mathbf{x}} \sum_{\tilde{\mathbf{x}}} \sum_{\hat{y}} p(\mathbf{x}, \tilde{\mathbf{x}}, \hat{y}) \log p(\hat{y}|\mathbf{x}) - \sum_{\mathbf{x}} \sum_{\tilde{\mathbf{x}}} \sum_{\hat{y}} p(\mathbf{x}, \tilde{\mathbf{x}}, \hat{y}) \log p(\hat{y}|\tilde{\mathbf{x}}) \\
&= \mathbb{E}[\log p(\hat{y}|\mathbf{x})] - \mathbb{E}[\log p(\hat{y}|\tilde{\mathbf{x}})] \\
&= -H(\hat{y}|\mathbf{x}) + H(\hat{y}) + H(\hat{y}|\tilde{\mathbf{x}}) - H(\hat{y}) \\
&= I(\hat{y}; \mathbf{x}) - I(\hat{y}; \tilde{\mathbf{x}}),
\end{aligned}
\tag{10}
$$

where the third line of equation uses the Markov chain property $\hat{y} \leftarrow \mathbf{x} \rightarrow \tilde{\mathbf{x}}$, i.e., $p(\hat{y}|\mathbf{x}, \tilde{\mathbf{x}}) = p(\hat{y}|\mathbf{x})$. Line five uses the definition of conditional entropy, i.e. $H(\hat{y}|x) = -\mathbb{E}[\log p(\hat{y}|x)]$. and line six the definition of mutual information $I(\hat{y}; \boldsymbol{x}) = H(\hat{y}) - H(\hat{y}|\boldsymbol{x})$ respectively.

Since $I(\hat{y}; \mathbf{x})$ is a constant that is independent of the optimization, minimizing the expected IB distortion $\mathbb{E}[D_{\text{KL}}[p(\hat{y}|\mathbf{x}); p(\hat{y}|\tilde{\mathbf{x}})]]$ is equivalent to maximizing $I(\hat{y}; \tilde{\mathbf{x}})$.

Note that, in neural networks implementation, $p(\hat{y}|\mathbf{x})$ is approximated by the output of last softmax layer, i.e., $\Phi_c(\mathbf{x})$. Similarly, $p(\hat{y}|\tilde{\mathbf{x}})$ is approximated by $\Phi_c(\tilde{\mathbf{x}})$.

On the other hand, due to the Pinsker's inequality (Pinsker, 1964; Canonne, 2022) which implies that for any two distributions $p$ and $q$,

$$
\min\{D_{\text{KL}}(p; q), D_{\text{KL}}(q; p)\} \geq \frac{1}{2 \log 2} \|p - q\|_1^2,
\tag{11}
$$

from which we obtain $D_{\text{KL}}(\Phi_c(\tilde{\mathbf{x}}); \Phi_c(\mathbf{x})) \geq \frac{1}{2 \log 2} \|\Phi_c(\tilde{\mathbf{x}}) - \Phi_c(\mathbf{x})\|_1^2$.

To summarize, maximizing $I(\hat{y}; \tilde{\mathbf{x}})$ is equivalent to minimizing the expected Kullback–Leibler (KL) divergence $\mathbb{E}(D_{\text{KL}}(\Phi_c(\tilde{\mathbf{x}}); \Phi_c(\mathbf{x})))$, and it provides an upper-bound to other loss functions such as the $\ell_1$-loss $\mathbb{E}(\|\Phi_c(\tilde{\mathbf{x}}) - \Phi_c(\mathbf{x})\|_1^2)$.

To motivate the link between the compression term in the IB principle and the sparsity term $\mathbf{m}_1$ in the RDE objective, we begin by rewriting the mutual information $I(\mathbf{x}; \tilde{\mathbf{x}})$ as:

$$
I(\mathbf{x}; \tilde{\mathbf{x}}) = H(\mathbf{x}) + H(\tilde{\mathbf{x}}) - H(\mathbf{x}, \tilde{\mathbf{x}})
\tag{12}
$$

Since $H(\mathbf{x})$ is constant when optimizing for a sparse mask $\mathbf{m} \in [0, 1]^d$ within the IB principle, it can be dropped, yielding:

$$
I(\mathbf{x}; \tilde{\mathbf{x}}) = H(\tilde{\mathbf{x}}) - H(\mathbf{x}, \tilde{\mathbf{x}})
\tag{13}
$$

Using the conditional entropy definition, we get:

$$H(\mathbf{x}, \tilde{\mathbf{x}}) = H(\tilde{\mathbf{x}}|\mathbf{x}) + H(\mathbf{x}) \tag{14}$$

Dropping $H(\mathbf{x})$ for the same reason as above, we obtain:

$$I(\mathbf{x}; \tilde{\mathbf{x}}) = H(\tilde{\mathbf{x}}) - H(\tilde{\mathbf{x}}|\mathbf{x}) \tag{15}$$

Since $\tilde{\mathbf{x}} \subseteq \mathbf{x}$, the conditional entropy $H(\tilde{\mathbf{x}}|\mathbf{x})$ is small, leading to the approximation:

$$I(\mathbf{x}; \tilde{\mathbf{x}}) \approx H(\tilde{\mathbf{x}}) \tag{16}$$

This demonstrates that the mutual information $I(\mathbf{x}; \tilde{\mathbf{x}})$ can be effectively approximated by the entropy of the masked input, $H(\tilde{\mathbf{x}})$ (Strouse & Schwab, 2017; Kirsch et al., 2020). A practical way to minimize $H(\tilde{\mathbf{x}})$ is to reduce the number of explanatory variables, which corresponds to encouraging sparsity in $\mathbf{m}$ (Bang et al., 2021; Tao et al., 2020). This sparsity constraint is commonly represented by $\|\mathbf{m}\|_0$ and is approximated using $\|\mathbf{m}\|_1$ for optimization purposes. Thus, the sparsity term in the RDE objective conceptually aligns with the compression term in the IB principle

### A.2 EXPECTED RDE LOSS AT INITIALIZATION UNIFORM VERSUS GRADIENT-BASED INITIALIZATION

Let $\mathbf{x} \in \mathbb{R}^d$ denote the original input with dimensionality $d$. Further let $\mathbf{m}_{\text{unif}} \in [0, 1]^d$ be the uniformly initialized mask where each $m_i$ is independently sampled from $\mathcal{U}(0, 1)$. Similarly, denote $\mathbf{m}_{\text{grad}}$ the gradient-based initialized mask as described in section 4, where we transformed the gradient values using a quantile transformation function so that $\mathbf{m}_{\text{grad}} \in [0, 1]^d$. We aim to prove the following inequality:

$$\mathbb{E}_{\mathbf{m}_{\text{grad}}, \mathbf{u} \sim V} \left[ \mathcal{D} \left( \Phi_c(\tilde{\mathbf{x}}_{\text{grad}}), \Phi_c(\mathbf{x}) \right) + \lambda \|\mathbf{m}_{\text{grad}}\|_1 \right] \leq \mathbb{E}_{\mathbf{m}_{\text{unif}}, \mathbf{u} \sim V} \left[ \mathcal{D} \left( \Phi_c(\tilde{\mathbf{x}}_{\text{unif}}), \Phi_c(\mathbf{x}) \right) + \lambda \|\mathbf{m}_{\text{unif}}\|_1 \right]$$

$$\tag{17}$$

Here, $\tilde{\mathbf{x}}_{\text{grad}} = \mathbf{m}_{\text{grad}} \odot \mathbf{x} + (1 - \mathbf{m}_{\text{grad}}) \odot \mathbf{u}$ and $\tilde{\mathbf{x}}_{\text{unif}} = \mathbf{m}_{\text{unif}} \odot \mathbf{x} + (1 - \mathbf{m}_{\text{unif}}) \odot \mathbf{u}$ are the distorted versions of the input $\mathbf{x} \in \mathbb{R}^d$ with random perturbation $\mathbf{u} \in \mathbb{R}^d$ drawn from a predefined probability distribution $V$ such as Gaussian. Here, $\lambda$ represents a hyperparameter, encouraging sparsity in $\mathbf{m}_{\text{grad}}$ and $\mathbf{m}_{\text{unif}}$.

Let further denote $\Phi_c(\mathbf{x})$ a classifier's prediction for the input $\mathbf{x} \in \mathbb{R}^d$ where $\Phi_c : \mathbb{R}^d \to [0, 1]^c$ with $c \geq 1$ representing the number of classes. Assume that the classifier's prediction $\Phi_c(\mathbf{x})$ is differentiable and locally linear in the neighborhood of $\mathbf{x}$. Formally, this neighborhood is defined as the open ball $B_\epsilon(\boldsymbol{x}) = \{\tilde{\boldsymbol{x}} \in \mathbb{R}^d : \|\tilde{\boldsymbol{x}} - \boldsymbol{x}\| < \epsilon\}$, where $\epsilon > 0$ ensures that higher-order terms in the Taylor expansion of $\Phi_c$ are negligible.

Then, we can use the a first-order Taylor expansion around $\mathbf{x}$:

$$\Phi_c(\tilde{\mathbf{x}}) \approx \Phi_c(\mathbf{x}) + \nabla \mathbf{x} \Phi_c(\mathbf{x}) \cdot (\tilde{\mathbf{x}} - \mathbf{x}) \tag{18}$$

where $\nabla \mathbf{x} \Phi_c(\mathbf{x})$ represents the gradient of $\Phi_c(\mathbf{x})$ with respect to $\mathbf{x}$ and $\Delta \mathbf{x} = \tilde{\mathbf{x}} - \mathbf{x}$. The distortion $\mathcal{D}(\Phi_c(\tilde{\mathbf{x}}), \Phi_c(\mathbf{x}))$ can therefore be approximated, for a norm $\| \cdot \|_p$, as:

$$\mathcal{D} \left( \Phi_c(\tilde{\mathbf{x}}), \Phi_c(\mathbf{x}) \right) = \|\Phi_c(\tilde{\mathbf{x}}) - \Phi_c(\mathbf{x}))\|_p \approx \|\nabla_{\mathbf{x}} \Phi_c(\mathbf{x}) \cdot \Delta \mathbf{x}\|_p \tag{19}$$

Using this, we get the following expression:

$$\mathbb{E}_{\mathbf{m}_{\text{grad}},\mathbf{u}\sim V}\left[\|\nabla_{\mathbf{x}}\Phi_c(\mathbf{x})\cdot\Delta\mathbf{x}\|_p + \lambda\|\mathbf{m}_{\text{grad}}\|_1\right] \leq \mathbb{E}_{\mathbf{m}_{\text{unif}},\mathbf{u}\sim V}\left[\|\nabla_{\mathbf{x}}\Phi_c(\mathbf{x})\cdot\Delta\mathbf{x}\|_p + \lambda\|\mathbf{m}_{\text{unif}}\|_1\right] \quad (20)$$

Since $\mathbf{m}_{\text{uniform}}, \mathbf{m}_{\text{grad}} \sim \mathcal{U}(0,1)^d$, the terms $\mathbb{E}_{\mathbf{m}_{\text{grad}}}[\lambda\|\mathbf{m}_{\text{grad}}\|_1]$ and $\mathbb{E}_{\mathbf{m}_{\text{unif}}}[\lambda\|\mathbf{m}_{\text{unif}}\|_1]$ simplify to $\frac{\lambda}{2}d$ as the expected value for each $m_i$ is $\frac{1}{2}$. Both sparsity terms can be omitted from the comparison since they are equal. Therefore, we only need to show that:

$$\mathbb{E}_{\mathbf{m}_{\text{grad}},\mathbf{u}\sim V}\left[\|\nabla_{\mathbf{x}}\Phi_c(\mathbf{x})\cdot(\tilde{\mathbf{x}}_{\text{grad}}-\mathbf{x})\|_p\right] \leq \mathbb{E}_{\mathbf{m}_{\text{unif}},\mathbf{u}\sim V}\left[\|\nabla_{\mathbf{x}}\Phi_c(\mathbf{x})\cdot(\tilde{\mathbf{x}}_{\text{unif}}-\mathbf{x})\|_p\right]. \quad (21)$$

holds. By using the definitions for $\tilde{\mathbf{x}}_{\text{grad}}$ and $\tilde{\mathbf{x}}_{\text{unif}}$, we get the following after rearranging terms:

$$\tilde{\mathbf{x}}_{\text{grad}} - \mathbf{x} = (\mathbf{m}_{\text{grad}}-1)\odot(\mathbf{x}-\mathbf{u}), \quad (22)$$
$$\tilde{\mathbf{x}}_{\text{unif}} - \mathbf{x} = (\mathbf{m}_{\text{unif}}-1)\odot(\mathbf{x}-\mathbf{u}). \quad (23)$$

Hence, we can rearrange 21:

$$\mathbb{E}_{\mathbf{m}_{\text{grad}},\mathbf{u}\sim V}\left[\|\nabla_{\mathbf{x}}\Phi_c(\mathbf{x})\odot(\mathbf{m}_{\text{grad}}-1)\odot(\mathbf{x}-\mathbf{u})\|_p\right] \leq \mathbb{E}_{\mathbf{m}_{\text{unif}},\mathbf{u}\sim V}\left[\|\nabla_{\mathbf{x}}\Phi_c(\mathbf{x})\odot(\mathbf{m}_{\text{unif}}-1)\odot(\mathbf{x}-\mathbf{u})\|_p\right]$$
$$(24)$$

The key observation is that the gradient-based mask $\mathbf{m}_{\text{grad}}$ is constructed to minimize the distortion by assigning values of $m_i$ close to 1 when $\nabla_{\mathbf{x}_i}\Phi_c(\mathbf{x})$ is large, thus reducing the effect of perturbations in important directions. On the other hand, the uniform mask $\mathbf{m}_{\text{unif}}$ is assigned independently of the gradient, making it less likely to reduce distortion effectively. Consequently, the gradient-based method is expected to achieve lower distortion compared to the uniform initialization *at initialization* by design, thereby validating the claim.

This completes the proof. $\square$

### A.3 EXPECTED RDE LOSS AT INITIALIZATION ALL-ONES VERSUS GRADIENT-BASED INITIALIZATION

Let $\mathbf{m}_{\text{ones}}$ denote the mask initialized with all ones, i.e. $\mathbf{m}_{\text{ones}} = \mathbf{1}_d$ where $\mathbf{1}_d$ is a vector of ones with dimension $d$. Let further denote $\Phi_c(\mathbf{x})$ a classifier's prediction for the input $\mathbf{x} \in \mathbb{R}^d$ where $\Phi_c : \mathbb{R}^d \to [0,1]^c$ with $c \geq 1$ representing the number of classes. Given the function $\mathcal{D} : [0,1]^c \times [0,1]^c \to \mathbb{R}_+$ that measures the distortion between the classifier's prediction for the original input $\mathbf{x}$ and the 'distorted' input $\tilde{\mathbf{x}}$, the all-ones initialization results in $\mathcal{D}(\Phi_c(\mathbf{x}), \Phi_c(\tilde{\mathbf{x}})) = 0$ as we have $\mathbf{x} = \tilde{\mathbf{x}}$. Furthermore, let $\mathbf{m}_{\text{grad}} \sim \mathcal{U}(0,1)^d$ denote the mask initialized using the gradient-based scheme as described in section 4. We aim to show that

$$\mathbb{E}_{\mathbf{m}_{\text{grad}},\mathbf{u}\sim V}\left[\mathcal{D}(\Phi_c(\tilde{\mathbf{x}}),\Phi_c(\mathbf{x})) + \lambda\|\mathbf{m}_{\text{grad}}\|_1\right] \leq \lambda\|\mathbf{m}_{\text{ones}}\|_1 \quad (25)$$

holds, where $\lambda$ represents a hyperparameter balancing the trade-off between distortion and sparsity. Since $\mathbf{m}_{\text{ones}}$ is initialized with all ones, we have $\|\mathbf{m}_{\text{ones}}\|_1 = d$. For $\mathbf{m}_{\text{grad}} \sim \mathcal{U}(0,1)^d$, the expected value of the $l_1$-norm is given by $\mathbb{E}_{\mathbf{m}_{\text{grad}}}[\|\mathbf{m}_{\text{grad}}\|_1] = \frac{d}{2}$. Therefore, we can simplify the inequality as follows:

$$\mathbb{E}_{\mathbf{m}_{\text{grad}},\mathbf{u}\sim V}\left[\mathcal{D}(\Phi_c(\tilde{\mathbf{x}}),\Phi_c(\mathbf{x}))\right] + \frac{\lambda}{2}d \leq \lambda d \quad (26)$$

$$\mathbb{E}_{\mathbf{m}_{\text{grad}},\mathbf{u}\sim V}\left[\mathcal{D}(\Phi_c(\tilde{\mathbf{x}}),\Phi_c(\mathbf{x}))\right] \leq \frac{\lambda}{2}d \quad (27)$$

If we use a norm $\|\cdot\|_p$ with $p \geq 1$ as a choice for the distortion function $\mathcal{D}$, we can upper bound the expected distortion on the left-hand side of equation in the following way:

$$\mathbb{E}_{\mathbf{m}_{\text{grad}}, \mathbf{u} \sim V} \left[ \mathcal{D} \left( \Phi_c(\tilde{\mathbf{x}}), \Phi_c(\mathbf{x}) \right) \right] \leq \left( 1 + (2-1) \mathbb{1}_{c \geq 2} \right)^{\frac{1}{p}} \leq \frac{\lambda}{2} d \tag{28}$$

as $\Phi_c : \mathbb{R}^d \to [0,1]^c$ for the classification setting we consider, where $\sum_{i=1}^{c} \Phi_{c,i}(x) = 1$ for any input $x \in \mathbb{R}^d$. Therefore, the gradient-based mask initialization has a lower expected loss in terms of distortion and sparsity *at initialization*, if

$$\frac{2 \left( 1 + (2-1) \mathbb{1}_{c \geq 2} \right)^{\frac{1}{p}}}{d} \leq \lambda \tag{29}$$

holds. In high-dimensional settings, which are typical for deep learning models and, consequently, for mask-based explanation methods, this condition is generally satisfied.

This completes the proof. $\qquad\square$

# B PSEUDOCODE

## B.1 GENERAL VERSION OF THE STARTGRAD ALGORITHM

While we assumed in the pseudocode for Algorithm 1 that we operate in the same input space $\mathcal{X}$, we outline in Algorithm 2 a more general version of the StartGrad algorithm, where we first use an invertible mapping $\mathcal{F}$.

---

**Algorithm 2:** General version: Gradient-based Mask Initialization (StartGrad)

---

**Input** : Pre-trained classifier $\Phi_c$, input samples $\{\boldsymbol{x}_i\}_{i=1}^N$ with $\boldsymbol{x}_i \in \mathbb{R}^d$, quantile transformation $\mathcal{T} : \mathbb{R}_+^d \to [0,1]^d$, invertible and differentiable function $\mathcal{F} : \mathbb{R}^d \to \mathbb{R}^k$

**Hyperparameters:** Output distribution $\mathcal{U}(0,1)$, number of quantiles (bins) for $\mathcal{T}$

**Output** : Masks $\mathbf{M} \in [0,1]^{N \times d}$

1 Initialize mask list $\mathbf{M} := []$ and quantile transformer $\mathcal{T}$;
2 **for** $i \leftarrow 1$ **to** $N$ **do**
3      $c_i \leftarrow \arg\max(\Phi_c(\boldsymbol{x}_i))$ // Class prediction
4      $\mathbf{S}_i \leftarrow \left|\nabla_{\mathcal{F}(\boldsymbol{x}_i)}\Phi_c(\boldsymbol{x}_i)\right|_c$ // Gradient magnitudes
5      $\mathbf{m}_i \leftarrow \mathcal{T}(\mathbf{S}_i)$ // Quantile transform
6      Append $\mathbf{m}_i$ to $\mathbf{M}$;
7 **end for**
8 **return** $\mathbf{M}$

---

Figure 2 visually illustrates the pseudocode.

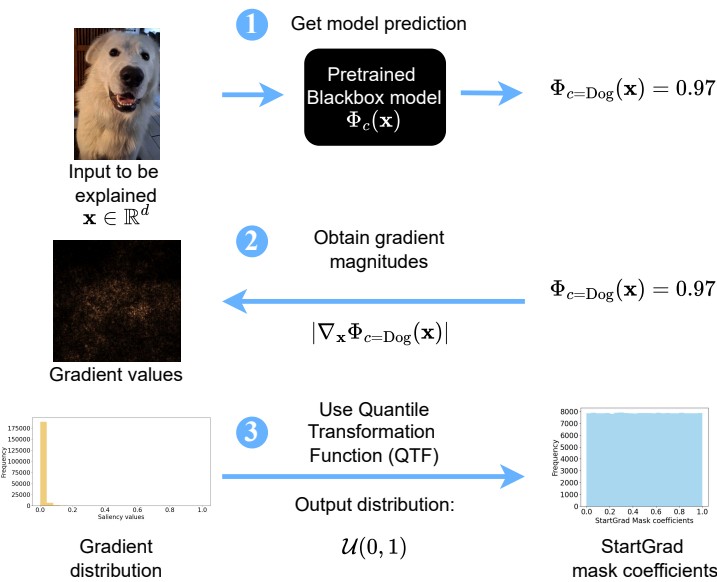

Figure 2: Visual illustration of the StartGrad algorithm.

## C  EXPERIMENTAL SETTINGS AND DETAILS

### C.1  VISION

Here we describe the explanation methods used for the vision part of the experiments in more detail. We use the same notation as already introduced in section 3.

For all vision experiments, we optimize the mask coefficients for 300 steps, using an Adam optimizer (Kingma & Ba, 2014) with learning rate of $10^{-1}$.

#### C.1.1  PIXELMASK

The objective for the PixelMask model as first introduced by Fong & Vedaldi (2017) is:

$$\min_{\mathbf{m}\in[0,1]^k} E_{\mathbf{u}\sim V}\left[\mathcal{D}\left(\Phi_c(\tilde{\mathbf{x}}), \Phi_c(\mathbf{x})\right)\right] + \lambda_1\|\mathbf{m}\|_1 \tag{30}$$

We set $\lambda_1 = 1$, and estimate the expectation via Monte Carlo sampling across 16 samples, where we draw the perturbation $u$ from an uniform distribution. In particular, we use an uniform noise from $[\mu - \sigma, \mu + \sigma]$, where $\mu$ and $\sigma$ are the empirical mean and standard deviation of the pixel values of the image as it is been done in (Kolek et al., 2022; 2023).

For the PixelMask (Fong & Vedaldi, 2017) model, we use 65,536 bins for the QTF function in Algorithm 1 which equals the width and the height (256 by 256) of the images used for the pretrained vision models.

#### C.1.2  SHEARLETX & WAVELETX

For the ShearletX (Kolek et al., 2023) and WaveletX model (Kolek et al., 2023), we utilized the original implementations and follow their experimental setting closely.

With the same notation as used in section 3, Kolek et al. (2023) extend equation 30 in the following manner:

$$\min_{\mathbf{m}\in[0,1]^k} E_{\mathbf{u}\sim V}\left[\mathcal{D}\left(\Phi_c(\tilde{\mathbf{x}}), \Phi_c(\mathbf{x})\right)\right] + \lambda_1\|\mathbf{m}\|_1 + \lambda_2\|\mathcal{F}^{-1}\left(\mathbf{m}\odot\mathcal{F}(\mathbf{x})\right)\|_1 \tag{31}$$

where $\tilde{\mathbf{x}} := \mathcal{F}^{-1}(\mathbf{m}\odot\mathcal{F}(\mathbf{x})+(1-\mathbf{m})\odot\mathbf{u})$ denotes now the distorted input image after applying the mask in the transformed space. The WaveletX model uses for the $\mathcal{F}$ the discrete wavelet transform (DWT), whereas the ShearletX makes use of the digital shearlet transform (Kutyniok et al., 2016; Kolek et al., 2023). For the WaveletX we use the daubechies-3 as a mother wavelet with five scales.

The expectation in 31 is approximated using Monte Carlo sampling using 16 samples where the perturbation for scale $a$ and shearing $s$ is sampled uniformly from $[\mu_{a,s} - \sigma_{a,s}, \mu_{a,s} + \sigma_{a,s}]$. Here $\sigma_{a,s}$ and $\mu_{a,s}$ are the empirical standard deviation and mean of the image's shearlet coefficients at scale $a$ and shearing $s$. In line with Kolek et al. (2023), we set for ShearletX $\lambda_1 = 1$ and $\lambda_2 = 2$. For WaveletX we set $\lambda_1 = 1$ and $\lambda_2 = 10$. We use the mean-squared error as a choice for the distortion function $\mathcal{D}$.

For the ShearletX and WaveletX (Kolek et al., 2023), we use 10,000 as the number of bins for the former, and the following bins for each of the scales, 144, 16900, 4489, 1296, 400, 144 for the latter.

### C.2  VISION EVALUATION METRICS

#### C.2.1  CP-L1 AND CP-PIXEL SCORE

As outlined in section 5, we use the conciseness-preciseness (CP) Pixel and L1 score first introduced by Kolek et al. (2023) as quantitative performance metrics which are defined as follows:

$$CP = \frac{\text{Retained class probability}}{\text{Retained image information}} \quad (32)$$

where the numerator is defined by:

$$\text{Retained class probability} = \frac{\Phi_c(\tilde{\mathbf{x}})}{\Phi_c(\mathbf{x})} \quad (33)$$

while the denominator in case of the CP-Pixel is defined as:

$$\text{Retained image information Pixel} = \frac{\|\mathbf{m} \odot \mathbf{x}\|_1}{\|\mathbf{x}\|_1} \quad (34)$$

whereas for the CP-L1 we have:

$$\text{Retained image information L1} = \frac{\|\mathbf{m} \odot \mathbf{c}\|_1}{\|\mathbf{c}\|_1} \quad (35)$$

where $\mathbf{c} \in \mathbb{R}^k$ represents the coefficients of the latent space such as the shearlet or wavelet space. Note that for the PixelMask model, the CP-L1 and CP-Pixel metrics are identical as the masking is only applied in the pixel space.

Given the aforementioned formulations of the CP scores, one can notice that these scores strive to balance both, preciseness and sparsity of the explanation. Higher scores indicate superior masks (Kolek et al., 2023).

### C.2.2 JUSTIFICATION OF CP-L1 AND CP-PIXEL SCORE AS OPPOSED TO DELETION AND INSERTION SCORE

Attribution methods like Integrated Gradients (Sundararajan et al., 2017) and SmoothGrad (Smilkov et al., 2017) are typically evaluated using insertion and deletion scores. Unlike the mask-based explanation methods in this study, they provide a clear feature importance ranking, usually in pixel space. In contrast, mask-based methods do not inherently rank coefficient relevance but instead optimize a binary, sparse mask using a relaxed $l_1$ norm to approximate the ideal $l_0$ norm. This binary nature complicates insertion and deletion scores, which require an ordered ranking. This issue was previously highlighted by Kolek et al. (2023), who introduced CP-scores; we refer to their work for further discussion.

To highlight the issues with using the insertion and deletion metrics to evaluate mask-based explanation methods, we computed the faithfulness scores which is the difference between the former two for PixelMask (Fong & Vedaldi, 2017), WaveletX, and ShearletX (Kolek et al., 2023) across multiple iterations, as shown in the following figure. As we can see from the figure 3 the faithfulness scores decreases with increasing iteration steps across all three methods and all initializations and reach their maximum scores at very early iterations steps which is in sharp contrast to the results obtained using the CP-scores and also counterintuitive as this implies that the learned mask gets less faithful over time. This illustrates the problem of faithfulness score which presumes an inherent ordering of the mask values, even though the mask-based explanation methods optimize a relaxed binary problem with no inherent ordering. For comparison reasons, we included two non-masked based attribution methods Integrated Gradients (Sundararajan et al., 2017) and GradCAM (Selvaraju et al., 2017)

For completeness, Table 4 includes a comparison between the two aforementioned gradient-based methods and the mask-based explanation methods used in this study using the CP-Pixel and CP-L1 evaluation metric. The results clearly demonstrate that mask-based explanation methods consistently outperform non-mask-based methods across all iteration steps. However, it is important to note that mask-based methods, with their learned binary mask, are explicitly designed to balance distortion and sparsity, giving them an advantage in this trade-off.

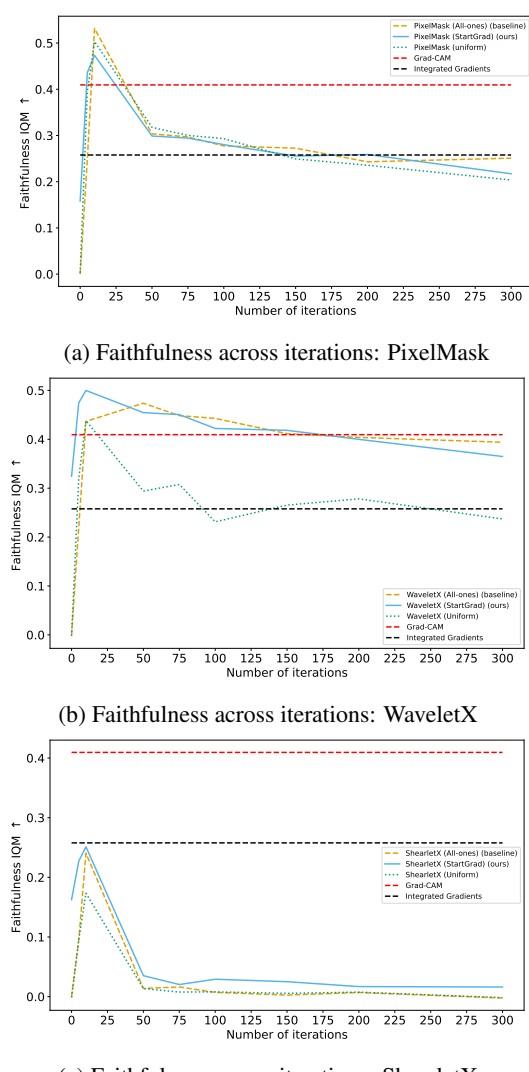

(a) Faithfulness across iterations: PixelMask

(b) Faithfulness across iterations: WaveletX

(c) Faithfulness across iterations: ShearletX

Figure 3: Faithfulness scores across iteration for the PixelMask (Fong & Vedaldi, 2017) (a), WaveletX (Kolek et al., 2023) (b), and ShearletX (Kolek et al., 2023) (c) models, comparing three initialization methods: StartGrad (ours) (solid line), All-ones (dashed line), and Uniform (dotted line). For comparison reasons, we added two non-masked based explanation methods, i.e. Gradient-weighted Class Activation Mapping (Grad-CAM) (Selvaraju et al., 2017) and Integrated Gradients (Sundararajan et al., 2017). As these methods are not iteratively optimized, the faithfulness values for these methods remain constant. The curves represent the interquartile mean (IQM) across 100 randomly selected ImageNet validation samples (Deng et al., 2009), evaluated with a pretrained ResNet18 classifier (He et al., 2016). ↑ indicates that higher is better. We can see that the faithfulness score decreases across all methods and initializations over the course of optimization which is counterintuitive and in contrast to the CP-scores, thereby illustrating the problem associated with using faithfulness scores as an evaluation metric for mask-based explanation methods that do not inherently order their mask-values according to importance.

Table 4: Interquartile mean score (IQM) for the PixelMask (Fong & Vedaldi, 2017), WaveletX (Kolek et al., 2023), ShearletX (Kolek et al., 2023), Integrated Gradients (IG) (Sundararajan et al., 2017), Gradient-weighted Class Activation Mapping (Grad-CAM) (Selvaraju et al., 2017) of the CP-Pixel and CP-L1 score across iteration steps. All mask-based methods are initialized with Start-Grad. To map the gradient-based attributions (IG, GradCAM) into the desired range of $[0, 1]$, either a min-max normalization (Min-Max) or a quantile transformation function (QTF) were used. All experiments use 500 randomly selected ImageNet (Deng et al., 2009) samples evaluated on a pre-trained ResNet18 (He et al., 2016) classifier. For each metric, $\uparrow$ indicates that higher is better, and $\downarrow$ that lower is better. Values that are two standard errors away from the second-best values are highlighted in bold. As IG (Sundararajan et al., 2017) and GradCAM (Selvaraju et al., 2017) are both methods that are not optimized iteratively, their scores remain constant across the shown iteration steps. The CP-Pixel and CP-L1 scores differ only for the WaveletX (Kolek et al., 2023) and ShearletX (Kolek et al., 2023), as only these methods operate also in a latent space.

| Metric | Initialization | Iteration steps | | |
| | | 50 | 100 | 300 |
|---|---|---|---|---|
| | PixelMask | $37.395 \pm 1.119$ | $51.217 \pm 1.478$ | $65.766 \pm 1.767$ |
| | WaveletX | $48.973 \pm 0.604$ | $64.551 \pm 0.809$ | $77.534 \pm 0.976$ |
| | ShearletX | $\mathbf{216.908 \pm 6.997}$ | $\mathbf{359.803 \pm 9.675}$ | $\mathbf{481.314 \pm 11.829}$ |
| CP-Pixel $\uparrow$ | IG (Min-Max) | $3.237 \pm 0.015$ | $3.237 \pm 0.015$ | $3.237 \pm 0.015$ |
| | IG (QTF) | $2.037 \pm 0.001$ | $2.037 \pm 0.001$ | $2.037 \pm 0.001$ |
| | GradCAM (Min-Max) | $2.841 \pm 0.019$ | $2.841 \pm 0.019$ | $2.841 \pm 0.019$ |
| | GradCAM (QTF) | $1.987 \pm 0.004$ | $1.987 \pm 0.004$ | $1.987 \pm 0.004$ |
| | PixelMask | $37.395 \pm 1.119$ | $51.217 \pm 1.477$ | $65.766 \pm 1.767$ |
| | WaveletX | $11.208 \pm 0.128$ | $14.505 \pm 0.173$ | $17.484 \pm 0.204$ |
| | ShearletX | $\mathbf{123.791 \pm 4.760}$ | $\mathbf{199.193 \pm 6.384}$ | $\mathbf{270.438 \pm 7.931}$ |
| CP-L1 $\uparrow$ | IG (Min-Max) | $3.237 \pm 0.015$ | $3.237 \pm 0.015$ | $3.237 \pm 0.015$ |
| | IG (QTF) | $2.037 \pm 0.001$ | $2.037 \pm 0.001$ | $2.037 \pm 0.001$ |
| | GradCAM (Min-Max) | $2.841 \pm 0.019$ | $2.841 \pm 0.019$ | $2.841 \pm 0.019$ |
| | GradCAM (QTF) | $1.987 \pm 0.004$ | $1.987 \pm 0.004$ | $1.987 \pm 0.004$ |

## C.3 TIME SERIES

### C.3.1 STATE DATASET

The dataset for the synthetic state dataset as introduced by Tonekaboni et al. (2020) generates data according to a 2-state hidden Markov model (HMM). Let $t \in [t : T]$ represent the discrete time steps and $s_t \in \{0, 1\}$ denote the specific states. For this dataset, the input $\mathbf{x} \in \mathbb{R}^3$ is generated using a multivariate Normal distribution $x_t \sim \mathcal{N}(\mu_{s_t}, \Sigma_{s_t})$, where the mean $\mu_{s_t}$ and the variance-covariance matrix $\Sigma_{s_t}$ are dependent on the state at $s_t$. In particular, we set $\mu_0 = [0.1, 1.6, 0.5]$ and $\mu_1 = [-0.1, -0.4, -1.5]$. The variance-covariance matrix is given by:

$$\Sigma = \begin{bmatrix} 0.8 & 0.01 & 0.01 \\ 0.01 & 0.8 & 0.0 \\ 0.01 & 0.0 & 0.8 \end{bmatrix}$$

The first feature dimension is irrelevant for the label $y_t$, whereas the label is distributed as:

$$y_t \sim \text{Bernoulli}(p_t)$$

$$p_t = \begin{cases} (1 + \exp[-2x_{2,t}])^{-1} & \text{if } s_t = 0 \\ (1 + \exp[-2x_{3,t}])^{-1} & \text{if } s_t = 1 \end{cases} \tag{36}$$

The transition probabilities are given by:

$$T = \begin{bmatrix} 0.1 & 0.9 \\ 0.01 & 0.9 \end{bmatrix}$$

with the starting state at $t = 0$ being determined with equal probability, i.e. $\pi = [\frac{1}{2}, \frac{1}{2}]$.

For the experiments, we generate 1,000 time series, 800 for training, and 200 for testing.

### C.3.2 SWITCH-FEATURE DATASET

Following the setup introduced by Tonekaboni et al. (2020), the switch-feature dataset uses a Gaussian process mixture in place of the multivariate normal distribution with means $\mu$

$$\mu = \begin{bmatrix} 0.8 & 0.5 & 0.2 \\ 0.0 & 1.0 & 0.0 \\ 0.2 & 0.2 & 0.8 \end{bmatrix}$$

The Gaussian process over time is governed by a RBF kernel with $\gamma = 0.2$, with marginal feature set to 0.1 for all states. In a similar vein to the state dataset, the label is distributed as:

$$y_t \sim \text{Bernoulli}(p_t)$$

$$p_t = \begin{cases} (1 + \exp[-3x_{1,t}])^{-1} & \text{if } s_t = 0 \\ (1 + \exp[-3x_{2,t}])^{-1} & \text{if } s_t = 1 \\ (1 + \exp[-3x_{3,t}])^{-1} & \text{if } s_t = 2 \end{cases} \tag{37}$$

The transition probabilities are given by:

$$T = \begin{bmatrix} 0.95 & 0.02 & 0.3 \\ 0.02 & 0.95 & 0.03 \\ 0.03 & 0.02 & 0.95 \end{bmatrix}$$

with the starting state at $t = 0$ being determined with equal probability, i.e. $\pi = [\frac{1}{3}, \frac{1}{3}, \frac{1}{3}]$.

For the experiments, we generate 1,000 time series, 800 for training, and 200 for testing.

### C.4 EXTREMALMASK

ExtremalMask (Enguehard, 2023) is a mask-based explanation method specifically designed for multivariate time-series data and is defined by:

$$\min_{\mathbf{m} \in [0,1]^{dxT}, \theta} \mathcal{D}\left(\Phi_c(\tilde{\mathbf{x}}), \Phi_c(\mathbf{x})\right) + \lambda_1 \|\mathbf{m}\|_1 + \lambda_2 \|f_\theta(\mathbf{x})\|_1 \tag{38}$$

where $\tilde{\mathbf{x}} = \mathbf{m} \odot \mathbf{x} + (1 - \mathbf{m}) \odot f_\theta(\mathbf{x})$ and $\mathbf{x} \in \mathbb{R}^{dxT}$ where $d$ is the input dimensionality, and $T$ represents the time dimension, respectively.

Equation 38 denotes the preservation game which aims to find the sparsest mask that preserves the most class probability. In the deletion game formulation of ExtremalMask (Enguehard, 2023), the objective function is defined as:

$$\min_{\mathbf{m} \in [0,1]^{dxT}, \theta} \mathcal{D}\left(\Phi_c(\tilde{\mathbf{x}}), \Phi_c(\mathbf{0})\right) + \lambda_1 \|1 - \mathbf{m}\|_1 + \lambda_2 \|f_\theta(\mathbf{x})\|_1 \tag{39}$$

In line with the experiments conducted in Enguehard (2023), we use a one-layer, bidirectional GRU (Cho et al., 2014) model with hidden dimensionality of 200 as choice for the $f_\theta$. Furthermore, we

use the Adam (Kingma & Ba, 2014) optimizer with learning rate $10^{-1}$. We run the ExtremalMask (Enguehard, 2023) model for 500 iterations. We use the mean-squared error as a choice for the distortion function.

As opposed to the original formulation of the distortion objective 39, we use a slightly modified version of the objective which has shown to significantly boost the performance (Bant et al., 2024):

$$\min_{\mathbf{m}\in[0,1]^{d_x T},\theta} \mathcal{D}\left(\Phi_c(\tilde{\mathbf{x}}), \Phi_c(\mathbf{0})\right) + \lambda_1\|1 - \mathbf{m}\|_1 + \lambda_2\|f_\theta(\mathbf{x}) - \mathbf{x}\|_2^2 \tag{40}$$

For all the experiments, we fit a single layer one-directional GRU (Cho et al., 2014) as a baseline classification model with hidden dimension 200 and train it for 50 epochs, batch size 128, learning rate of $1e - 4$ with the Adam (Kingma & Ba, 2014) optimizer. We minimize the cross-entropy loss for training the GRU model.

For both time-series datasets, we use 600 (which is equal to the signal length times the feature dimension) as the number of bins for the Quantile Transformation Function (QTF) in our StartGrad algorithm 2.

## C.5 TIME-SERIES EVALUATION METRICS

Given that the ground truth salient features are know in our synthetic dataset, we use the two standard performance metric area under recall (AUR) and area under precision (AUP). We further evaluate StartGrad using two additional metrics, information and mask entropy, with

$$\mathbf{I}_M(A) = -\sum_{t,i\in A} \log(1 - m_{t,i}) \tag{41}$$

for $A \subseteq [1 : T] \times [1 : d_X]$ defining the information, and

$$\mathbf{S}_M(A) = -\sum_{t,i\in A} m_{t,i}\log(m_{t,i}) + (1 - m_{t,i})\log(1 - m_{t,i}) \tag{42}$$

the entropy respectively (Crabbé & Van Der Schaar, 2021). A higher score indicates a better score for information, whereas a lower score indicates a better performance with respect to entropy.

# D  VISION EXPERIMENTS

## D.1  DISTRIBUTION OF ABSOLUTE GRADIENT VALUES

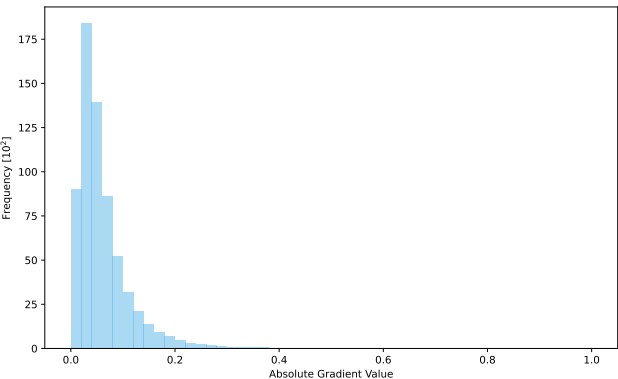

Figure 4: Absolute gradient values for the input pixels for 100 randomly selected validation ImageNet (Deng et al., 2009) samples using a pretrained ResNet18 (He et al., 2016) model. Here, the gradient was taken with respect to the class prediction.

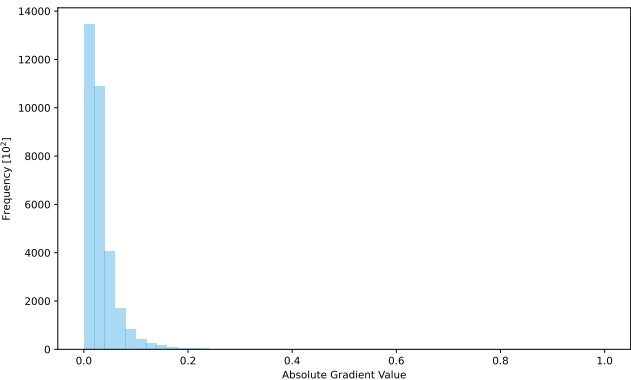

Figure 5: Absolute gradient values in the shearlet domain for the input pixels for 100 randomly selected validation ImageNet (Deng et al., 2009) samples using a pretrained ResNet18 (He et al., 2016) model. Here, the gradient was taken with respect to the class prediction.

## D.2  RUNTIME ANALYSIS

On a Apple Macbook Pro with a M1 chip, we ran the StartGrad algorithm for all three vision models, i.e. PixelMask (Fong & Vedaldi, 2017), WaveletX (Kolek et al., 2023) and ShearletX (Kolek et al., 2023) across 100 randomly selected ImageNet validation samples (Deng et al., 2009) and obtain the following average runtime (in seconds):

- PixelMask: 3.62s
- WaveletX: 0.72s
- ShearletX: 1.31s

## D.3 Additional Figures for Vision Experiments

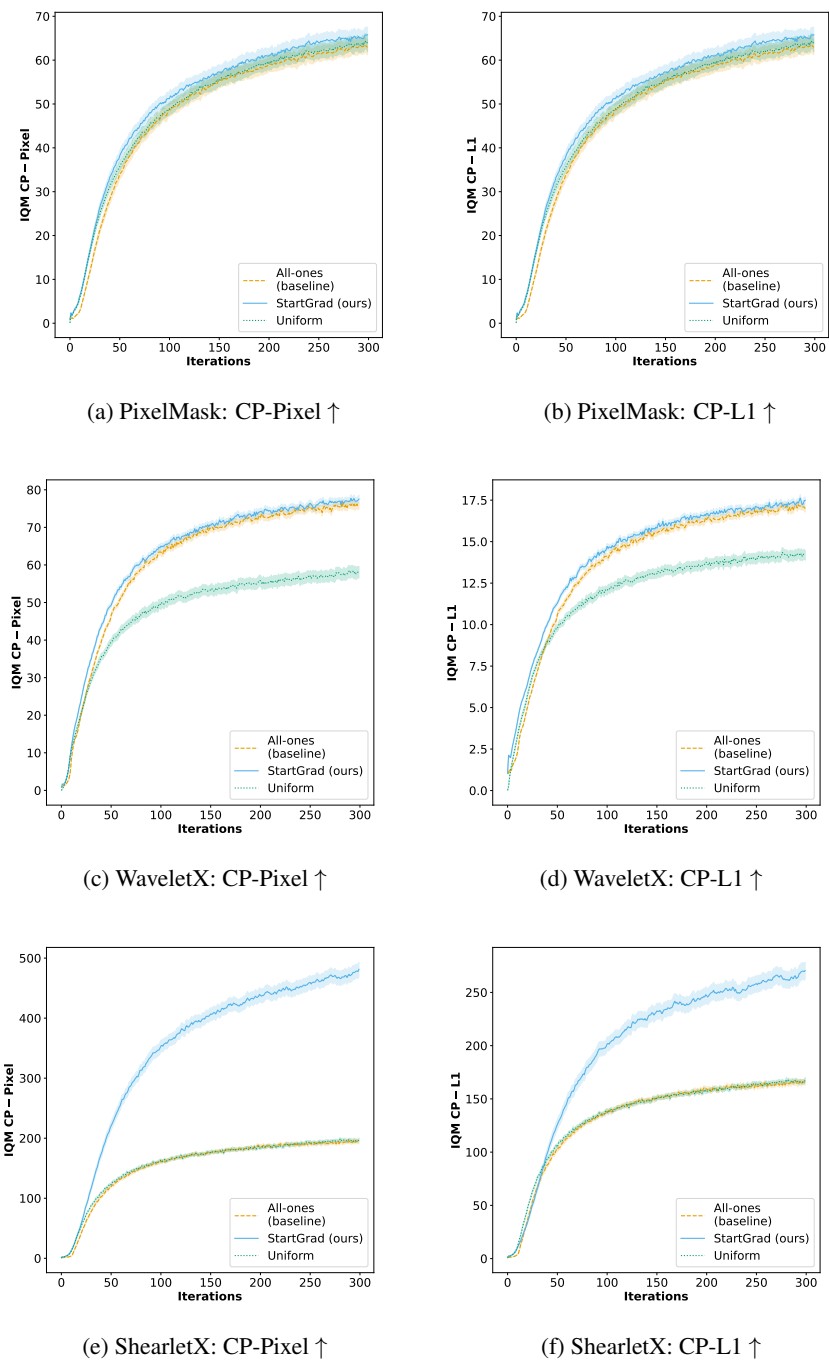

(a) PixelMask: CP-Pixel ↑

(b) PixelMask: CP-L1 ↑

(c) WaveletX: CP-Pixel ↑

(d) WaveletX: CP-L1 ↑

(e) ShearletX: CP-Pixel ↑

(f) ShearletX: CP-L1 ↑

Figure 6: Comparison of StartGrad initialization (ours) with standard baseline initialization schemes for the PixelMask (Fong & Vedaldi, 2017) (first row), WaveletX (Kolek et al., 2023) (second row) and ShearletX (Kolek et al., 2023) (third row). Baseline refers to the originally used initialization scheme for the respective mask explanation method. The solid line represents the average of the interquartile mean (IQM) performance across 500 randomly selected validation ImageNet (Deng et al., 2009) samples, while the shaded area denotes the standard errors respectively. ↑ indicates that higher is better, and ↓ that lower is better. We use a pretrained ResNet18 (He et al., 2016) classifier.

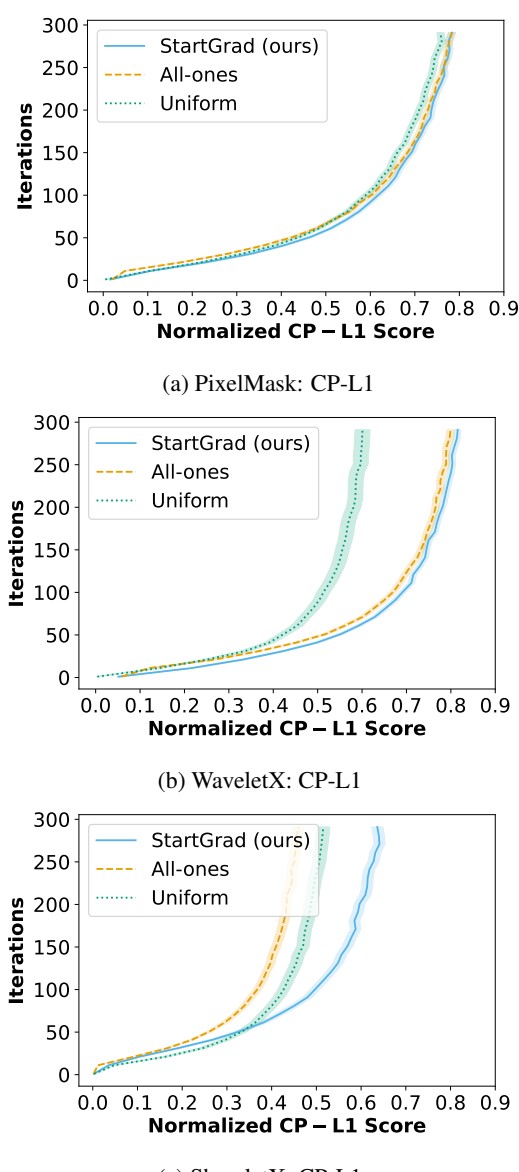

(a) PixelMask: CP-L1

(b) WaveletX: CP-L1

(c) ShearletX: CP-L1

Figure 7: Normalized CP-L1 score vs. iteration steps for the PixelMask (Fong & Vedaldi, 2017) (a), WaveletX (Kolek et al., 2023) (b), and ShearletX (Kolek et al., 2023) (c) models, comparing three initialization methods: StartGrad (ours) (solid line), All-ones (dashed line), and Uniform (dotted line). The curves represent the average number of iteration steps needed to reach a normalized target CP-L1 score. This target score is defined as the highest CP-L1 value achieved across all three initialization methods for each model, then normalized by the overall maximum score observed. The shaded regions represent the standard error across 500 randomly selected ImageNet validation samples (Deng et al., 2009)xs, with a pretrained ResNet18 (He et al., 2016) classifier. StartGrad enables all three models to achieve target scores with fewer iterations compared to standard initialization schemes, with the effect being most notable for ShearletX (Kolek et al., 2023).

# E    TIME-SERIES EXPERIMENTS

In this section, the main results from the time-series experiments are presented which were omitted from the main text for the sake of brevity. A complete description of the datasets, models, hyperparameters and metrics used can be found in the Appendix C.3.

## E.1    FIGURES

### E.1.1    STATE DATASET

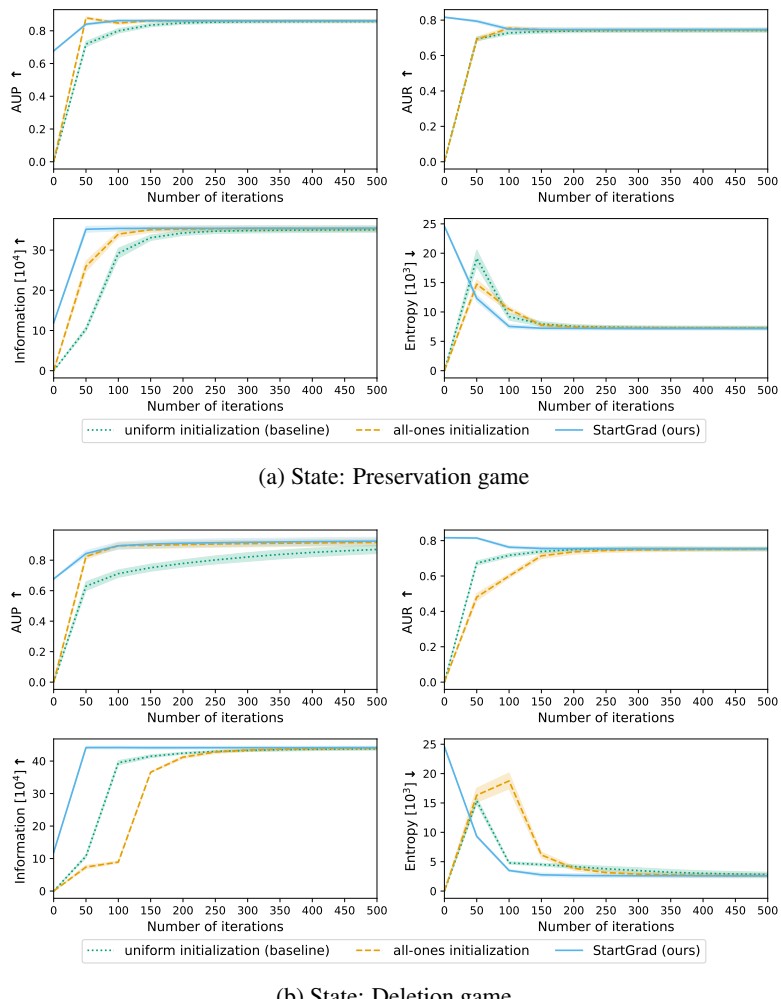

(a) State: Preservation game

(b) State: Deletion game

Figure 8: Comparison of the performance of the StartGrad (ours), all-ones and uniform (baseline) initialization on the state synthetic dataset. A one-layer GRU (Cho et al., 2014) is employed as a classifier, whereas the ExtremalMask (Enguehard, 2023) is used as a mask-based explanation method. The first panel (a) shows the results for the preservation game formulation of the ExtremalMask (Enguehard, 2023) model, where the goal is to preserve class probability as possible, while returning a sparse mask. The second panel (b) shows the results for the deletion game formulation of the ExtremalMask (Enguehard, 2023) model, where the objective is to find the least possible changes to the original input that distorts the class probability maximally. The solid line represents the average across five runs, with the shaded area highlighting the standard deviation. The preservation and deletion games objective versions are shown for both datasets. For each metric, ↑ indicates that higher is better, and ↓ that lower is better.

### E.1.2    SWITCH-FEATURE DATASET

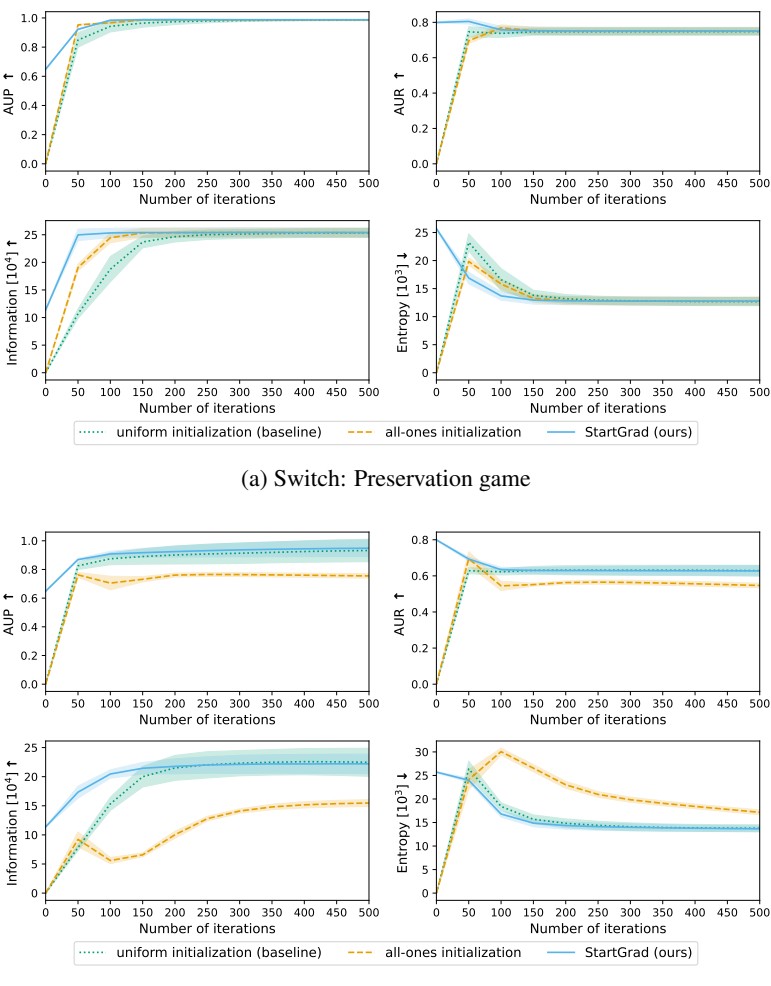

(a) Switch: Preservation game

(b) Switch: Deletion game

Figure 9: Comparison of the performance of the StartGrad (ours), all-ones and uniform (baseline) initialization on the switch-feature synthetic dataset. A one-layer GRU (Cho et al., 2014) is employed as a classifier, whereas the ExtremalMask (Enguehard, 2023) is used as a mask-based explanation method. The first panel (a) shows the results for the preservation game formulation of the ExtremalMask (Enguehard, 2023) model, where the goal is to preserve class probability as possible, while returning a sparse mask. The second panel (b) shows the results for the deletion game formulation of the ExtremalMask (Enguehard, 2023) model, where the objective is to find the least possible changes to the original input that distorts the class probability maximally. The solid line represents the average across five runs, with the shaded area highlighting the standard deviation. The preservation and deletion games objective versions are shown for both datasets. For each metric, $\uparrow$ indicates that higher is better, and $\downarrow$ that lower is better.

## E.2 Average performance results

### E.2.1 State Dataset

Table 5: Average performance for the ExtremalMask (Enguehard, 2023) explanation method (preservation game objective) across iteration steps for the state dataset when initialized with Start-Grad, all-ones and uniformly. The reported numbers are the mean and standard deviation across five folds. For each metric, ↑ indicates that higher is better, and ↓ that lower is better. Baseline refers to the initialization scheme originally used for the respective mask explanation method. Outcomes that are one standard deviation away from the second-best one are highlighted in bold.

| Metric | Initialization | Iteration steps | | | |
| --- | --- | --- | --- | --- | --- |
| | | 50 | 100 | 300 | 500 |
| AUP ↑ | Uniform (baseline) | $0.719 \pm 0.016$ | $0.800 \pm 0.013$ | $0.855 \pm 0.008$ | $0.856 \pm 0.008$ |
| | All-ones | $\mathbf{0.879 \pm 0.006}$ | $0.847 \pm 0.008$ | $0.860 \pm 0.007$ | $0.860 \pm 0.007$ |
| | StartGrad (ours) | $0.840 \pm 0.007$ | $\mathbf{0.862 \pm 0.007}$ | $0.861 \pm 0.008$ | $0.861 \pm 0.007$ |
| AUR ↑ | Uniform (baseline) | $0.694 \pm 0.011$ | $0.728 \pm 0.009$ | $0.741 \pm 0.010$ | $0.742 \pm 0.010$ |
| | All-ones | $0.691 \pm 0.013$ | $\mathbf{0.755 \pm 0.010}$ | $0.745 \pm 0.010$ | $0.745 \pm 0.010$ |
| | StartGrad (ours) | $\mathbf{0.794 \pm 0.007}$ | $0.749 \pm 0.010$ | $0.746 \pm 0.010$ | $0.746 \pm 0.010$ |
| I $[10^5]$ ↑ | Uniform (baseline) | $1.034 \pm 0.052$ | $2.923 \pm 0.122$ | $3.485 \pm 0.059$ | $3.505 \pm 0.061$ |
| | All-ones | $2.599 \pm 0.125$ | $3.398 \pm 0.074$ | $3.532 \pm 0.064$ | $3.532 \pm 0.066$ |
| | StartGrad (ours) | $\mathbf{3.538 \pm 0.074}$ | $\mathbf{3.538 \pm 0.060}$ | $3.545 \pm 0.059$ | $3.544 \pm 0.059$ |
| E $[10^4]$ ↓ | Uniform (baseline) | $1.914 \pm 0.137$ | $0.921 \pm 0.060$ | $0.735 \pm 0.027$ | $0.729 \pm 0.027$ |
| | All-ones | $1.473 \pm 0.075$ | $1.047 \pm 0.037$ | $0.729 \pm 0.027$ | $0.726 \pm 0.026$ |
| | StartGrad (ours) | $\mathbf{1.232 \pm 0.055}$ | $\mathbf{0.753 \pm 0.033}$ | $0.720 \pm 0.027$ | $0.720 \pm 0.027$ |

Table 6: Average performance for the ExtremalMask (Enguehard, 2023) explanation method (deletion game objective) across iteration steps for the state dataset when initialized with StartGrad, all-ones and uniformly. The reported numbers are the mean and standard deviation across five folds. For each metric, ↑ indicates that higher is better, and ↓ that lower is better. Baseline refers to the initialization scheme originally used for the respective mask explanation method. Outcomes that are one standard deviation away from the second-best one are highlighted in bold.

| Metric | Initialization | Iteration steps | | | |
| --- | --- | --- | --- | --- | --- |
| | | 50 | 100 | 300 | 500 |
| AUP ↑ | Uniform (baseline) | $0.630 \pm 0.029$ | $0.712 \pm 0.024$ | $0.821 \pm 0.028$ | $0.870 \pm 0.026$ |
| | All-ones | $0.824 \pm 0.017$ | $\mathbf{0.895 \pm 0.017}$ | $\mathbf{0.909 \pm 0.022}$ | $\mathbf{0.916 \pm 0.022}$ |
| | StartGrad (ours) | $\mathbf{0.843 \pm 0.018}$ | $\mathbf{0.895 \pm 0.024}$ | $\mathbf{0.917 \pm 0.025}$ | $\mathbf{0.925 \pm 0.025}$ |
| AUR ↑ | Uniform (baseline) | $0.673 \pm 0.012$ | $0.717 \pm 0.011$ | $0.754 \pm 0.011$ | $0.752 \pm 0.009$ |
| | All-ones | $0.481 \pm 0.020$ | $0.600 \pm 0.011$ | $0.747 \pm 0.011$ | $0.752 \pm 0.010$ |
| | StartGrad (ours) | $\mathbf{0.815 \pm 0.004}$ | $\mathbf{0.763 \pm 0.007}$ | $0.754 \pm 0.009$ | $0.754 \pm 0.009$ |
| I $[10^5]$ ↑ | Uniform (baseline) | $1.072 \pm 0.049$ | $3.948 \pm 0.064$ | $4.326 \pm 0.032$ | $4.377 \pm 0.039$ |
| | All-ones | $0.739 \pm 0.052$ | $0.891 \pm 0.030$ | $4.340 \pm 0.042$ | $4.395 \pm 0.042$ |
| | StartGrad (ours) | $\mathbf{4.415 \pm 0.037}$ | $\mathbf{4.416 \pm 0.042}$ | $\mathbf{4.416 \pm 0.041}$ | $4.413 \pm 0.040$ |
| E $[10^4]$ ↓ | Uniform (baseline) | $1.536 \pm 0.053$ | $0.479 \pm 0.019$ | $0.348 \pm 0.048$ | $0.283 \pm 0.037$ |
| | All-ones | $1.632 \pm 0.119$ | $1.875 \pm 0.135$ | $\mathbf{0.287 \pm 0.036}$ | $0.264 \pm 0.341$ |
| | StartGrad (ours) | $\mathbf{0.932 \pm 0.016}$ | $\mathbf{0.350 \pm 0.015}$ | $\mathbf{0.261 \pm 0.031}$ | $0.260 \pm 0.032$ |

### E.2.2 SWITCH-FEATURE DATASET

Table 7: Average performance for the ExtremalMask (Enguehard, 2023) explanation method (deletion game objective) across iteration steps for the switch-feature dataset when initialized with Start-Grad, all-ones and uniformly. The reported numbers are the mean and standard deviation across five folds. For each metric, ↑ indicates that higher is better, and ↓ that lower is better. Baseline refers to the initialization scheme originally used for the respective mask explanation method. Outcomes that are one standard deviation away from the second-best one are highlighted in bold.

| Metric | Initialization | Iteration steps | | | |
|---|---|---|---|---|---|
| | | 50 | 100 | 300 | 500 |
| AUP ↑ | Uniform (baseline) | $0.825 \pm 0.025$ | $0.875 \pm 0.041$ | $\mathbf{0.913 \pm 0.071}$ | $\mathbf{0.932 \pm 0.078}$ |
| | All-ones | $0.762 \pm 0.016$ | $0.705 \pm 0.047$ | $0.764 \pm 0.013$ | $0.755 \pm 0.015$ |
| | StartGrad (ours) | $\mathbf{0.869 \pm 0.007}$ | $\mathbf{0.908 \pm 0.019}$ | $\mathbf{0.936 \pm 0.048}$ | $\mathbf{0.948 \pm 0.048}$ |
| AUR ↑ | Uniform (baseline) | $0.629 \pm 0.017$ | $\mathbf{0.622 \pm 0.017}$ | $\mathbf{0.631 \pm 0.027}$ | $\mathbf{0.629 \pm 0.029}$ |
| | All-ones | $\mathbf{0.695 \pm 0.038}$ | $0.545 \pm 0.026$ | $0.563 \pm 0.012$ | $0.547 \pm 0.013$ |
| | StartGrad (ours) | $\mathbf{0.693 \pm 0.011}$ | $\mathbf{0.634 \pm 0.013}$ | $\mathbf{0.629 \pm 0.024}$ | $\mathbf{0.628 \pm 0.029}$ |
| I $[10^5]$ ↑ | Uniform (baseline) | $0.781 \pm 0.052$ | $1.532 \pm 0.112$ | $\mathbf{2.232 \pm 0.217}$ | $\mathbf{2.248 \pm 0.241}$ |
| | All-ones | $0.923 \pm 0.128$ | $0.559 \pm 0.053$ | $1.410 \pm 0.037$ | $1.548 \pm 0.058$ |
| | StartGrad (ours) | $\mathbf{1.734 \pm 0.103}$ | $\mathbf{2.046 \pm 0.072}$ | $\mathbf{2.211 \pm 0.155}$ | $\mathbf{2.220 \pm 0.168}$ |
| E $[10^4]$ ↓ | Uniform (baseline) | $2.639 \pm 0.163$ | $1.842 \pm 0.077$ | $\mathbf{1.410 \pm 0.082}$ | $\mathbf{1.378 \pm 0.074}$ |
| | All-ones | $\mathbf{2.411 \pm 0.210}$ | $3.006 \pm 0.072$ | $1.985 \pm 0.053$ | $1.717 \pm 0.051$ |
| | StartGrad (ours) | $\mathbf{2.399 \pm 0.044}$ | $\mathbf{1.680 \pm 0.065}$ | $\mathbf{1.398 \pm 0.073}$ | $\mathbf{1.366 \pm 0.060}$ |

## F    ADDITIONAL ABLATION STUDIES

### F.1    VISION EXPERIMENTS WITH VGG16 AS BASELINE MODEL

#### F.1.1    FIGURES FOR THE VGG16 EXPERIMENT

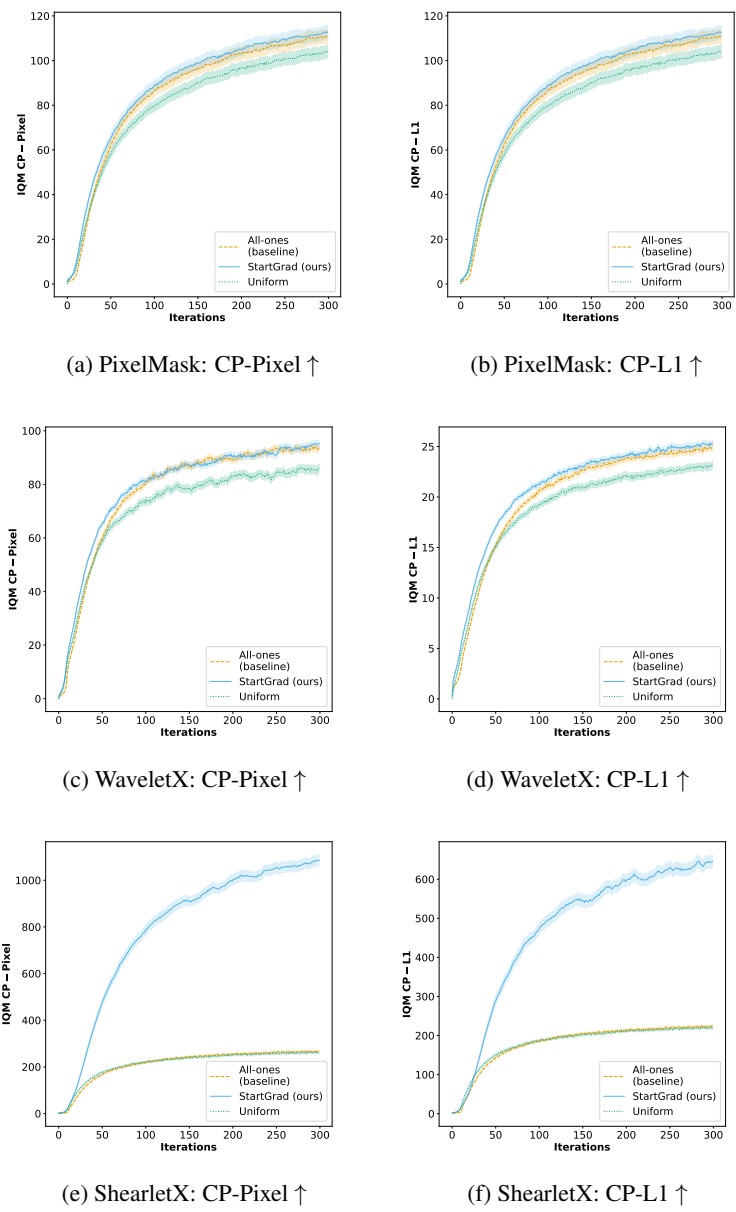

Figure 10: Comparison of StartGrad initialization (ours) with standard baseline initialization schemes for the PixelMask (Fong & Vedaldi, 2017) (first row), WaveletX (Kolek et al., 2023) (second row) and ShearletX (Kolek et al., 2023) (third row) explanation models. Baseline refers to the originally used initialization scheme for the respective mask explanation method. The solid line represents the average of the interquartile mean (IQM) performance across 500 randomly selected validation ImageNet (Deng et al., 2009) samples, while the shaded area denotes the standard errors respectively. For each metric, ↑ indicates that higher is better, and ↓ that lower is better. We employ a pretrained VGG16 (Simonyan & Zisserman, 2014) model as a classifier.

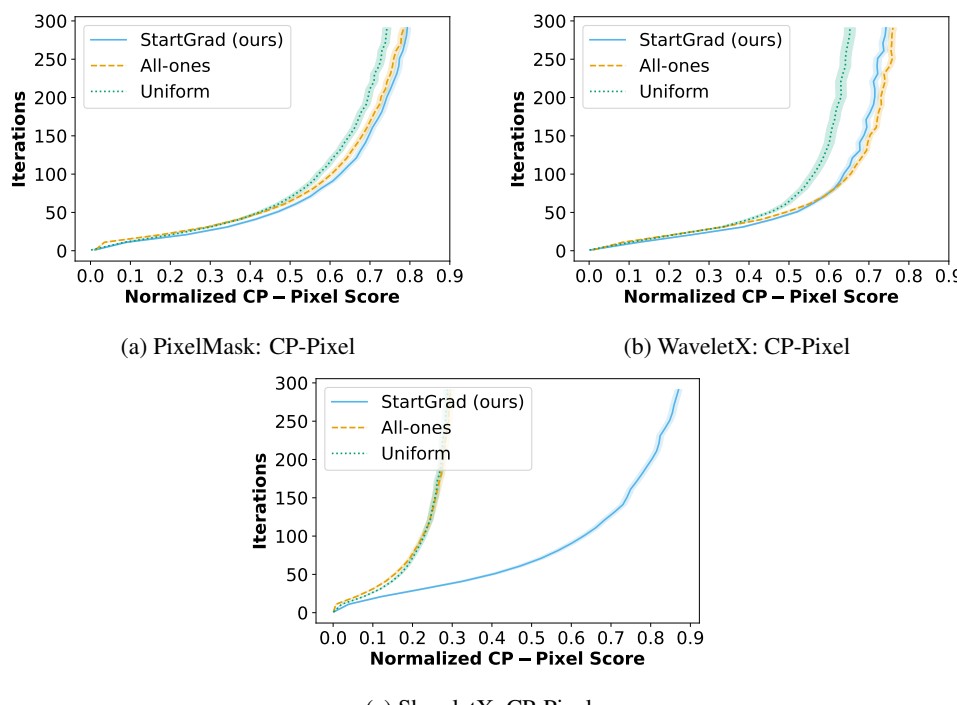

(a) PixelMask: CP-Pixel

(b) WaveletX: CP-Pixel

(c) ShearletX: CP-Pixel

Figure 11: Normalized CP-Pixel score vs. iteration steps for the PixelMask (Fong & Vedaldi, 2017) (a), WaveletX (Kolek et al., 2023) (b), and ShearletX (Kolek et al., 2023) (c) models, comparing three initialization methods: StartGrad (ours) (solid line), All-ones (dashed line), and Uniform (dotted line). The curves represent the average number of iteration steps needed to reach a normalized target CP-L1 score. This target score is defined as the highest CP-L1 value achieved across all three initialization methods for each model, then normalized by the overall maximum score observed. The shaded regions represent the standard error across 500 randomly selected ImageNet validation samples (Deng et al., 2009), with a pretrained VGG16 (Simonyan & Zisserman, 2014) classifier. StartGrad allows all three models to reach target scores in fewer iterations than the uniform initialization. Compared to all-ones initialization, it accelerates PixelMask (Fong & Vedaldi, 2017) and ShearletX (Kolek et al., 2023) while performing slightly worse for WaveletX (Kolek et al., 2023) for medium to high scores.

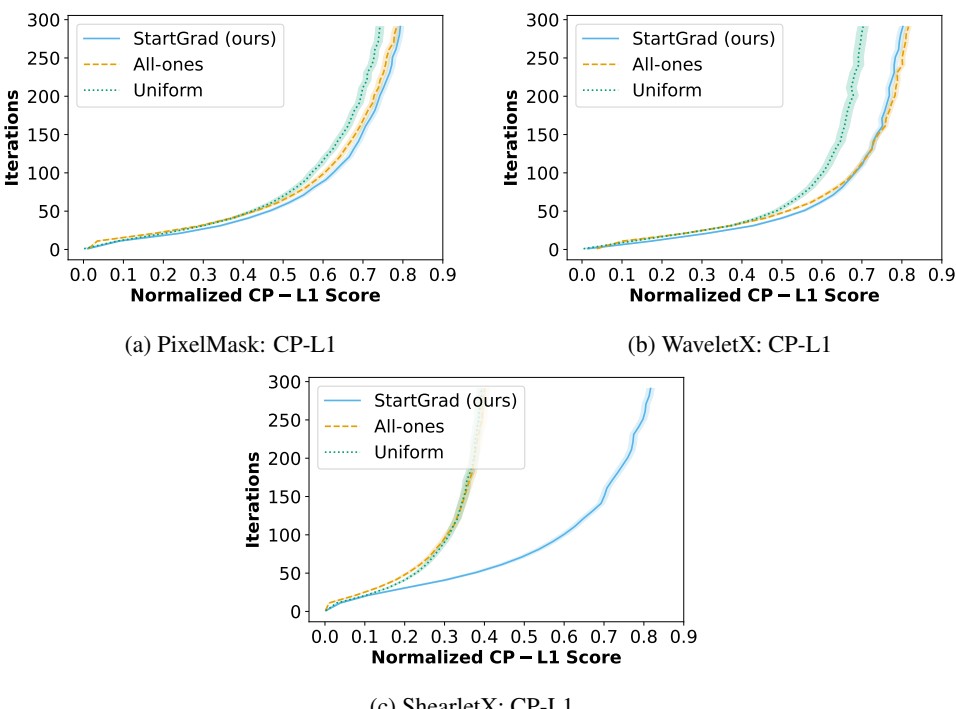

(a) PixelMask: CP-L1

(b) WaveletX: CP-L1

(c) ShearletX: CP-L1

Figure 12: Normalized CP-L1 score vs. iteration steps for the PixelMask (Fong & Vedaldi, 2017) (a), WaveletX (Kolek et al., 2023) (b), and ShearletX (Kolek et al., 2023) (c) models, comparing three initialization methods: StartGrad (ours) (solid line), All-ones (dashed line), and Uniform (dotted line). The curves represent the average number of iteration steps needed to reach a normalized target CP-L1 score. This target score is defined as the highest CP-L1 value achieved across all three initialization methods for each model, then normalized by the overall maximum score observed. The shaded regions represent the standard error across 500 randomly selected ImageNet validation samples (Deng et al., 2009), with a pretrained VGG16 (Simonyan & Zisserman, 2014) classifier. StartGrad allows all three models to reach target scores in fewer iterations than the uniform initialization. Compared to all-ones initialization, it accelerates PixelMask (Fong & Vedaldi, 2017) and ShearletX (Kolek et al., 2023) while performing slightly worse for WaveletX (Kolek et al., 2023) for medium to high scores.

## F.2 COMPARISON PERFORMANCE DIFFERENCES STARTGRAD VS. BASELINES

Table 8: Median pairwise performance difference between StartGrad and baseline initialization methods across different iteration steps for 500 randomly selected validation ImageNet (Deng et al., 2009) samples using a VGG16 (Simonyan & Zisserman, 2014) classifier. Baseline refers to the originally used initialization scheme for the respective mask explanation method. Statistical significance is denoted by **, **, and * at the 1%, 5%, and 10% levels, respectively, based on an one-sided Wilcoxon signed-rank test with Bonferroni correction at the method and metric level. Since Pixel-Mask (Fong & Vedaldi, 2017) does only apply a mask to the pixel space, the CP-Pixel and CP-L1 scores are identical.

| | | $\triangle$ CP-Pixel $\uparrow$ | | | $\triangle$ CP-L1 $\uparrow$ | | |
| | | Iteration steps | | | Iteration steps | | |
| Method | Baseline initialization | 50 | 100 | 300 | 50 | 100 | 300 |
|---|---|---|---|---|---|---|---|
| PixelMask | All-ones (baseline) | $2.29^{***}$ | $2.04^{***}$ | $1.65^{**}$ | $2.29^{***}$ | $2.04^{***}$ | $1.65^{**}$ |
| | Uniform | $2.60^{***}$ | $4.06^{***}$ | $3.33^{***}$ | $2.60^{***}$ | $4.06^{***}$ | $3.33^{***}$ |
| WaveletX | All-ones (baseline) | $3.61^{***}$ | $0.88$ | $-0.11$ | $1.11^{***}$ | $0.52^{**}$ | $0.10$ |
| | Uniform | $2.56^{***}$ | $2.04^{***}$ | $2.82^{***}$ | $0.85^{***}$ | $0.67^{***}$ | $0.85^{***}$ |
| ShearletX | All-ones (baseline) | $180.31^{***}$ | $454.65^{***}$ | $767.72^{***}$ | $55.22^{***}$ | $192.11^{***}$ | $351.26^{***}$ |
| | Uniform | $173.49^{***}$ | $462.95^{***}$ | $770.24^{***}$ | $52.93^{***}$ | $201.90^{***}$ | $370.77^{***}$ |

Table 9: Average iteration steps required for the ShearletX (Kolek et al., 2023) method initialized with StartGrad to match the corresponding performance of the same method initialized with all-ones or uniform. The table reports the time (in iteration steps) needed to match the interquartile mean (IQM) and median performance of the CP-Pixel and CP-L1 score obtained. Additionally, speedup is calculated as the ratio between the baseline iteration steps and the time taken under StartGrad initialization. All experiments use 500 random ImageNet (Deng et al., 2009) samples evaluated on a pretrained VGG16 (Simonyan & Zisserman, 2014) classifier. The target metrics are obtained for running the ShearletX method across 300 iteration steps. To obtain the average and standard error, we use bootstrapping across 250 resampled sets. For each metric, $\uparrow$ indicates that higher is better, and $\downarrow$ that lower is better. Values that are 2 standard errors away from the baseline values (300 for iteration, 1 for speedup) are highlighted in bold.

| Reference | Target metric (at 300 iterations) | CP-Pixel | | CP-L1 | |
| | | Iterations $\downarrow$ | Speedup $\uparrow$ | Iterations $\downarrow$ | Speedup $\uparrow$ |
|---|---|---|---|---|---|
| All-ones (baseline) | IQM | $\mathbf{33.10 \pm 1.55}$ | $\mathbf{9.05 \pm 0.42}$ | $\mathbf{41.43 \pm 2.30}$ | $\mathbf{7.24 \pm 0.40}$ |
| | Median | $\mathbf{36.75 \pm 2.57}$ | $\mathbf{8.18 \pm 0.58}$ | $\mathbf{48.82 \pm 4.08}$ | $\mathbf{6.17 \pm 0.52}$ |
| Uniform | IQM | $\mathbf{32.58 \pm 1.50}$ | $\mathbf{9.20 \pm 0.42}$ | $\mathbf{40.79 \pm 2.23}$ | $\mathbf{7.35 \pm 0.40}$ |
| | Median | $\mathbf{36.23 \pm 2.49}$ | $\mathbf{8.29 \pm 0.58}$ | $\mathbf{48.20 \pm 4.02}$ | $\mathbf{6.25 \pm 0.53}$ |

## F.3 VISION EXPERIMENTS WITH SWIN TRANSFORMER AS BASELINE MODEL

### F.3.1 FIGURES FOR THE SWIN TRANSFORMER EXPERIMENT

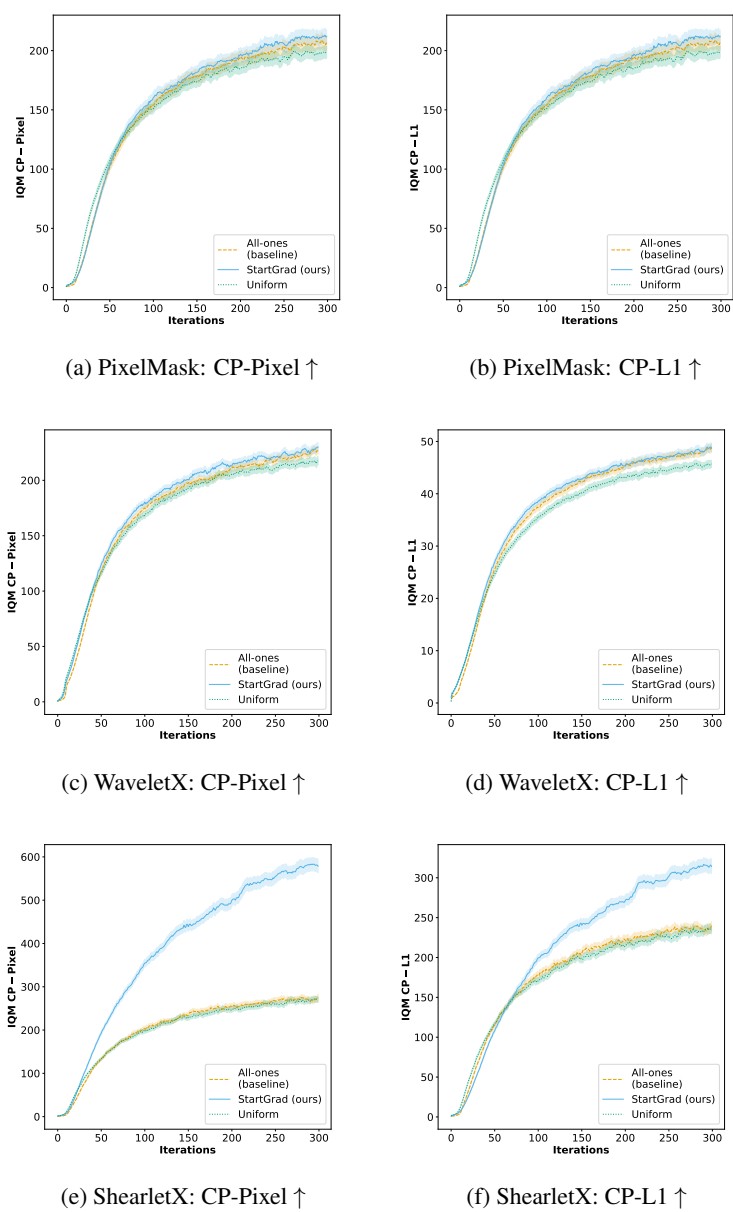

(a) PixelMask: CP-Pixel ↑

(b) PixelMask: CP-L1 ↑

(c) WaveletX: CP-Pixel ↑

(d) WaveletX: CP-L1 ↑

(e) ShearletX: CP-Pixel ↑

(f) ShearletX: CP-L1 ↑

Figure 13: Comparison of StartGrad initialization (ours) with standard baseline initialization schemes for the PixelMask (Fong & Vedaldi, 2017) (first row), WaveletX (Kolek et al., 2023) (second row) and ShearletX (Kolek et al., 2023) (third row) explanation models. Baseline refers to the originally used initialization scheme for the respective mask explanation method. The solid line represents the average of the interquartile mean (IQM) performance across 250 randomly selected validation ImageNet (Deng et al., 2009) samples, while the shaded area denotes the standard errors respectively. For each metric, ↑ indicates that higher is better, and ↓ that lower is better. We employ a pretrained Swin Transformer (Liu et al., 2022) model as a classifier.

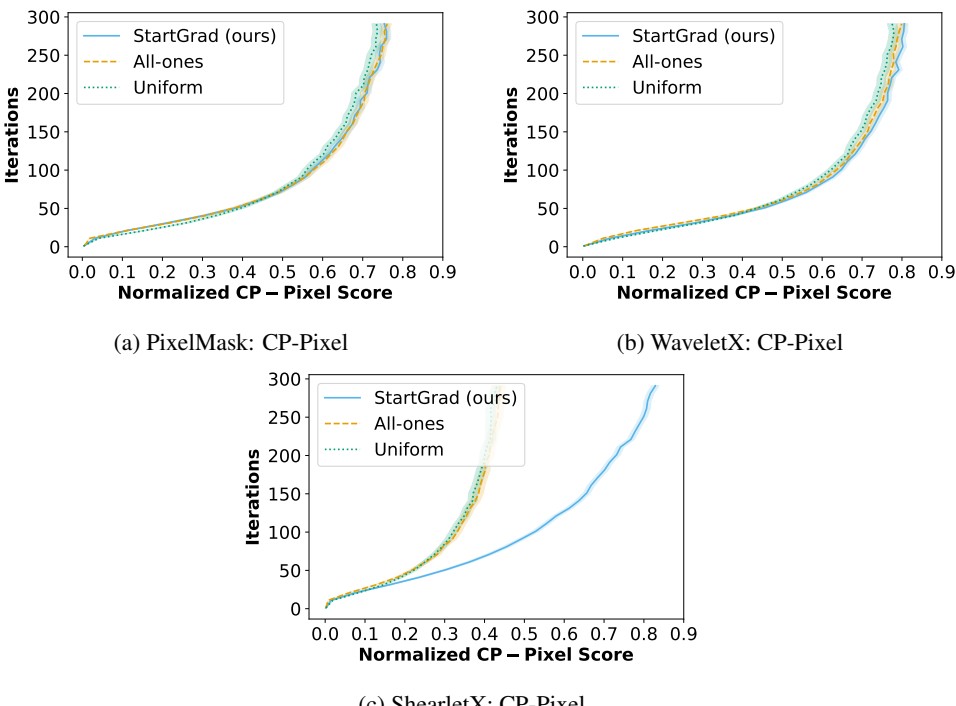

(a) PixelMask: CP-Pixel

(b) WaveletX: CP-Pixel

(c) ShearletX: CP-Pixel

Figure 14: Normalized CP-Pixel score vs. iteration steps for the PixelMask (Fong & Vedaldi, 2017) (a), WaveletX (Kolek et al., 2023) (b), and ShearletX (Kolek et al., 2023) (c) models, comparing three initialization methods: StartGrad (ours) (solid line), All-ones (dashed line), and Uniform (dotted line). The curves represent the average number of iteration steps needed to reach a normalized target CP-Pixel score. This target score is defined as the highest CP-Pixel value achieved across all three initialization methods for each model, then normalized by the overall maximum score observed. The shaded regions represent the standard error across 250 randomly selected ImageNet validation samples (Deng et al., 2009), with a pretrained Swin Transformer (Liu et al., 2022) classifier. StartGrad enables all three models to achieve target scores with fewer iterations compared to uniform initialization scheme. Compared to the all-ones initialization, StartGrad enables to achieve target scores with fewer iterations for the WaveletX (Kolek et al., 2023) and ShearletX (Kolek et al., 2023) model, and performs on par for the PixelMask (Fong & Vedaldi, 2017) model.

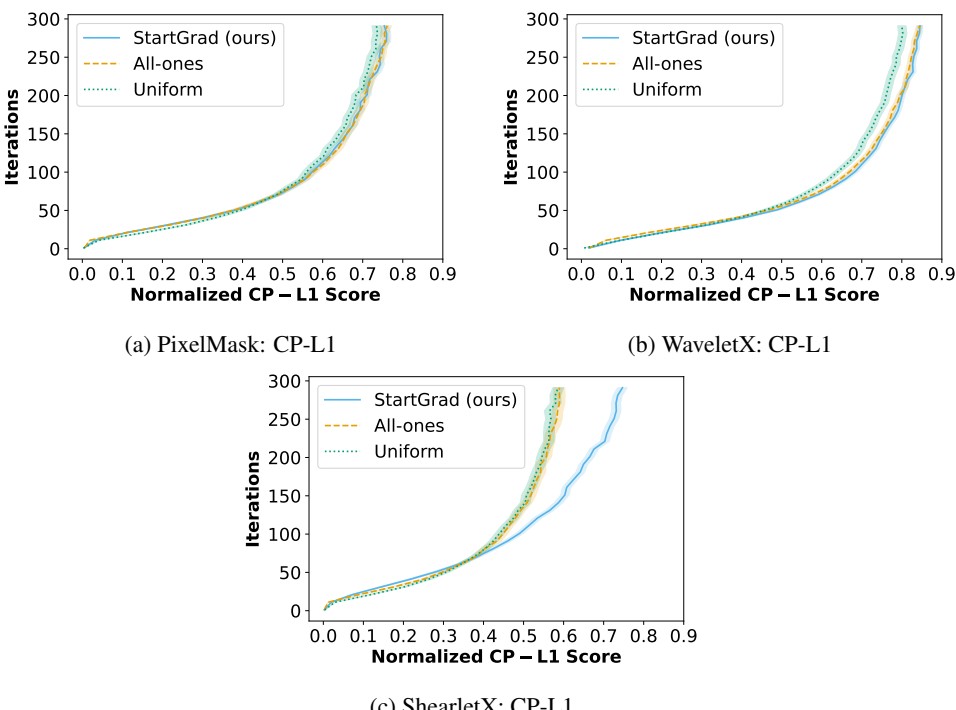

(a) PixelMask: CP-L1

(b) WaveletX: CP-L1

(c) ShearletX: CP-L1

Figure 15: Normalized CP-L1 score vs. iteration steps for the PixelMask (Fong & Vedaldi, 2017) (a), WaveletX (Kolek et al., 2023) (b), and ShearletX (Kolek et al., 2023) (c) models, comparing three initialization methods: StartGrad (ours) (solid line), All-ones (dashed line), and Uniform (dotted line). The curves represent the average number of iteration steps needed to reach a normalized target CP-L1 score. This target score is defined as the highest CP-L1 value achieved across all three initialization methods for each model, then normalized by the overall maximum score observed. The shaded regions represent the standard error across 250 randomly selected ImageNet validation samples (Deng et al., 2009), with a pretrained Swin Transformer (Liu et al., 2022) classifier. Start-Grad enables all three models to achieve target scores with fewer iterations compared to uniform initialization scheme. Compared to the all-ones initialization, StartGrad enables to achieve target scores with fewer iterations for the WaveletX (Kolek et al., 2023) and ShearletX (Kolek et al., 2023) model, and performs on par for the PixelMask (Fong & Vedaldi, 2017) model.

## F.4 VISION EXPERIMENTS WITH VISION TRANSFORMER AS BASELINE MODEL

### F.4.1 FIGURES FOR THE VISION TRANSFORMER EXPERIMENT

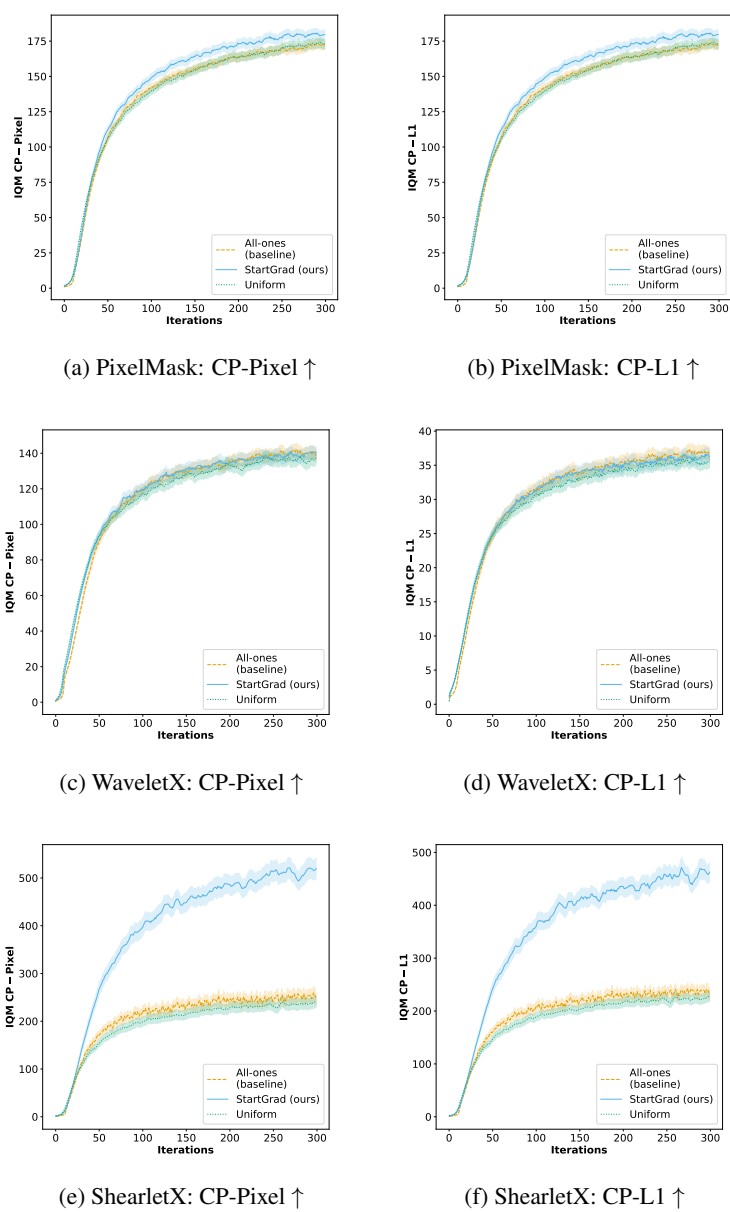

(a) PixelMask: CP-Pixel ↑

(b) PixelMask: CP-L1 ↑

(c) WaveletX: CP-Pixel ↑

(d) WaveletX: CP-L1 ↑

(e) ShearletX: CP-Pixel ↑

(f) ShearletX: CP-L1 ↑

Figure 16: Comparison of StartGrad initialization (ours) with standard baseline initialization schemes for the PixelMask (Fong & Vedaldi, 2017) (first row), WaveletX (Kolek et al., 2023) (second row) and ShearletX (Kolek et al., 2023) (third row) explanation models. Baseline refers to the originally used initialization scheme for the respective mask explanation method. The solid line represents the average of the interquartile mean (IQM) performance across 100 randomly selected validation ImageNet (Deng et al., 2009) samples, while the shaded area denotes the standard errors respectively. For each metric, ↑ indicates that higher is better, and ↓ that lower is better. We employ a pretrained Vision Transformer model (Dosovitskiy et al., 2021) model as a classifier.

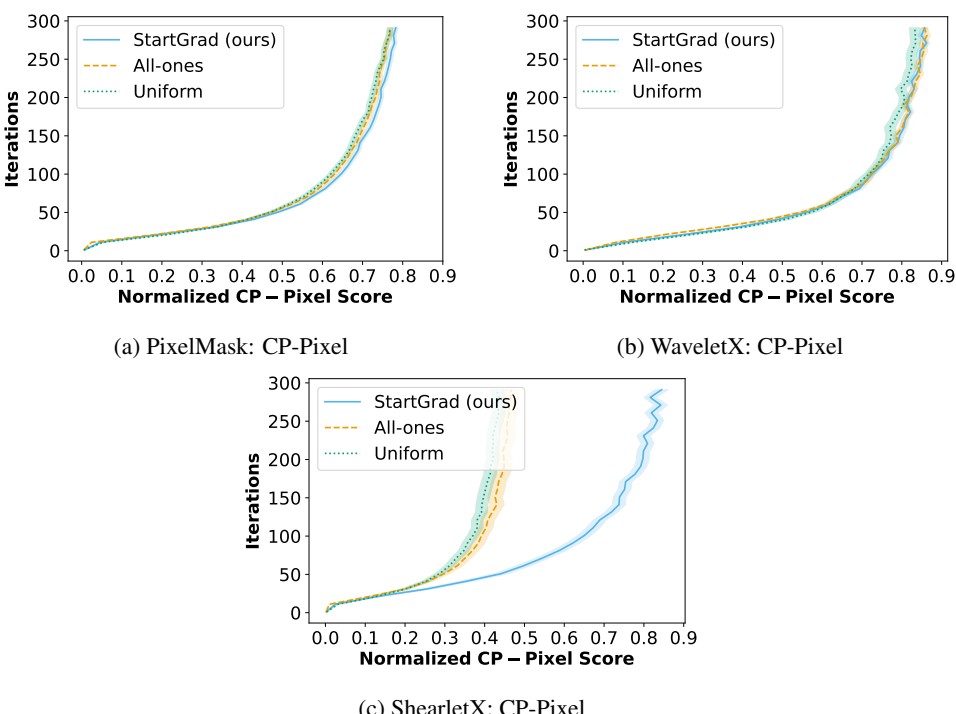

(a) PixelMask: CP-Pixel

(b) WaveletX: CP-Pixel

(c) ShearletX: CP-Pixel

Figure 17: Normalized CP-Pixel score vs. iteration steps for the PixelMask (Fong & Vedaldi, 2017) (a), WaveletX (Kolek et al., 2023) (b), and ShearletX (Kolek et al., 2023) (c) models, comparing three initialization methods: StartGrad (ours) (solid line), All-ones (dashed line), and Uniform (dotted line). The curves represent the average number of iteration steps needed to reach a normalized target CP-Pixel score. This target score is defined as the highest CP-Pixel value achieved across all three initialization methods for each model, then normalized by the overall maximum score observed. The shaded regions represent the standard error across 100 randomly selected ImageNet validation samples (Deng et al., 2009), with a Vision Transformer model (Dosovitskiy et al., 2021) classifier. StartGrad enables all three models to achieve target scores with fewer iterations compared to uniform initialization scheme. Compared to the all-ones initialization, StartGrad enables to achieve target scores with fewer iterations for the PixelMask (Fong & Vedaldi, 2017) and ShearletX (Kolek et al., 2023) model, and performs on par for the WaveletX (Kolek et al., 2023) model.

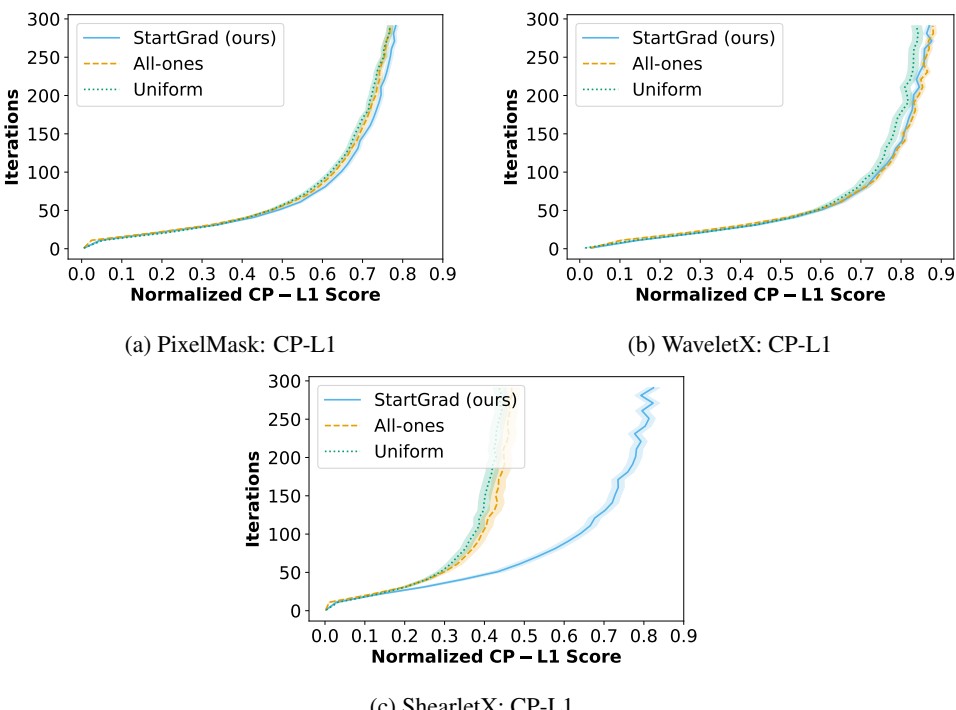

(a) PixelMask: CP-L1

(b) WaveletX: CP-L1

(c) ShearletX: CP-L1

Figure 18: Normalized CP-L1 score vs. iteration steps for the PixelMask (Fong & Vedaldi, 2017) (a), WaveletX (Kolek et al., 2023) (b), and ShearletX (Kolek et al., 2023) (c) models, comparing three initialization methods: StartGrad (ours) (solid line), All-ones (dashed line), and Uniform (dotted line). The curves represent the average number of iteration steps needed to reach a normalized target CP-L1 score. This target score is defined as the highest CP-L1 value achieved across all three initialization methods for each model, then normalized by the overall maximum score observed. The shaded regions represent the standard error across 100 random ImageNet validation samples (Deng et al., 2009), with a Vision Transformer model (Dosovitskiy et al., 2021) classifier. StartGrad allows all three models to reach target scores in fewer iterations than the uniform initialization. Compared to all-ones initialization, it accelerates PixelMask (Fong & Vedaldi, 2017) and ShearletX (Kolek et al., 2023) while performing on par with WaveletX (Kolek et al., 2023).

F.5 EFFECT OF NOISY GRADIENTS ON STARTGRAD

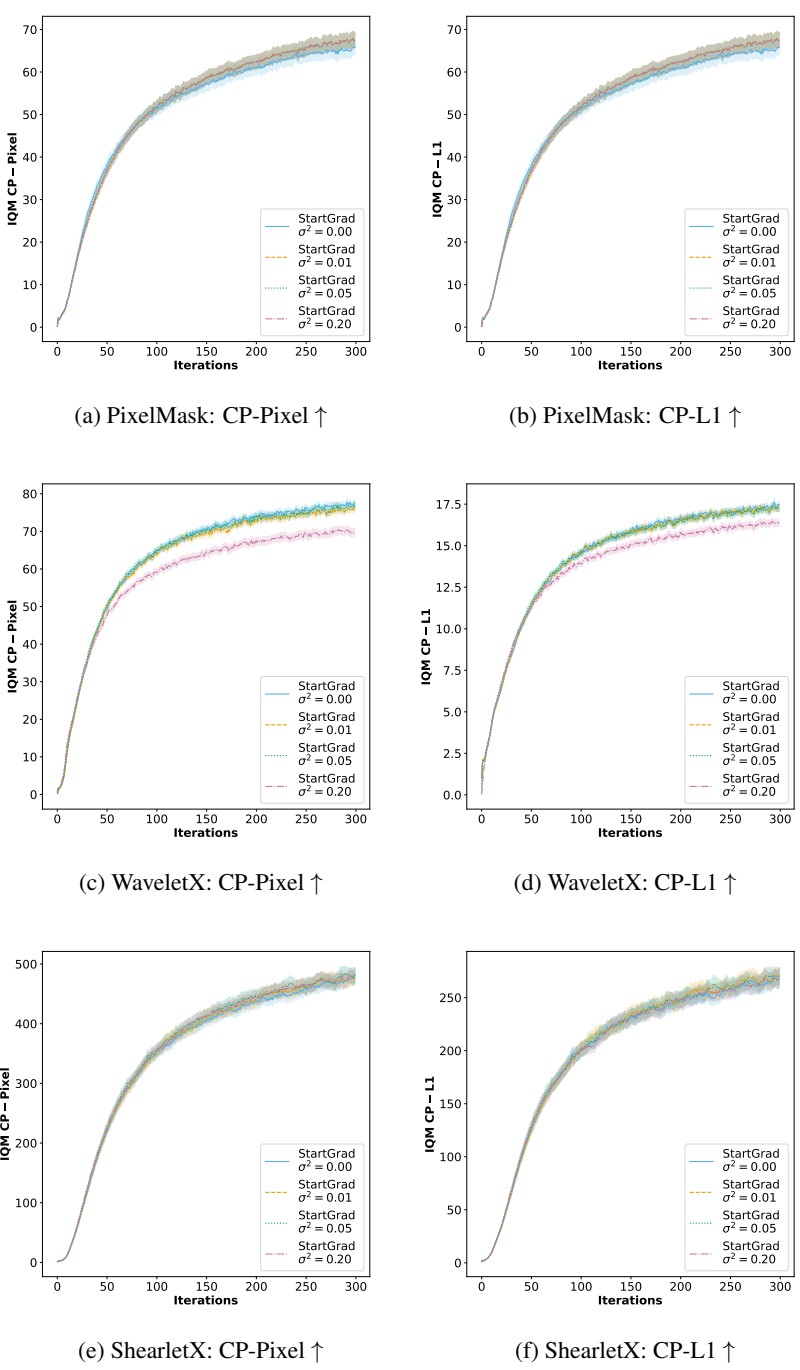

(a) PixelMask: CP-Pixel ↑                    (b) PixelMask: CP-L1 ↑

(c) WaveletX: CP-Pixel ↑                      (d) WaveletX: CP-L1 ↑

(e) ShearletX: CP-Pixel ↑                     (f) ShearletX: CP-L1 ↑

Figure 19: Investigation of StartGrad's robustness to noisy gradients by progressively multiplying the gradient with Gaussian noise of increasing standard deviation for the PixelMask (Fong & Vedaldi, 2017) (first row), WaveletX (Kolek et al., 2023) (second row) and ShearletX (Kolek et al., 2023) (third row) explanation models. The solid line represents the average of the interquartile mean (IQM) performance across 500 randomly selected validation ImageNet (Deng et al., 2009) samples, while the shaded area denotes the standard errors respectively. ↑ indicates that higher is better, and ↓ that lower is better. We use a pretrained ResNet18 (He et al., 2016) classifier.

### F.6 IMPACT OF GRADIENT-BASED INITIALIZATION ON DISTORTION AND SPARSITY DYNAMICS FOR PIXELMASK

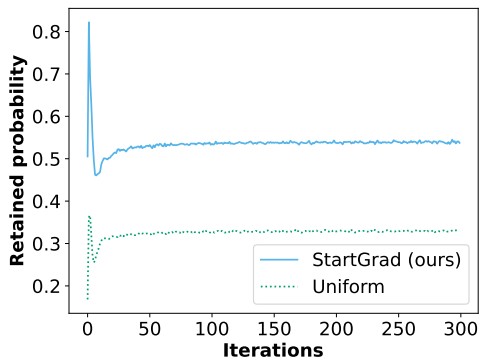

(a) PixelMask: Retained probability ↑

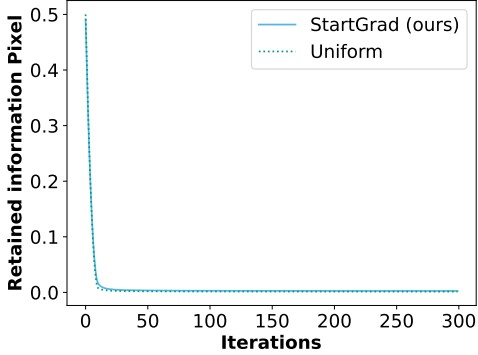

(b) PixelMask: Retained information Pixel ↓

Figure 20: Overview of optimization dynamics of retained probability (a), measured as the ratio between the masked input prediction and the original prediction, and retained information (in pixel space) (b), for the StartGrad (ours) and uniform initialization method, using the PixelMask (Fong & Vedaldi, 2017) model. We employ a pretrained ResNet18 (He et al., 2016) model as a classifier. The solid lines represent the average performance across 500 randomly selected ImageNet validation samples (Deng et al., 2009). For each metric, ↑ indicates that higher is better, and ↓ that lower is better. Results indicate that the gradient-information that StartGrad uses to initialize the mask guides the mask-based explanation method effectively to achieve minimal distortion (as indicated by high retained probability) early in the optimization process, consistent with theoretical predictions. However, this comes at the cost of reduced sparsity, as evidenced by higher retained information (b). In contrast, uniform initialization exhibits better sparsity (lower retained information) but struggles to retain critical information as indicated by comparable low retained probability, highlighting a trade-off between higher retained probability and increased sparsity. The optimization dynamics further reveal that the early iteration steps are especially crucial for retaining probability, as the mask-based explanation methods quickly converge in terms of retained probability for both initialization schemes. This behavior underscores the importance of leveraging gradient signals for efficient initialization, achieving a favorable starting point for optimization in terms of minimal distortion (high retained probability).

### F.7  IMPACT OF GRADIENT-BASED INITIALIZATION ON DISTORTION AND SPARSITY DYNAMICS FOR WAVELETX

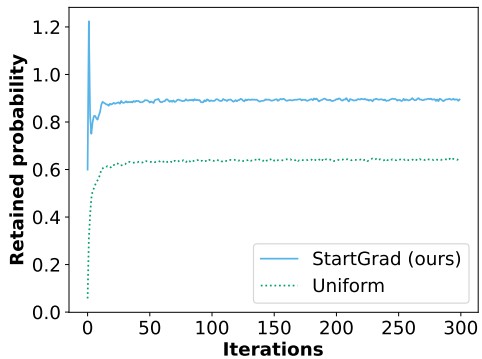

(a) WaveletX: Retained probability ↑

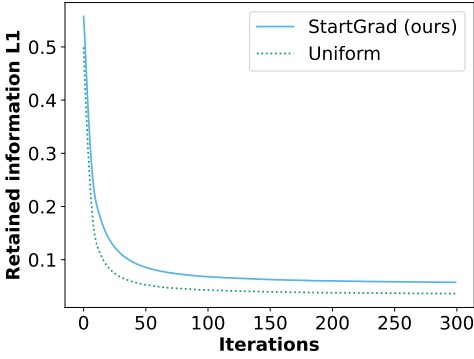

(b) WaveletX: Retained information L1 ↓

Figure 21: Overview of optimization dynamics of retained probability (a), measured as the ratio between the masked input prediction and the original prediction, and retained information ($L_1$) (b), for the StartGrad (ours) and uniform initialization method, using the WaveletX model (Kolek et al., 2023). We employ a pretrained ResNet18 (He et al., 2016) model as a classifier. The solid lines represent the average performance across 500 randomly ImageNet validation (Deng et al., 2009) samples. For each metric, ↑ indicates that higher is better, and ↓ that lower is better. Results indicate that the gradient-information that StartGrad uses to initialize the mask guides the mask-based explanation method effectively to achieve minimal distortion (as indicated by high retained probability) early in the optimization process, consistent with theoretical predictions. However, this comes at the cost of reduced sparsity, as evidenced by higher retained information (b). In contrast, uniform initialization exhibits better sparsity (lower retained information) but struggles to retain critical information as indicated by comparable low retained probability, highlighting a trade-off between higher retained probability and increased sparsity. The optimization dynamics further reveal that the early iteration steps are especially crucial for retaining probability, as the mask-based explanation methods quickly converge in terms of retained probability for both initialization schemes. This behavior underscores the importance of leveraging gradient signals for efficient initialization, achieving a favorable starting point for optimization in terms of minimal distortion (high retained probability).

## F.8    Impact of Gradient-Based Initialization on Distortion and Sparsity Dynamics for ShearletX

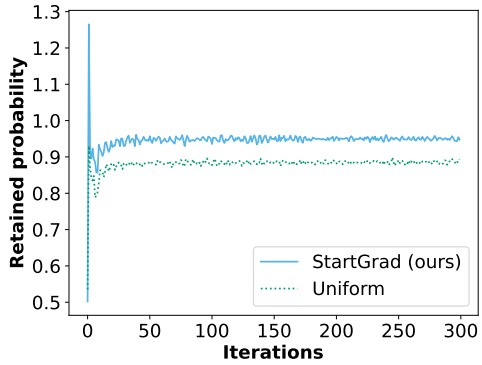

(a) ShearletX: Retained probability ↑

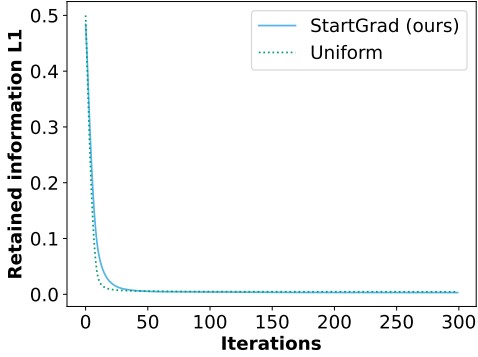

(b) ShearletX: Retained information L1 ↓

Figure 22: Overview of optimization dynamics of retained probability (a), measured as the ratio between the masked input prediction and the original prediction, and retained information ($L_1$) (b), for the StartGrad (ours) and uniform initialization method, using the ShearletX model (Kolek et al., 2023). We employ a pretrained ResNet18 (He et al., 2016) model as a classifier. The solid lines represent the average performance across 500 randomly selected ImageNet (Deng et al., 2009) validation samples. For each metric, ↑ indicates that higher is better, and ↓ that lower is better. Results indicate that the gradient-information that StartGrad uses to initialize the mask guides the mask-based explanation method effectively to achieve minimal distortion (as indicated by high retained probability) early in the optimization process, consistent with theoretical predictions. However, this comes at the cost of reduced sparsity, as evidenced by higher retained information (b). In contrast, uniform initialization exhibits better sparsity (lower retained information) but struggles to retain critical information as indicated by comparable low retained probability, highlighting a trade-off between higher retained probability and increased sparsity. The optimization dynamics further reveal that the early iteration steps are especially crucial for retaining probability, as the mask-based explanation methods quickly converge in terms of retained probability for both initialization schemes. This behavior underscores the importance of leveraging gradient signals for efficient initialization, achieving a favorable starting point for optimization in terms of minimal distortion (high retained probability).

### F.9   EFFECT OF UNINFORMATIVE AND ADVERSARIAL GRADIENTS ON STARTGRAD

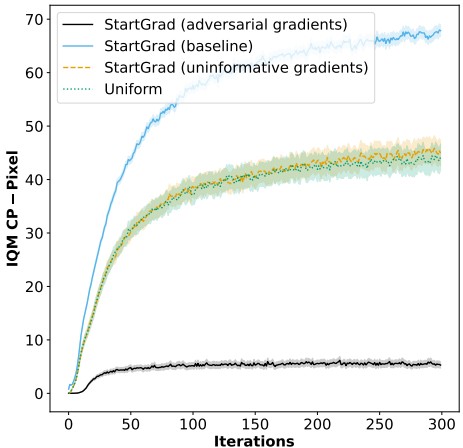

(a) WaveletX: CP-Pixel ↑

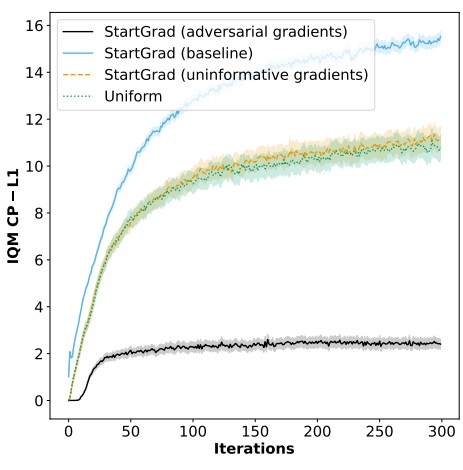

(b) WaveletX: CP-L1 ↑

Figure 23: Investigation of the robustness of StartGrad initialization under scenarios where gradients are either uninformative or adversarial (misleading) for the WaveletX model (Kolek et al., 2023). In the uninformative setting, gradient values were randomly shuffled to destroy their structural signal, effectively rendering them meaningless for guiding mask initialization. In the adversarial setting, the correspondence between gradient values and mask initialization values was reversed, such that features with higher gradient values were assigned lower mask values and vice versa. The solid line represents the average of the interquartile mean (IQM) performance across 250 randomly selected validation ImageNet (Deng et al., 2009) samples, while the shaded area denotes the standard errors respectively. For each metric, ↑ indicates that higher is better, and ↓ that lower is better. We employ a pretrained ResNet18 (He et al., 2016) model as a classifier. Results reveal that under uninformative gradient conditions, StartGrad's performance is comparable to the baseline uniform initialization, indicating that StartGrad relies on meaningful gradient signals to offer an advantage. In the adversarial setting, performance degradation highlights the importance of smart initialization, as the algorithm struggles to recover from a poorly initialized mask during optimization. These findings emphasize the critical role of robust and informed initialization in achieving optimal performance.

## F.10 EFFECT OF USING DIFFERENT GRADIENT METHODS ON STARTGRAD

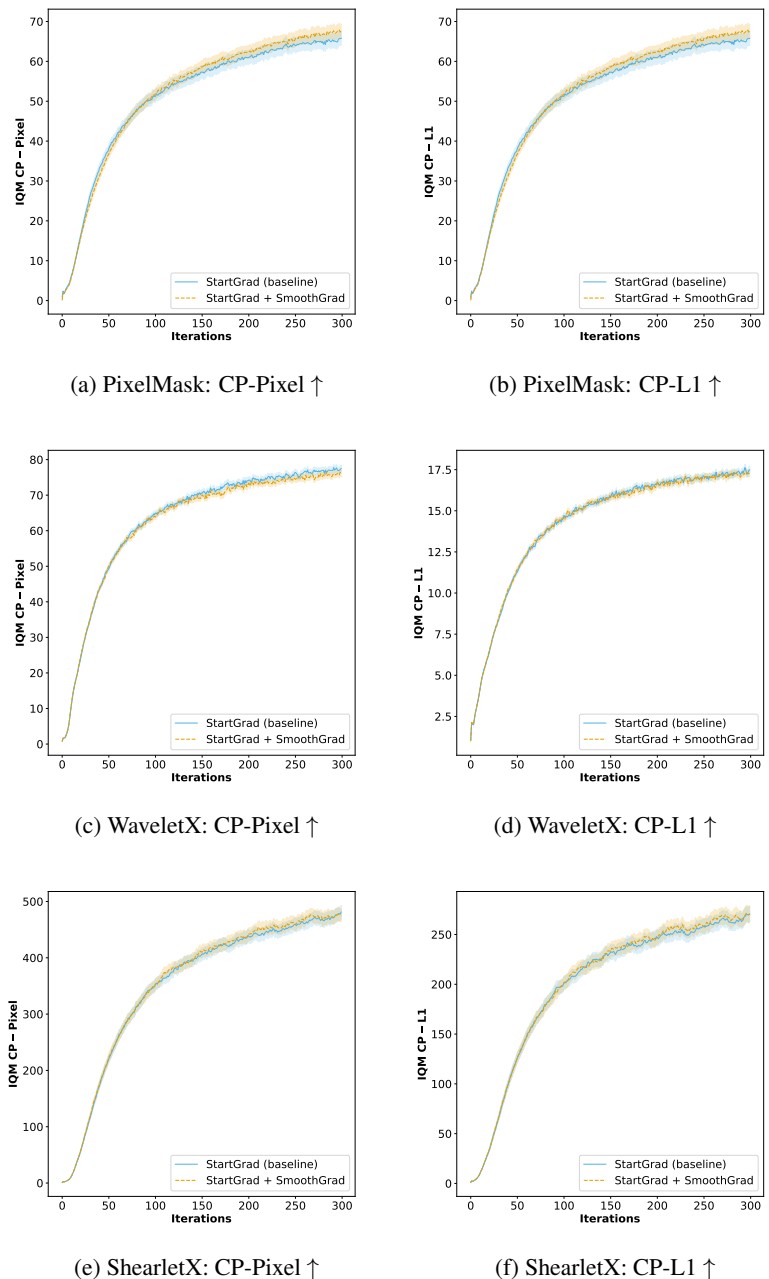

(a) PixelMask: CP-Pixel ↑

(b) PixelMask: CP-L1 ↑

(c) WaveletX: CP-Pixel ↑

(d) WaveletX: CP-L1 ↑

(e) ShearletX: CP-Pixel ↑

(f) ShearletX: CP-L1 ↑

Figure 24: Effect of using a more advanced gradient-based saliency method (here SmoothGrad (Smilkov et al., 2017)) in connection with our proposed StartGrad algorithm for the PixelMask (Fong & Vedaldi, 2017) (first row), WaveletX (Kolek et al., 2023) (second row) and ShearletX (Kolek et al., 2023) (third row) explanation models. The solid line represents the average of the interquartile mean (IQM) performance across 500 randomly selected validation ImageNet (Deng et al., 2009) samples, while the shaded area denotes the standard errors respectively. For each metric, ↑ indicates that higher is better, and ↓ that lower is better. We employ a pretrained ResNet18 (He et al., 2016) model as a classifier.

F.11  EFFECT OF USING A DIFFERENT TRANSFORMATION FUNCTION INSTEAD OF QTF

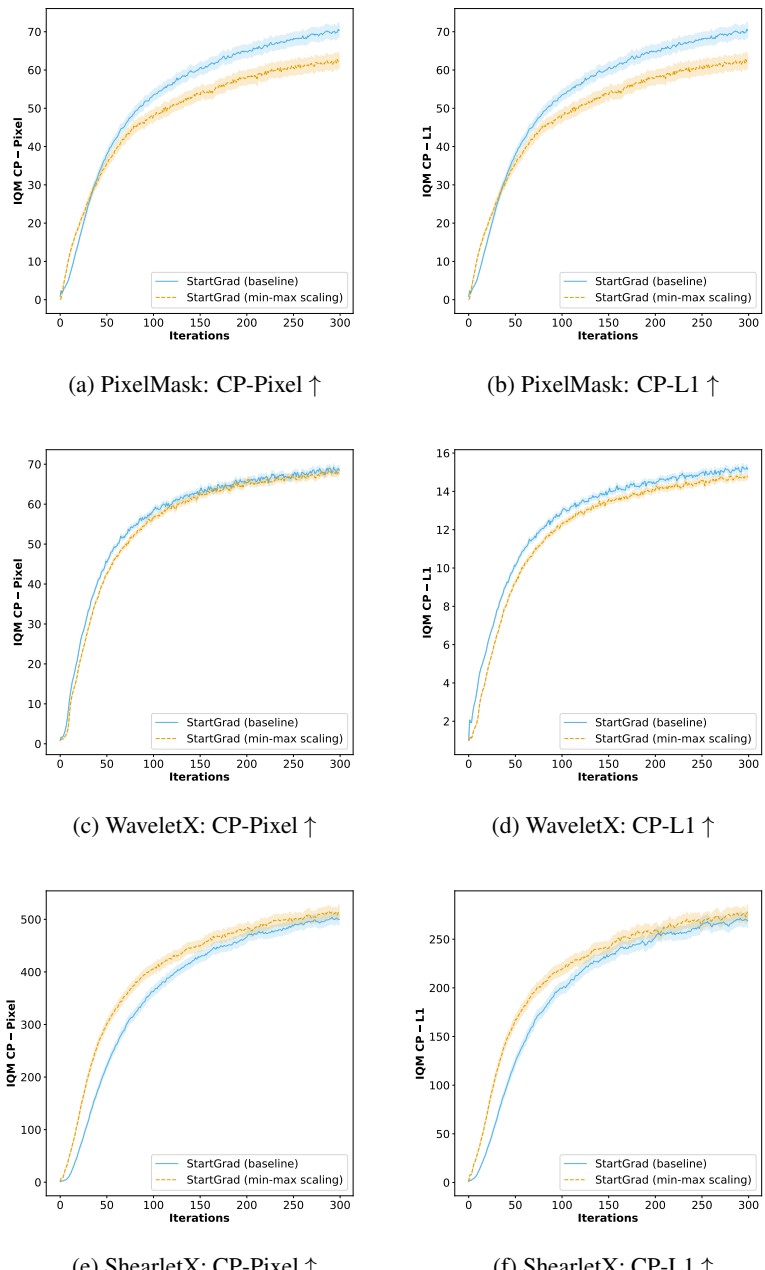

(a) PixelMask: CP-Pixel ↑         (b) PixelMask: CP-L1 ↑

(c) WaveletX: CP-Pixel ↑         (d) WaveletX: CP-L1 ↑

(e) ShearletX: CP-Pixel ↑         (f) ShearletX: CP-L1 ↑

Figure 25: Effect of using a different transformation function in place of the proposed quantile transformation function (QTF) for the PixelMask (Fong & Vedaldi, 2017) (first row), WaveletX (Kolek et al., 2023) (second row) and ShearletX (Kolek et al., 2023) (third row). We took the square root before the min-max scaling to account for the skewness typically observed in the dataset used (see section D). The solid line represents the average of the interquartile mean (IQM) performance across 500 randomly selected validation ImageNet (Deng et al., 2009) samples, while the shaded area denotes the standard errors respectively. ↑ indicates that higher is better, and ↓ that lower is better. We employ a pretrained ResNet18 (He et al., 2016) model as a classifier.

## F.12  Effect of varying $\lambda_1$ for PixelMask

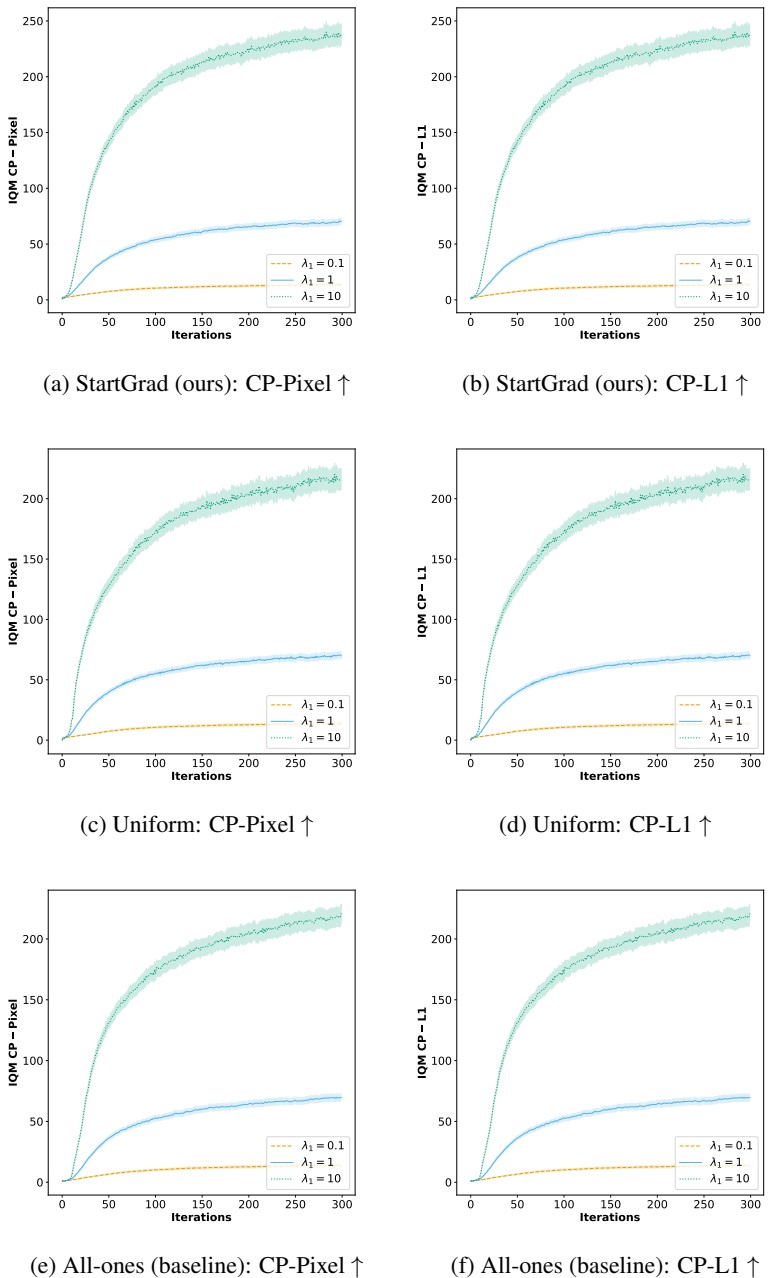

Figure 26: Effect of varying the $\lambda_1$ hyperparameter for the PixelMask (Fong & Vedaldi, 2017) model initialized with StartGrad (first row), uniform (second row) and all-ones (third row) on the interquartile mean (IQM) performance across 250 randomly selected validation ImageNet (Deng et al., 2009) samples for the CP-Pixel score (first column) and the CP-L1 score (second column). For each metric, $\uparrow$ indicates that higher is better, and $\downarrow$ that lower is better. We employ a pretrained ResNet18 (He et al., 2016) model as a classifier. Note that the CP-Pixel score is identical to the CP-L1 score as PixelMask (Fong & Vedaldi, 2017) framework does not apply the mask to a latent representation. Increasing $\lambda_1$ improves the CP-L1 and CP-Pixel metric across all initialization methods.

### F.13 PixelMask Performance $\lambda_1 = 10$

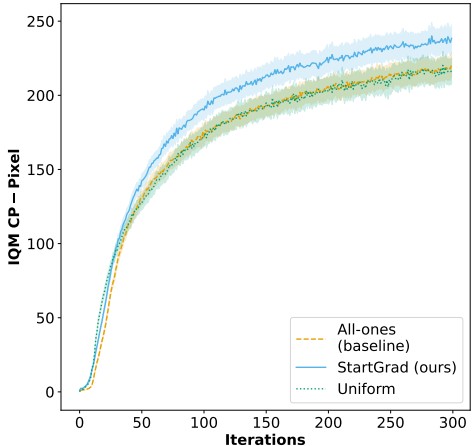

(a) PixelMask: CP-Pixel ↑

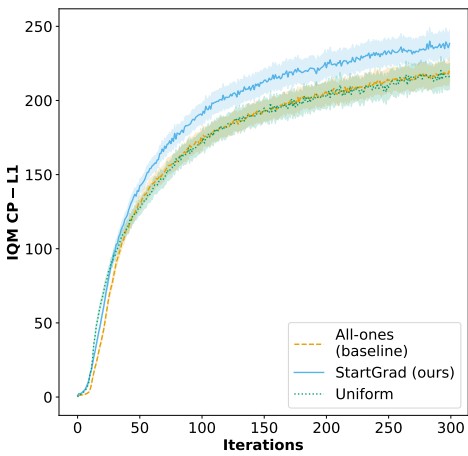

(b) PixelMask: CP-L1 ↑

Figure 27: Comparison of StartGrad initialization (ours) with standard baseline initialization schemes for the PixelMask (Fong & Vedaldi, 2017) model for $\lambda_1 = 10$. The solid line represents the average of the interquartile mean (IQM) CP-Pixel (a) and CP-L1 (b) performance across 250 randomly selected validation ImageNet (Deng et al., 2009) samples, while the shaded area denotes the standard errors respectively. For each metric, ↑ indicates that higher is better, and ↓ that lower is better. The CP-Pixel score is identical to the CP-L1 score as the mask-based model does not apply the mask to a latent representation. We employ a pretrained ResNet18 model (He et al., 2016) model as a classifier. We see that compared to all-ones and uniform, StartGrad leads to a performance boost.

## F.14 EFFECT OF VARYING $\lambda_1$ FOR WAVELETX

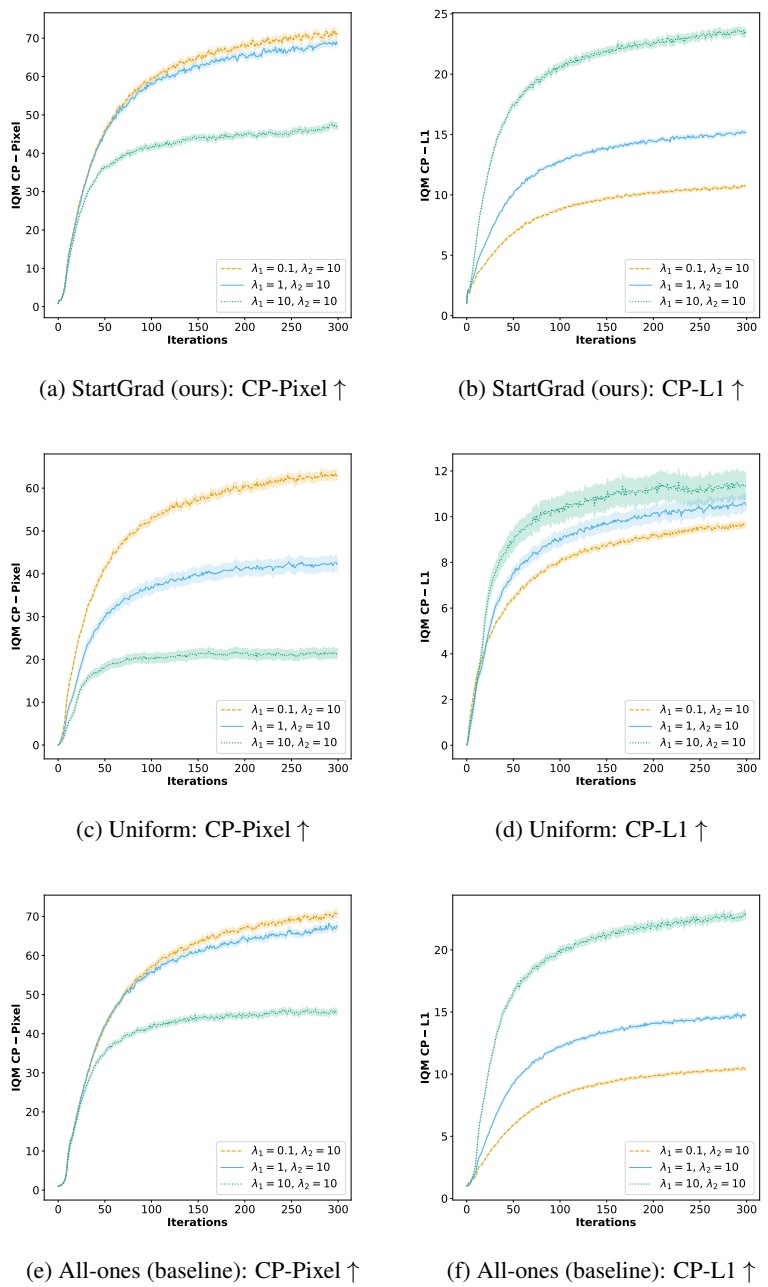

(a) StartGrad (ours): CP-Pixel ↑

(b) StartGrad (ours): CP-L1 ↑

(c) Uniform: CP-Pixel ↑

(d) Uniform: CP-L1 ↑

(e) All-ones (baseline): CP-Pixel ↑

(f) All-ones (baseline): CP-L1 ↑

Figure 28: Effect of varying the $\lambda_1$ hyperparameter for the WaveletX (Kolek et al., 2023) model initialized with StartGrad (first row), uniform (second row) and all-ones (third row) on the interquartile mean (IQM) performance across 500 randomly selected validation ImageNet (Deng et al., 2009) samples for the CP-Pixel score (first column) and the CP-L1 score (second column). For each metric, ↑ indicates that higher is better, and ↓ that lower is better. We employ a pretrained ResNet18 (He et al., 2016) model as a classifier. Increasing $\lambda_1$ improves the CP-L1 metric across all initialization methods, but decreases the CP-Pixel metric. The hyperparameter choice of $\lambda_1 = 1$ and $\lambda_2 = 10$ therefore represents the middle ground with respect to CP performance.

## F.15   EFFECT OF VARYING $\lambda_2$ FOR WAVELETX

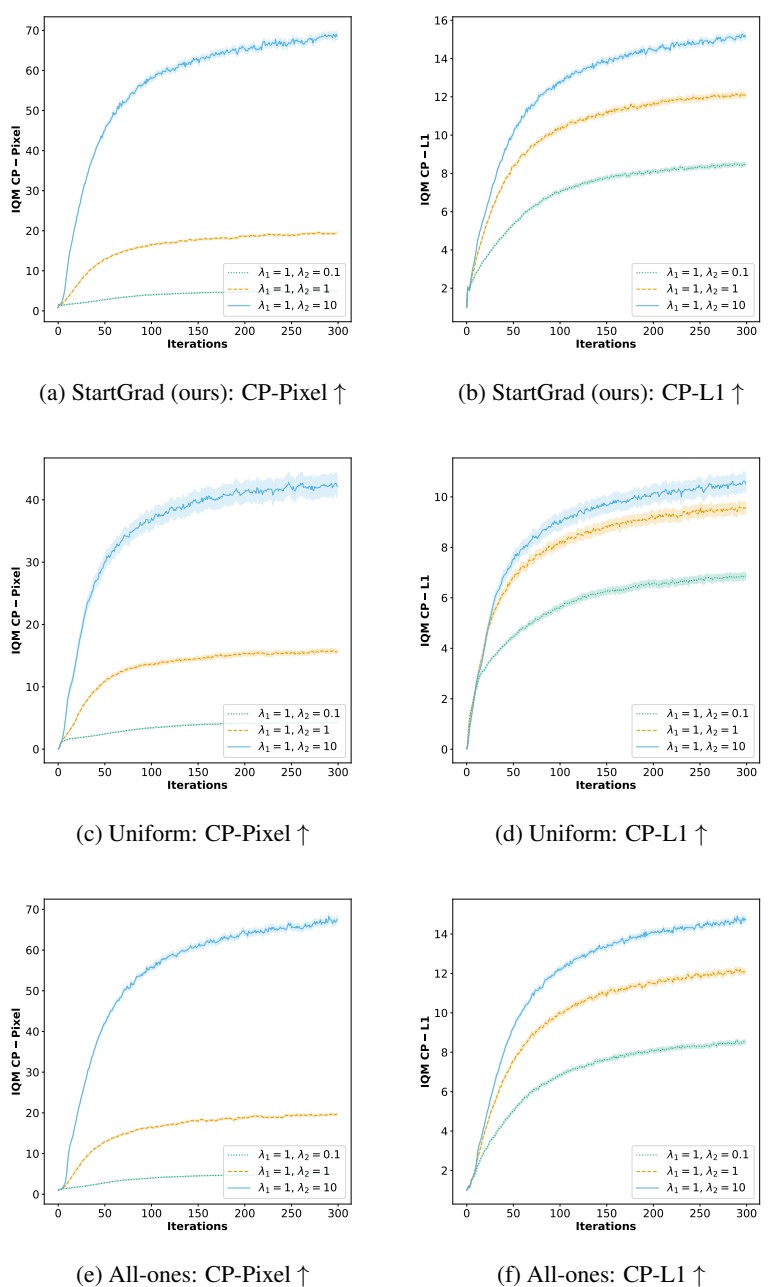

Figure 29: Effect of varying the $\lambda_2$ hyperparameter for the WaveletX (Kolek et al., 2023) model initialized with StartGrad (first row), uniform (second row) and all-ones (third row) on the interquartile mean (IQM) performance across 500 randomly selected validation ImageNet (Deng et al., 2009) samples for the CP-Pixel score (first column) and the CP-L1 score (second column). For each metric, ↑ indicates that higher is better, and ↓ that lower is better. We employ a pretrained ResNet18 (He et al., 2016) model as a classifier. Increasing the $\lambda_2$ improves the scores across all initialization methods. The hyperparameter choice of $\lambda_1 = 1$ and $\lambda_2 = 10$ leads to the best overall performance across all initialization methods.

## F.16 EFFECT OF VARYING $\lambda_1$ FOR SHEARLETX

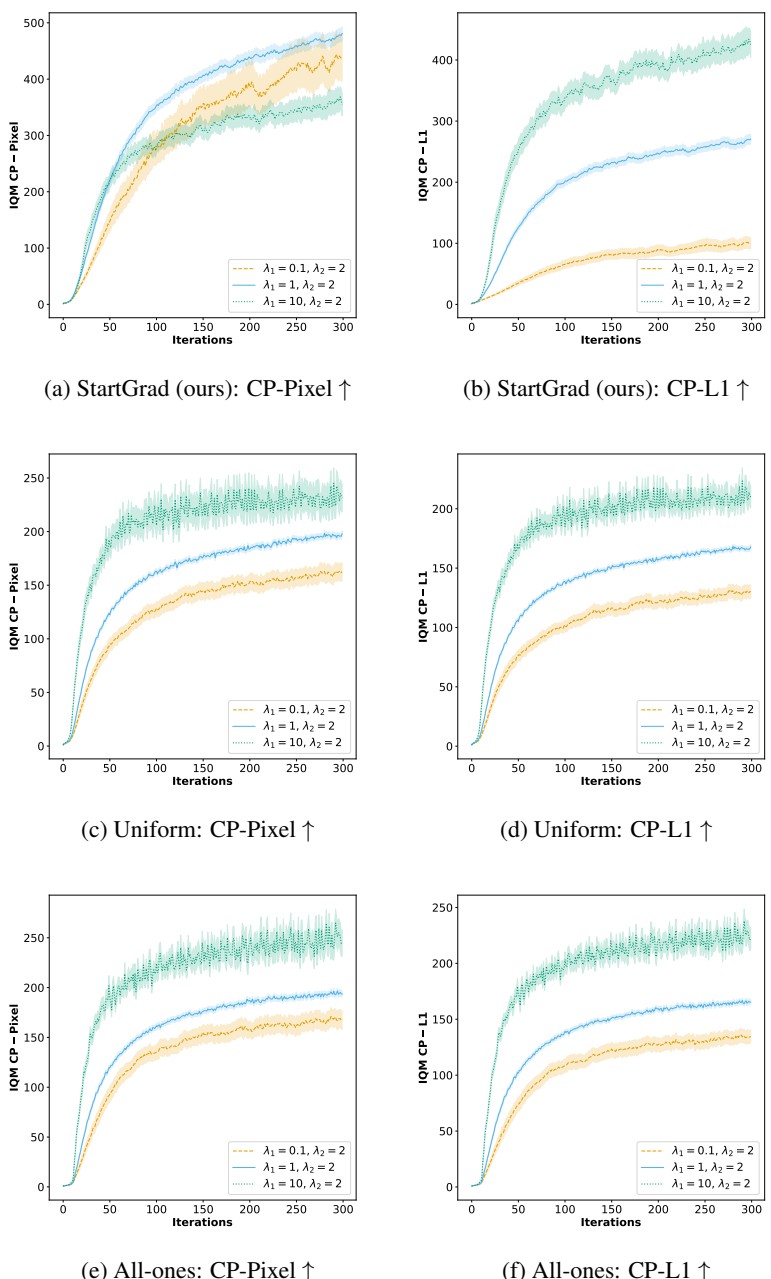

Figure 30: Effect of varying the $\lambda_1$ hyperparameter for the ShearletX model initialized with Start-Grad (first row), uniform (second row) and all-ones (third row) on the interquartile mean (IQM) performance for the CP-Pixel score (first column) and the CP-L1 score (second column). For each metric, $\uparrow$ indicates that higher is better, and $\downarrow$ that lower is better. We employ a pretrained ResNet18 (He et al., 2016) model as a classifier. Due to time constraints, 500 samples were used for the $\lambda_1 = 1$ and $\lambda_2 = 2$, whereas for the other hyperparameter we used a sample size of 50 explaining the larger variation. With the exception of the CP-Pixel score for the StartGrad initialization, increasing the $\lambda_1$ increases both the CP-L1 and CP-Pixel scores across all initialization methods for both metrics.

## F.17 EFFECT OF VARYING $\lambda_2$ FOR SHEARLETX

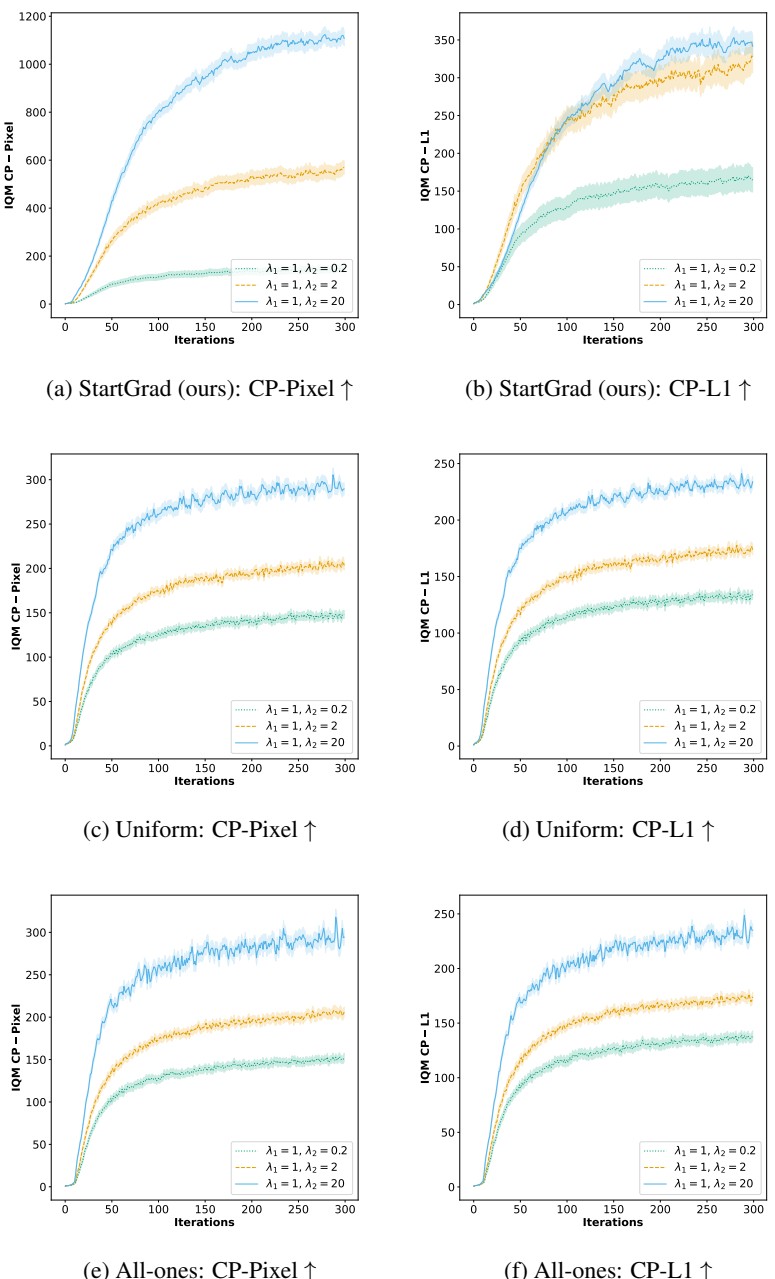

Figure 31: Effect of varying the $\lambda_2$ hyperparameter for the ShearletX (Kolek et al., 2023) model initialized with StartGrad (first row), uniform (second row) and all-ones (third row) on the interquartile mean (IQM) performance across 500 randomly selected validation ImageNet (Deng et al., 2009) samples for the CP-Pixel score (first column) and the CP-L1 score (second column). For each metric, ↑ indicates that higher is better, and ↓ that lower is better. We employ a pretrained ResNet18 (He et al., 2016) model as a classifier. Increasing the $\lambda_2$ increases both the CP-L1 and CP-Pixel scores across all initialization methods for both metrics.

## F.18  VISUALIZATIONS PIXELMASK

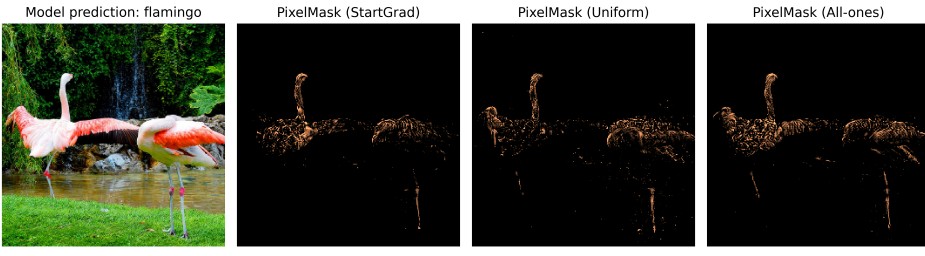

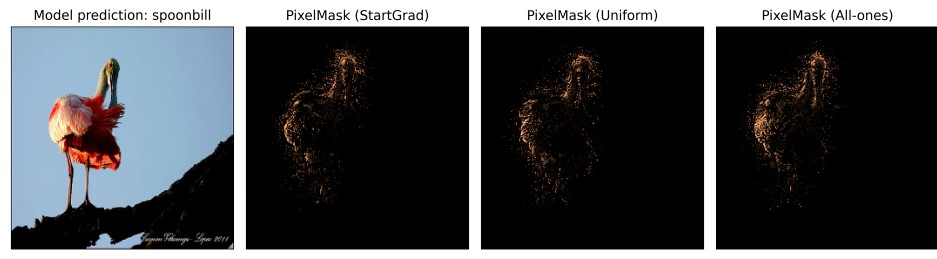

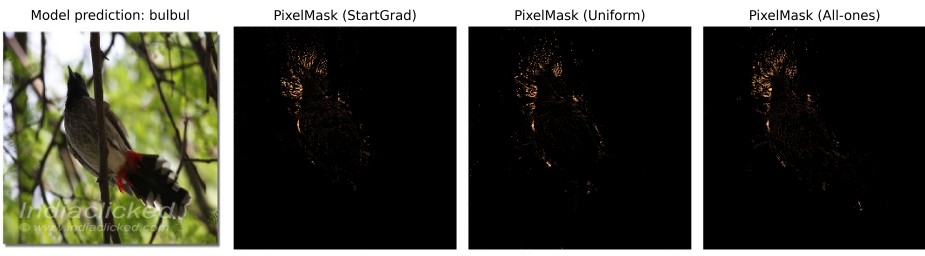

Figure 32: Illustrative visual examples of PixelMask (Fong & Vedaldi, 2017) applied to randomly selected validation ImageNet (Deng et al., 2009) samples using a pretrained ResNet18 (He et al., 2016) classifier. The optimization was run for 25 iteration steps to emphasize early visual differences between initialization methods.

## F.19   VISUALIZATIONS WAVELETX

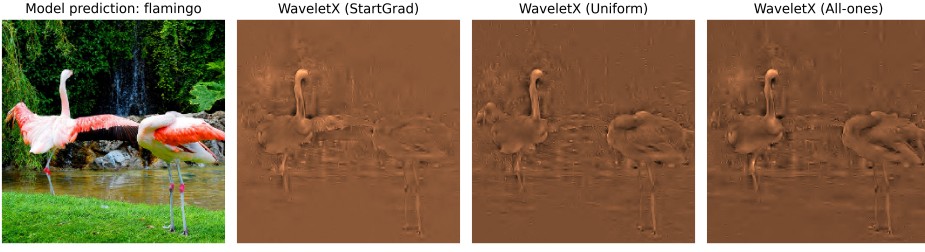

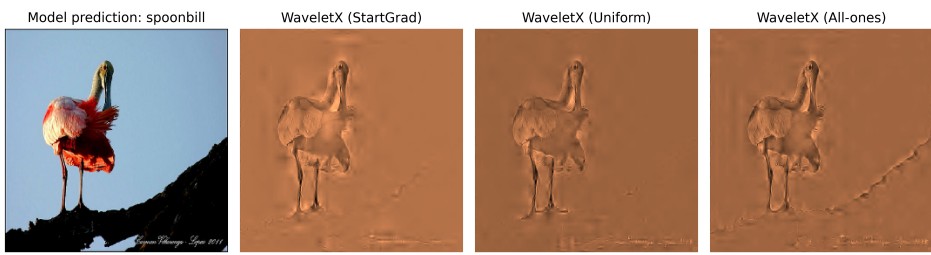

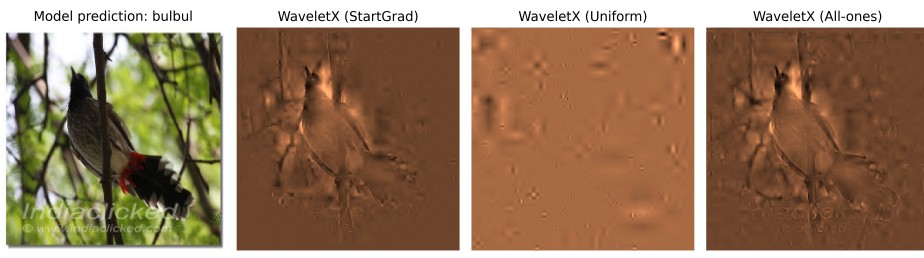

Figure 33: Illustrative visual examples of WaveletX (Kolek et al., 2023) applied to randomly selected validation ImageNet (Deng et al., 2009) samples using a pretrained ResNet18 (He et al., 2016) classifier. The optimization was run for 25 iteration steps to emphasize early visual differences between initialization methods.

F.20   VISUALIZATIONS SHEARLETX

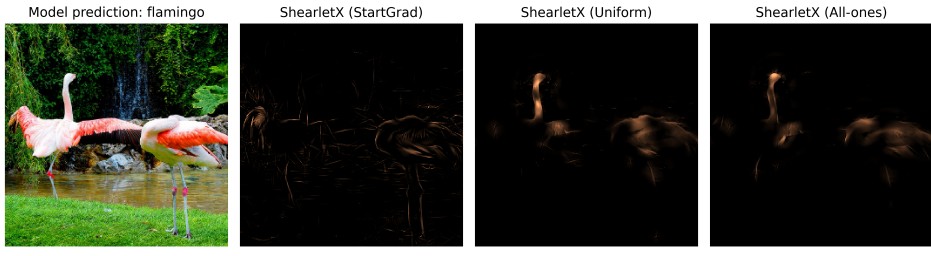

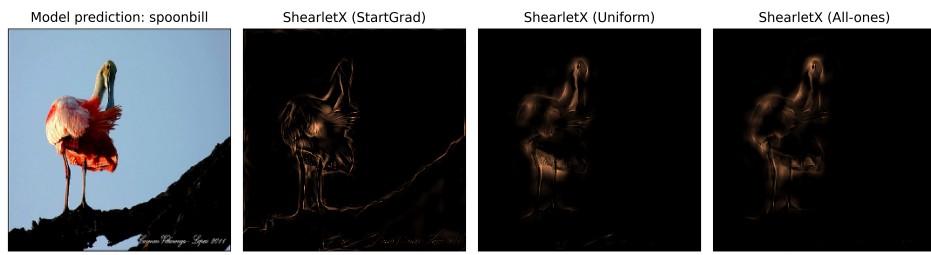

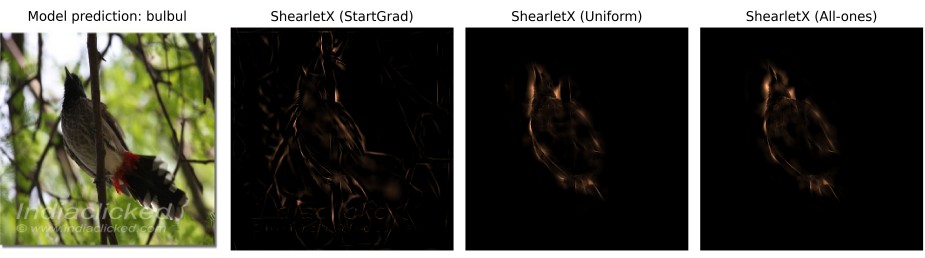

Figure 34: Illustrative visual examples of ShearletX (Kolek et al., 2023) applied to randomly selected validation ImageNet (Deng et al., 2009) samples using a pretrained ResNet18 (He et al., 2016) classifier. The optimization was run for 25 iteration steps to emphasize early visual differences between initialization methods. The results indicate that initializing with StartGrad preserves edges better, which are key for ShearletX's performance, compared to alternative initialization methods. This advantage in edge preservation likely explains StartGrad's superior performance by guiding the optimization to focus on critical features, such as edges, for improved mask-based explanations.

### F.21 RANDOM SEED STABILITY

#### F.21.1 VISION

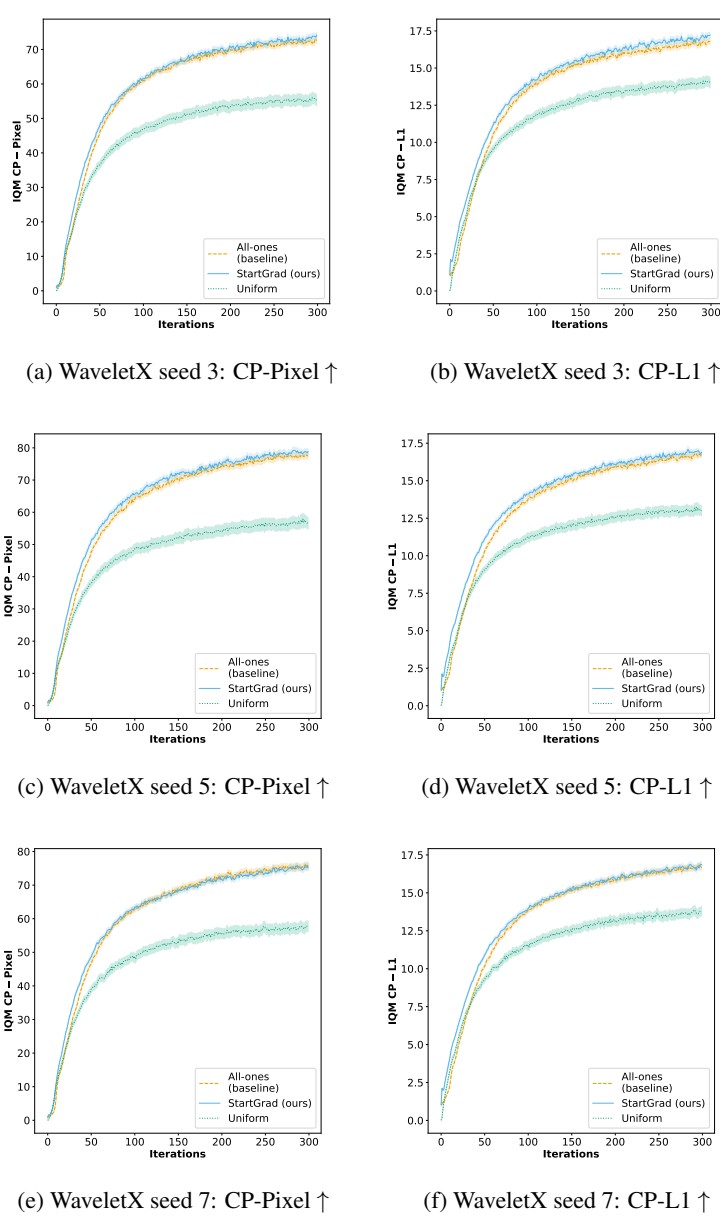

(a) WaveletX seed 3: CP-Pixel ↑      (b) WaveletX seed 3: CP-L1 ↑

(c) WaveletX seed 5: CP-Pixel ↑      (d) WaveletX seed 5: CP-L1 ↑

(e) WaveletX seed 7: CP-Pixel ↑      (f) WaveletX seed 7: CP-L1 ↑

Figure 35: Comparison of StartGrad initialization (ours) with standard baseline initialization schemes for the WaveletX (Kolek et al., 2023) explanation models across different seeds. Baseline refers to the originally used initialization scheme for the respective mask explanation method. The solid line represents the average of the interquartile mean (IQM) performance across 500 randomly selected validation ImageNet (Deng et al., 2009) samples, while the shaded area denotes the standard errors respectively. For each metric, ↑ indicates that higher is better, and ↓ that lower is better. We employ a pretrained ResNet18 (He et al., 2016) model as a classifier. The initialization methods show stable behavior across random seeds: StartGrad consistently outperforms uniform initialization and performs better (seeds 3, 5) or on par (for later iteration steps seed 7) with all-ones initialization.

### F.21.2  TIME SERIES

Table 10: Average performance of the ExtremalMask method (Enguehard, 2023) (preservation game objective) across iteration steps on the state dataset using StartGrad, all-ones, and uniform initializations. Reported values are the mean and standard deviation of nine runs of the experiments using different random seeds. Each experiment averages results over five folds as stated in the main paper. Metrics are denoted with ↑ (higher is better) or ↓ (lower is better). Baseline refers to the original initialization scheme for the method. Results more than one standard deviation better than the second-best are highlighted in bold.

| Metric | Initialization | Iteration steps | | | |
| --- | --- | --- | --- | --- | --- |
| | | 50 | 100 | 300 | 500 |
| AUP ↑ | Uniform (baseline) | $0.723 \pm 0.024$ | $0.805 \pm 0.025$ | $0.855 \pm 0.030$ | $0.857 \pm 0.030$ |
| | All-ones | $\mathbf{0.875 \pm 0.023}$ | $\mathbf{0.845 \pm 0.028}$ | $0.862 \pm 0.030$ | $0.862 \pm 0.030$ |
| | StartGrad (ours) | $0.834 \pm 0.031$ | $\mathbf{0.863 \pm 0.031}$ | $0.863 \pm 0.031$ | $0.863 \pm 0.031$ |
| AUR ↑ | Uniform (baseline) | $0.692 \pm 0.009$ | $0.727 \pm 0.008$ | $0.738 \pm 0.008$ | $0.739 \pm 0.007$ |
| | All-ones | $0.688 \pm 0.010$ | $\mathbf{0.755 \pm 0.008}$ | $0.744 \pm 0.008$ | $0.744 \pm 0.008$ |
| | StartGrad (ours) | $\mathbf{0.795 \pm 0.006}$ | $\mathbf{0.751 \pm 0.008}$ | $0.746 \pm 0.008$ | $0.746 \pm 0.008$ |
| I $[10^5]$ ↑ | Uniform (baseline) | $1.035 \pm 0.037$ | $2.863 \pm 0.189$ | $3.356 \pm 0.120$ | $3.376 \pm 0.117$ |
| | All-ones | $2.474 \pm 0.098$ | $3.269 \pm 0.112$ | $3.409 \pm 0.117$ | $3.418 \pm 0.117$ |
| | StartGrad (ours) | $\mathbf{3.353 \pm 0.163}$ | $\mathbf{3.405 \pm 0.134}$ | $3.421 \pm 0.121$ | $3.423 \pm 0.121$ |
| E $[10^4]$ ↓ | Uniform (baseline) | $1.866 \pm 0.088$ | $0.941 \pm 0.084$ | $0.777 \pm 0.063$ | $0.770 \pm 0.063$ |
| | All-ones | $1.561 \pm 0.058$ | $1.096 \pm 0.059$ | $0.773 \pm 0.060$ | $0.769 \pm 0.060$ |
| | StartGrad (ours) | $\mathbf{1.286 \pm 0.070}$ | $\mathbf{0.820 \pm 0.067}$ | $0.767 \pm 0.060$ | $0.766 \pm 0.060$ |

Table 11: Average performance of the ExtremalMask method (Enguehard, 2023) (deletion game objective) across iteration steps on the state dataset using StartGrad, all-ones, and uniform initializations. Reported values are the mean and standard deviation of nine runs of the experiments using different random seeds. Each experiment averages results over five folds as stated in the main paper. Metrics are denoted with ↑ (higher is better) or ↓ (lower is better). Baseline refers to the original initialization scheme for the method. Results more than one standard deviation better than the second-best are highlighted in bold.

| Metric | Initialization | Iteration steps | | | |
| --- | --- | --- | --- | --- | --- |
| | | 50 | 100 | 300 | 500 |
| AUP ↑ | Uniform (baseline) | $0.648 \pm 0.051$ | $0.721 \pm 0.050$ | $0.817 \pm 0.062$ | $0.853 \pm 0.064$ |
| | All-ones | $\mathbf{0.818 \pm 0.084}$ | $\mathbf{0.880 \pm 0.083}$ | $0.884 \pm 0.072$ | $0.886 \pm 0.070$ |
| | StartGrad (ours) | $\mathbf{0.815 \pm 0.062}$ | $0.853 \pm 0.074$ | $0.871 \pm 0.072$ | $0.875 \pm 0.071$ |
| AUR ↑ | Uniform (baseline) | $0.674 \pm 0.010$ | $0.712 \pm 0.009$ | $0.743 \pm 0.012$ | $0.743 \pm 0.013$ |
| | All-ones | $0.459 \pm 0.023$ | $0.601 \pm 0.015$ | $0.739 \pm 0.011$ | $0.741 \pm 0.012$ |
| | StartGrad (ours) | $\mathbf{0.804 \pm 0.009}$ | $\mathbf{0.751 \pm 0.012}$ | $0.740 \pm 0.012$ | $0.739 \pm 0.012$ |
| I $[10^5]$ ↑ | Uniform (baseline) | $1.112 \pm 0.072$ | $3.888 \pm 0.067$ | $4.217 \pm 0.083$ | $4.256 \pm 0.092$ |
| | All-ones | $0.664 \pm 0.056$ | $0.932 \pm 0.047$ | $4.223 \pm 0.078$ | $4.268 \pm 0.084$ |
| | StartGrad (ours) | $\mathbf{4.245 \pm 0.070}$ | $\mathbf{4.270 \pm 0.083}$ | $4.267 \pm 0.084$ | $4.266 \pm 0.084$ |
| E $[10^4]$ ↓ | Uniform (baseline) | $1.523 \pm 0.101$ | $0.493 \pm 0.019$ | $0.386 \pm 0.016$ | $0.338 \pm 0.020$ |
| | All-ones | $1.753 \pm 0.047$ | $1.838 \pm 0.071$ | $0.343 \pm 0.026$ | $0.317 \pm 0.026$ |
| | StartGrad (ours) | $\mathbf{1.004 \pm 0.033}$ | $\mathbf{0.414 \pm 0.015}$ | $\mathbf{0.311 \pm 0.021}$ | $0.304 \pm 0.021$ |

Table 12: Average performance of the ExtremalMask method (Enguehard, 2023) (preservation game objective) across iteration steps on the switch-feature dataset using StartGrad, all-ones, and uniform initializations. Reported values are the mean and standard deviation of nine runs of the experiments using different random seeds. Each experiment averages results over five folds as stated in the main paper. Metrics are denoted with ↑ (higher is better) or ↓ (lower is better). Baseline refers to the original initialization scheme for the method. Results more than one standard deviation better than the second-best are highlighted in bold.

| Metric | Initialization | Iteration steps | | | |
| | | 50 | 100 | 300 | 500 |
|---|---|---|---|---|---|
| AUP ↑ | Uniform (baseline) | $0.878 \pm 0.007$ | $0.964 \pm 0.003$ | $0.984 \pm 0.004$ | $0.985 \pm 0.004$ |
| | All-ones | $\mathbf{0.947 \pm 0.002}$ | $0.961 \pm 0.006$ | $0.984 \pm 0.004$ | $0.986 \pm 0.003$ |
| | StartGrad (ours) | $0.920 \pm 0.008$ | $\mathbf{0.982 \pm 0.003}$ | $0.986 \pm 0.003$ | $0.986 \pm 0.003$ |
| AUR ↑ | Uniform (baseline) | $0.724 \pm 0.012$ | $0.725 \pm 0.013$ | $0.733 \pm 0.018$ | $0.732 \pm 0.017$ |
| | All-ones | $0.689 \pm 0.010$ | $\mathbf{0.761 \pm 0.014}$ | $0.739 \pm 0.018$ | $0.738 \pm 0.018$ |
| | StartGrad (ours) | $\mathbf{0.799 \pm 0.016}$ | $0.747 \pm 0.019$ | $0.739 \pm 0.019$ | $0.739 \pm 0.018$ |
| I $[10^5]$ ↑ | Uniform (baseline) | $1.032 \pm 0.037$ | $2.039 \pm 0.149$ | $2.661 \pm 0.194$ | $2.668 \pm 0.198$ |
| | All-ones | $1.978 \pm 0.059$ | $\mathbf{2.597 \pm 0.162}$ | $2.678 \pm 0.198$ | $2.674 \pm 0.200$ |
| | StartGrad (ours) | $\mathbf{2.617 \pm 0.183}$ | $2.668 \pm 0.193$ | $2.675 \pm 0.200$ | $2.674 \pm 0.198$ |
| E $[10^4]$ ↓ | Uniform (baseline) | $2.266 \pm 0.098$ | $1.487 \pm 0.080$ | $1.126 \pm 0.109$ | $1.162 \pm 0.114$ |
| | All-ones | $1.933 \pm 0.034$ | $1.475 \pm 0.075$ | $1.137 \pm 0.110$ | $1.137 \pm 0.109$ |
| | StartGrad (ours) | $\mathbf{1.604 \pm 0.104}$ | $\mathbf{1.232 \pm 0.099}$ | $1.138 \pm 0.107$ | $1.136 \pm 0.107$ |

Table 13: Average performance of the ExtremalMask method (Enguehard, 2023) (deletion game objective) across iteration steps on the switch-feature dataset using StartGrad, all-ones, and uniform initializations. Reported values are the mean and standard deviation of nine runs of the experiments using different random seeds. Each experiment averages results over five folds as stated in the main paper. Metrics are denoted with ↑ (higher is better) or ↓ (lower is better). Baseline refers to the original initialization scheme for the method. Results more than one standard deviation better than the second-best are highlighted in bold.

| Metric | Initialization | Iteration steps | | | |
| | | 50 | 100 | 300 | 500 |
|---|---|---|---|---|---|
| AUP ↑ | Uniform (baseline) | $0.835 \pm 0.039$ | $0.894 \pm 0.050$ | $0.930 \pm 0.060$ | $0.935 \pm 0.058$ |
| | All-ones | $0.764 \pm 0.075$ | $0.751 \pm 0.074$ | $0.804 \pm 0.086$ | $0.805 \pm 0.093$ |
| | StartGrad (ours) | $\mathbf{0.880 \pm 0.024}$ | $\mathbf{0.926 \pm 0.031}$ | $0.949 \pm 0.027$ | $0.954 \pm 0.026$ |
| AUR ↑ | Uniform (baseline) | $0.602 \pm 0.029$ | $\mathbf{0.598 \pm 0.029}$ | $\mathbf{0.607 \pm 0.032}$ | $\mathbf{0.599 \pm 0.032}$ |
| | All-ones | $0.638 \pm 0.054$ | $0.527 \pm 0.025$ | $0.554 \pm 0.039$ | $0.547 \pm 0.043$ |
| | StartGrad (ours) | $\mathbf{0.670 \pm 0.020}$ | $0.607 \pm 0.019$ | $0.598 \pm 0.012$ | $0.595 \pm 0.011$ |
| I $[10^5]$ ↑ | Uniform (baseline) | $0.763 \pm 0.053$ | $1.620 \pm 0.189$ | $\mathbf{2.139 \pm 0.211}$ | $\mathbf{2.107 \pm 0.188}$ |
| | All-ones | $0.830 \pm 0.099$ | $0.553 \pm 0.028$ | $1.354 \pm 0.180$ | $1.570 \pm 0.277$ |
| | StartGrad (ours) | $\mathbf{1.740 \pm 0.094}$ | $\mathbf{1.987 \pm 0.096}$ | $2.100 \pm 0.055$ | $2.098 \pm 0.044$ |
| E $[10^4]$ ↓ | Uniform (baseline) | $2.500 \pm 0.084$ | $1.685 \pm 0.097$ | $\mathbf{1.388 \pm 0.036}$ | $\mathbf{1.365 \pm 0.031}$ |
| | All-ones | $2.339 \pm 0.073$ | $2.860 \pm 0.101$ | $1.889 \pm 0.079$ | $1.649 \pm 0.099$ |
| | StartGrad (ours) | $\mathbf{2.404 \pm 0.028}$ | $\mathbf{1.613 \pm 0.051}$ | $\mathbf{1.354 \pm 0.036}$ | $\mathbf{1.336 \pm 0.036}$ |

