# OpenReview forum: "Start Smart: Leveraging Gradients For Enhancing Mask-based XAI Methods"
_ICLR.cc/2025/Conference — ICLR 2025 Poster_

### Official Review · Reviewer_A8Pa · 2024-10-23

**Soundness:** 3
**Presentation:** 3
**Contribution:** 3
**Rating:** 6
**Confidence:** 3

**Summary:**

This paper introduce StartGrad, a efficient novel-gradient based mask initialize method designed for mask-based post-hoc attribution method, such as ShearletX or WaveletX. StartGrad utilizes gradients of the input features to initialize the masks more effectively, resulting fast convergence. The authors also theoretically proved that StartGrad is superior to commonly used initialized strategies like uniform or all-ones initialization.

The authors conduct extensive experiments across both vision and time-series domains. The results showed that StartGrad consistantly accelerates the optimization process.

**Strengths:**

This paper is well written; especially when the authors introduce rate-distortion explanation(RDE) framework. The methodology, theoretical background is well organized and clearly explained.

Good novelty. The authors introduced a new perspective by focusing on mask initialization that has been overlooked. This approach seems to be radical rather than incremental.

The authors provide solid theoretical proofs under the RDE framework, showing that StartGrad offers a better trade-off between distortion and sparsity compared to traditional approaches.

The authors conducted extensive experiments across different domains(vision and time-series) with state-of-art RDE explanation methods, and demonstrated the effectiveness and practicality of StartGrad.

Extensive ablation studies. The authors provide extensive ablation studies, analyzing effect of various components such as quantile transformation.

**Weaknesses:**

Lack of Visualization: It would be beneficial to include a figure explaining the StartGrad method. Additionally, providing qualitative results for StartGrad would enhance the paper—for example, illustrating the differences in attribution maps after 50 iterations with different initializations of ExtremalMask.

**Questions:**

Can the RDE framework's attribution maps handle multi-label objects in a single image, such as an image containing both a cat and a dog? (This question is not related to the paper's rating; it's just out of the reviewer's curiosity.)

To quantify attribution methods on vision dataset, the fidelity metrics Most Relevant First (MoRF) deletion and Least Relevant First (LeRF) deletion are widely used. If these fidelity metrics were employed, comparison with other state-of-the-art attribution methods, such as the Layer-wise Relevance Propagation (LRP) family and the Class Activation Map (CAM) family, would have been possible. Why did the authors choose the conciseness-preciseness (CP) Pixel and L1 scores rather than MoRF and LeRF?

---

> ### Author Response · Authors · 2024-11-21
>
> Dear reviewer A8Pa,
>
> First of all, we would like to thank you very much for your thorough and insightful review. We sincerely appreciate the time you spent evaluating our work, and very much appreciate your positive comments about the clarity of our methodology, the novelty of our approach, and the scope of our experimental evaluation.
> We also sincerely appreciate your thoughtful questions and will address all your comments, questions, and concerns in detail.
>
> **Weaknesses: Lack of visualization**
>
> We agree that adding visualizations could be beneficial for the reader. For this reason, we added a visualization of the StartGrad algorithm in Figure 2 as an aid to facilitate the understanding of the pseudocode which we put in the Appendix B.1 in the revised version of the paper. Unfortunately, due to space limitations, we could not fit the visualization aid within the 10 page limits.
> We hope that the visualization will be helpful for the reader to understand the StartGrad algorithm, alongside the pseudocode that was already included in the first version of the paper.
>
> For the final version of the paper, we aim to also add some additional qualitative results of StartGrad.
>
> **Question 1: RDE framework for handling multi-label classification**
>
> Yes, the RDE framework can indeed be adapted to handle multi-label objectives. Specifically, the distortion function can be modified to account for multiple target labels simultaneously. In this scenario, the objective of the RDE framework would shift to identifying a sparse mask that minimizes distortion across the predictions of the top-k classes. For instance, one could measure the Mean Squared Error (MSE) between the original and perturbed predictions for each relevant label (e.g., both "cat" and "dog" in a multi-label setting). While our current experiments focus on single label predictions for simplicity, the flexibility of RDE makes it well-suited for extending the framework to multi-label classification tasks which further underscores the broad potential of using the RDE framework.
>
> **Question 2: Justification of the choice of the CP-scores as opposed to Insertion and Deletion score and comparison other alternative post-hoc explainability methods**
>
> We thank the reviewer for their insightful comments regarding our choice of evaluation metrics. Here, we provide a detailed explanation of why we selected the conciseness-preciseness (CP) Pixel and L1 scores instead of the commonly-used insertion and deletion scores.
>
> Insertion and deletion scores are effective evaluation metrics for methods that rank feature importance. However, the mask-based methods in our study aim to learn a binary, sparse mask by optimizing an objective that approximates $L_{0}$-norm through a $L_{1}$-norm. Since mask-based explanation methods do not inherently rank feature importance, applying insertion and deletion metrics, which rely on ordered rankings, becomes problematic. This limitation has been discussed in [1], who introduced CP-scores to address such challenges, and we align with their approach to enable meaningful comparisons of StartGrad with alternative initialization schemes.
>
> To illustrate this issue, we computed faithfulness scores (difference between insertion and deletion) for PixelMask, WaveletX, and ShearletX across iterations and found that faithfulness scores consistently decrease during optimization. This counterintuitive behavior highlights the unsuitability of the faithfulness metric for mask-based methods due to their lack of inherently ordering mask values. For completeness, we also included Integrated Gradients (IG) [2] and Grad-CAM [3] for comparison reasons. In the revised version of our paper, we clarified our rationale by expanding the justification for the CP scores in the main paper, and by adding supplementary explanations and figures in Appendix C.2.2.
>
> **References**:
>
> [1] Stefan Kolek, Robert Windesheim, Hector Andrade Loarca, Gitta Kutyniok, and Ron Levie. Explaining image classifiers with multiscale directional image representation. In Proceedings of the IEEE Conference on Computer Vision and Pattern Recognition (CVPR). IEEE, 2023.
>
> [2] Mukund Sundararajan, Ankur Taly, and Qiqi Yan. Axiomatic attribution for deep networks. In Doina Precup and Yee Whye Teh (eds.), Proceedings of the 34th International Conference on Machine Learning, volume 70 of Proceedings of Machine Learning Research, pp. 3319–3328. PMLR, 06– 11 Aug 2017.
>
> [3] Ramprasaath R. Selvaraju, Michael Cogswell, Abhishek Das, Ramakrishna Vedantam, Devi Parikh, and Dhruv Batra. Grad-cam: Visual explanations from deep networks via gradient-based local- ization. In 2017 IEEE International Conference on Computer Vision (ICCV), pp. 618–626, 2017. doi: 10.1109/ICCV.2017.74.
>
> **Final words**
>
> We hope that these responses clarify your questions and concerns. Please let us know if you have any follow-up / additional questions.
>
> Best regards,
>
> The Authors

---

> ### Comment · Reviewer_A8Pa · 2024-11-26
>
> Thank you for the clear explanation.
>
> For the multi-label classification section, it would have been better to include example images that explain multi-label scenarios, such as an image containing both a cat and a dog. However, since it is too late at this stage, this is not necessary.

---

> ### Comment · Reviewer_A8Pa · 2024-11-27
>
> I apologize for my initial comment; I did not carefully review the responses from the reviewers.
>
> 1. I acknowledge that the Insertion-Deletion metric cannot be applied to binary mask attributions. However, by using a quantile transformation function, methods like Integrated Gradients and Grad-CAM can be converted into binary mask attributions, allowing CP-scores to be applied. I believe that a proper comparison using this approach should have been included. That said, given the time constraints, this addition is not strictly necessary.
>
> 2. Considering the two-week review period, it is concerning that no qualitative comparisons between different initialization strategies were provided. Since qualitative results are a natural complement to quantitative findings, including such comparisons—particularly in the appendix—should have been straightforward. While the authors cannot amend the PDF at this stage, I strongly suggest they include qualitative comparisons between StartGrad and other methods, especially **quantile-transformed Integrated Gradients** and **quantile-transformed Grad-CAM**, via an anonymous GitHub link.
>
> 3. Additionally, I share the concern raised by Reviewer 1UMB regarding potential overclaims. While all propositions and remarks are well-derived, they do not appear to strongly connect to the performance of the mask *after* optimization concludes. The authors must explicitly explain how better initialization leads to better final mask performance. Although almost half of the experiments support this claim, the authors have not adequately articulated it in the paper.

---

> ### Comment · Reviewer_A8Pa · 2024-11-29
>
> I have carefully reviewed all the qualitative results, as well as the metrics for IG and GradCAM on insertion, deletion, and CP scores. However, I still find it unconvincing that the CP-score is a faithful metric. This is related to the following question:
>
> ## Is the Binary Mask Necessary?
>
> The use of binary masks raises a fundamental question about their validity in faithfully representing attribution. Vision models utilize diverse information from important regions, including shape and texture. Proper attribution often requires dense pixel-level information. For instance, if a bird's white-gray fur texture is critical for classifying it as a bulbul (since many birds share the same general shape), it may be true that some white pixels are more important than gray pixels. However, the gray pixels still play a crucial role and cannot be ignored even though they have less importance than rarer white pixels.
>
> Binary masks fail to capture this nuance, as they overlook regions that are not densely important. Instead, they overly emphasize shape, potentially ignoring finer-grained but essential details like texture and color. Therefore, can we truly claim that this binary attribution method, which cannot assign nuanced importance to pixels (or wavelets), is faithful for image classification?
>
> ---
>
> ## Assessment of Contribution and Final Decision
>
> After reflecting on the contribution of the method carefully, I find it somewhat marginal. This is particularly evident since all the propositions and remarks apply only at the initialization stage, and the paper lacks clarity in explaining the connection between initialization performance and final performance.
>
> That said, the extensive experiments and practical applicability of the work in the field of mask-based attribution still hold merit. While my score has decreased compared to the starting phase of the discussion, the paper remains *slightly* above my acceptance threshold.

---

> ### Author Response · Authors · 2024-11-29
> **Response reviewer 1A8Pa: Thank you**
>
> Dear reviewer A8Pa,
>
> Thank you for your thoughtful and constructive feedback throughout this process. We sincerely appreciate the time and effort you’ve dedicated to reviewing our work and engaging in this discussion.
>
> We acknowledge your concern regarding the limitations of binary masks in capturing fine-grained attributions, such as texture and color, which are critical for nuanced image classification tasks. While binary masks are effective for highlighting sparse explanations — a core aspect of interpretability — they may overlook subtle yet significant features like texture. We believe that ShearletX is for instance a valuable first approach, as this model is explicitly designed for preserving important edges in the final explanation by operating in the shearlet space. However, we agree that exploring new methods that better account for subtle nuances such as texture, is an exciting direction for future research. We therefore thank you for your valuable insights as these not only provide important guidance for us, but also for the broader XAI research community.
>
> To address some of your concerns regarding our focus on initialization and its impact on final performance, we have added further clarifications and supporting results in the revised paper, including additional experiments and figures (see Appendix F.6, F.7, F.8). These results illustrate how gradient-informed initialization as done with our novel algorithm StartGrad provides a crucial early advantage by guiding optimization toward regions with minimal distortion. This early benefit proves critical, as uniform initialization, lacking gradient-based guidance, fails to achieve comparable distortion reduction, even by the final iteration. These findings highlight the practical significance of initialization strategies and directly connect initialization to final mask performance.
>
> **Closing Remarks**
>
> We are particularly grateful for your acknowledgment of the practical relevance of our work and the extensive experiments we conducted. Your thoughtful suggestions have been instrumental in improving the quality of this paper. Thank you again for your review and your final decision to keep the score above the acceptance threshold.
>
> Best regards,
>
> The Authors

---

### Official Review · Reviewer_1UMB · 2024-10-27

**Soundness:** 3
**Presentation:** 3
**Contribution:** 3
**Rating:** 5
**Confidence:** 3

**Summary:**

The paper introduces a post-hoc explainability technique within the domain of XAI, focusing on a mask-based approach to identify which parts of an input are most crucial for generating an explanation. Existing approaches aim to optimize the mask such that the masked input achieves similar predictive performance as the original. However, the authors address an overlooked aspect: the initialization of these masks. Their method operates on the premise that standard initialization techniques are suboptimal. They propose and mathematically demonstrate that initializing the mask based on the most salient gradients is more effective. By using a quantile transformation function to identify the top gradients, they initialize the mask in a more targeted manner. This technique, named StartGrad, aims to improve the performance of mask-based methods from the start.

**Strengths:**

The writing is clear and easy to read.
The paper provides mathematical proofs to support most of its claims.
The topic is engaging and relevant.

**Weaknesses:**

**1. Typographical Error**

It appears that lines 217–218 contradict Equation 7. This discrepancy could lead to confusion and should be addressed for clarity.

**2. Overclaims**

The paper seems to overstate certain findings. For instance, lines 175–176 present an equation without constraints on the mask, and much of the analysis substitutes entropy with a norm p, which limits generalizability. Propositions 2 and 3 focus solely on initialization rather than the optimization process as a whole. This raises a question: can we be confident that StartGrad improves optimization across all optimizers, or might this only hold in specific cases?

**3. Experimental Section**

Results and Scope: Table 1 does not appear to provide results for StartGrad, which limits the insight we can gain into the method’s effectiveness. Additionally, the experimental section is underdeveloped. The current setup only uses ResNet-18 on a subset of 500 random ImageNet images, which restricts the findings’ generalizability in the vision domain. Including additional datasets, such as Pascal-Part [2] and Monumai[1], where ground truth is available, would strengthen the empirical validation. It would also be interesting to test the techniques on more DNN such as Swin Transformer and ViT.
*Convergence Speed:* It would be valuable to evaluate whether the proposed initialization method enhances the convergence speed of XAI algorithms.
*Comparative Analysis:* Comparing StartGrad to other post-hoc explainability [3,4,5,6,7,8] techniques would provide more context and relevance to the findings.

**4. Metrics Used**

The choice of metrics is unclear. A more detailed explanation of the metrics used in the evaluation would be beneficial. Additionally, considering established XAI metrics from frameworks like Quantus or Xplique could improve comparability and transparency.

[1] Lamas, Alberto, et al. "MonuMAI: Dataset, deep learning pipeline and citizen science based app for monumental heritage taxonomy and classification." Neurocomputing 420 (2021): 266-280.

[2] Chen, Xianjie, et al. "Detect what you can: Detecting and representing objects using holistic models and body parts." Proceedings of the IEEE conference on computer vision and pattern recognition. 2014.

[3] Selvaraju, R. R., Das, A., Vedantam, R., Cogswell, M., Parikh, D., & Batra, D. (2016). Grad-CAM: Why did you say that?. arXiv preprint arXiv:1611.07450.

Jamil, Md Shafayat, et al. "Advanced gradcam++: improved visual explanations of CNN decisions in diabetic retinopathy." Computer Vision and Image Analysis for Industry 4.0. Chapman and Hall/CRC, 2023. 64-75.

[4] Wang, Haofan, et al. "Score-CAM: Score-weighted visual explanations for convolutional neural networks." Proceedings of the IEEE/CVF conference on computer vision and pattern recognition workshops. 2020.

[5] Muhammad, Mohammed Bany, and Mohammed Yeasin. "Eigen-cam: Class activation map using principal components." 2020 international joint conference on neural networks (IJCNN). IEEE, 2020.

[6] Srinivas, Suraj, and François Fleuret. "Full-gradient representation for neural network visualization." Advances in neural information processing systems 32 (2019).

[7] Sattarzadeh, Sam, et al. "Integrated grad-cam: Sensitivity-aware visual explanation of deep convolutional networks via integrated gradient-based scoring." ICASSP 2021-2021 IEEE International Conference on Acoustics, Speech and Signal Processing (ICASSP). IEEE, 2021.

[8] Kapishnikov, Andrei, et al. "Guided integrated gradients: An adaptive path method for removing noise." Proceedings of the IEEE/CVF conference on computer vision and pattern recognition. 2021.

**Questions:**

See most of the questions on the weakness.

**Details Of Ethics Concerns:**

1. Typographical Error

It appears that lines 217–218 contradict Equation 7. This discrepancy could lead to confusion and should be addressed for clarity.

2. Overclaims

The paper seems to overstate certain findings. For instance, lines 175–176 present an equation without constraints on the mask, and much of the analysis substitutes entropy with a norm p, which limits generalizability. Propositions 2 and 3 focus solely on initialization rather than the optimization process as a whole. This raises a question: can we be confident that StartGrad improves optimization across all optimizers, or might this only hold in specific cases?

3. Experimental Section

Results and Scope: Table 1 does not appear to provide results for StartGrad, which limits the insight we can gain into the method’s effectiveness. Additionally, the experimental section is underdeveloped. The current setup only uses ResNet-18 on a subset of 500 random ImageNet images, which restricts the findings’ generalizability in the vision domain. Including additional datasets, such as Pascal-Part and Monumai, where ground truth is available, would strengthen the empirical validation. It would also be interesting to test the techniques on more DNN such as Swin Transformer and ViT.
*Convergence Speed:* It would be valuable to evaluate whether the proposed initialization method enhances the convergence speed of XAI algorithms.
*Comparative Analysis:* Comparing StartGrad to other post-hoc explainability techniques would provide more context and relevance to the findings.

4. Metrics Used

The choice of metrics is unclear. A more detailed explanation of the metrics used in the evaluation would be beneficial. Additionally, considering established XAI metrics from frameworks like Quantus or Xplique could improve comparability and transparency.

[1] Lamas, Alberto, et al. "MonuMAI: Dataset, deep learning pipeline and citizen science based app for monumental heritage taxonomy and classification." Neurocomputing 420 (2021): 266-280.
[2] Chen, Xianjie, et al. "Detect what you can: Detecting and representing objects using holistic models and body parts." Proceedings of the IEEE conference on computer vision and pattern recognition. 2014.
[3] Selvaraju, R. R., Das, A., Vedantam, R., Cogswell, M., Parikh, D., & Batra, D. (2016). Grad-CAM: Why did you say that?. arXiv preprint arXiv:1611.07450.
Jamil, Md Shafayat, et al. "Advanced gradcam++: improved visual explanations of CNN decisions in diabetic retinopathy." Computer Vision and Image Analysis for Industry 4.0. Chapman and Hall/CRC, 2023. 64-75.
[4] Wang, Haofan, et al. "Score-CAM: Score-weighted visual explanations for convolutional neural networks." Proceedings of the IEEE/CVF conference on computer vision and pattern recognition workshops. 2020.
[5] Muhammad, Mohammed Bany, and Mohammed Yeasin. "Eigen-cam: Class activation map using principal components." 2020 international joint conference on neural networks (IJCNN). IEEE, 2020.
[6] Srinivas, Suraj, and François Fleuret. "Full-gradient representation for neural network visualization." Advances in neural information processing systems 32 (2019).
[7] Sattarzadeh, Sam, et al. "Integrated grad-cam: Sensitivity-aware visual explanation of deep convolutional networks via integrated gradient-based scoring." ICASSP 2021-2021 IEEE International Conference on Acoustics, Speech and Signal Processing (ICASSP). IEEE, 2021.
[8] Kapishnikov, Andrei, et al. "Guided integrated gradients: An adaptive path method for removing noise." Proceedings of the IEEE/CVF conference on computer vision and pattern recognition. 2021.

---

> ### Author Response · Authors · 2024-11-21
> **Response reviewer 1UMB: Typographical error and overclaims**
>
> Dear reviewer 1UMB,
>
> First of all, we would like to thank you very much for your thorough and insightful review. We sincerely appreciate the time you spent evaluating our work, and very much appreciate your positive comments about the writing of our paper, the clarity of our methodology, and the novelty of our approach and the overall relevance.
> We also sincerely appreciate your thoughtful questions and will address all your comments, questions, and concerns in detail.
>
> **Typographical Error**
>
> Thank you for your insightful comment. To clarify this, we have revised the corresponding section to better distinguish between the trivial solution (which minimizes distortion but does not achieve sparsity at all) and our proposed gradient-based heuristic. Specifically, we now explain that while setting $\Delta \mathbf{x} = 0$ (or equivalently $\mathbf{m} = 1$) minimizes distortion entirely, this approach fails to address the sparsity objective central to the RDE framework.
> To avoid confusion, we have relegated the discussion of the trivial solution to a footnote, emphasizing the practical strategy of using gradient information to balance distortion and sparsity in the main text. This revision highlights the motivation behind our heuristic while acknowledging the trivial solution for completeness. We believe this addresses the potential contradiction and improves the overall clarity of the section.
>
> **Weaknesses: Overclaims**
>
> Thank you for your insightful comments and questions. We go into these questions in detail here:
>
> Regarding the resemblance between the RDE objective in Eq. (2) and the IB objective in Eq.(4), we feel that there is a misunderstanding here. Both objectives contain two terms. Our argument is that:
>
> * The term min $E(\Phi_c(x), \Phi_c(\tilde{x}))$ is essentially equivalent to $max I(\tilde{x};\hat{y})$ as illustrated in Proposition 1.
> * Minimizing the second term in Eq.(4) $I(x;\tilde{x})$ can be seen as an implicit way to control the sparsity of the mask, i.e., $\|m\|_0$ or $\|m\|_1$ in Eq.(2). This is because, in our case, $\tilde{x}$ is a subset of x, so $I(x;\tilde{x})$ is approximately equal to the entropy $H(\tilde{x})$. Generally, for a random variable, the entropy tends to increase with dimensionality, as each additional dimension introduces more uncertainty to the joint distribution $p(\tilde{x})$.  So our argument is that both $I(x;\tilde{x})$ and $\|m\|$ add constraints on the mask in terms of sparsity. In fact, in real-world IB objective implementations, people also usually replace $I(x;\tilde{x}$) with $\|m\|$ [1, 2].
>
> **References**
>
> [1] Seojin Bang, Pengtao Xie, Heewook Lee, Wei Wu, and Eric Xing. Explaining a black-box by using
> a deep variational information bottleneck approach. In Proceedings of the AAAI conference on
> artificial intelligence, volume 35, pp. 11396–11404, 2021.
>
> [2] Ruo Yu Tao, Vincent Francois-Lavet, and Joelle Pineau. Novelty search in representational space for
> sample efficient exploration. Advances in Neural Information Processing Systems, 33:8114–8126,
> 2020.

---

> ### Author Response · Authors · 2024-11-21
> **Response reviewer 1UMB: Proposition 2 and 3, Clarification Table 1**
>
> **Question: Propositions 2 and 3 focus solely on initialization rather than the optimization process as a whole. This raises a question: can we be confident that StartGrad improves optimization across all optimizers, or might this only hold in specific cases?**
>
> We thank the reviewer for this thoughtful observation. You are correct that Propositions 2 and 3 focus exclusively on the initialization phase. As stated in lines 276–277, our theoretical analysis is limited to demonstrating that StartGrad provides a provably superior initialization by balancing the trade-off between prediction accuracy and the mask constraint compared to other initialization strategies. Thus, we respectfully note that there is no overstatement in our claims regarding this aspect.
>
> That said, we acknowledge that proving StartGrad’s superiority across the entire training process would require additional theoretical assumptions—such as good initialization accelerating convergence and increasing the likelihood of reaching global or high-quality local minima. While a rigorous theoretical proof remains an open challenge, we provide extensive empirical evidence to support the hypothesis in the current paper that a smart initialization enhances mask-based explanation methods. Specifically, our experiments span diverse mask-based explanation methods across both the vision domain and the time-series domain incorporating various architectures (VGG, ResNet, Recurrent neural networks). We also added transformer architectures in the revised version of our paper.
>
> Our results demonstrate that StartGrad not only offers theoretical advantages at initialization but also enhances performance across different mask-based explanation methods and modalities, particularly during early iteration steps. Moreover, StartGrad often contributes to strong overall performance improvements (e.g., as shown in the results for ShearletX). This indicates that StartGrad provides a robust initialization that yields significant benefits early in training without compromising overall performance. However, we do observe some variation in its effectiveness across different mask-based methods, which warrants further investigation.
>
> Finally, we note that extending our theoretical analysis to encompass the entire training process and generalizing StartGrad’s effectiveness across all methods and optimizers is indeed a non-trivial challenge. We leave this as an exciting avenue for future work. For this study, we align with prior research by using the commonly adopted Adam optimizer with hyperparameters chosen as reported in the original papers to ensure fair and consistent comparisons.
>
> **Weakness: Table 1 does not appear to provide results for StartGrad, which limits the insight we can gain into the method’s effectiveness.**
>
> We appreciate the reviewer’s observation and would like to clarify that Table 1 presents the effectiveness of StartGrad by quantifying its performance against alternative initialization methods (All-ones and Uniform). Specifically, for each mask-based explanation method (PixelMask, WaveletX, and ShearletX), we compare the same set of 500 randomly selected validation images from ImageNet, initializing the masks either with StartGrad or the alternative methods. The table reports the median pairwise performance difference between StartGrad and the baseline initializations for the CP-Pixel and CP-L1 metrics, measured across different iteration steps (50, 100, and 300). To ensure the statistical robustness of these comparisons, we use a one-sided Wilcoxon signed-rank test with Bonferroni correction. As it can be seen from the results, StartGrad provides measurable and significant advantages consistently at early iterations compared to alternative initialization strategies, but can also benefit performance even at the very late stage of optimization (iteration step 300 for  ShearletX). This can translate to substantial speedups as demonstrated by the results provided in Table 2.
>
> We would also point out that these gains come only minimal computational overhead as calculating the gradient and using the QTF are negligible compared to the overall optimization costs associated with running these mask-based methods (especially for the shearletX method) for 300 iterations.
>
> We hope this detailed explanation clarifies the relevance of Table 1 and its insights into StartGrad’s effectiveness.

---

> ### Author Response · Authors · 2024-11-21
> **Response reviewer 1UMB: Experimental scope, Convergence speed**
>
> **Weakness: Scope of the experiments**
>
> Thank you for your thorough and insightful comments regarding the scope of the experiments. In our initial version, we had already results included for the VGG 16 model, in addition to the ResNet18 model.  We added an additional sentence in the revised version of the paper to make this clear, as we acknowledge that one could have easily missed this in the first version of the paper.
>
> In the updated paper, we added results for the vision transformer and swin transformer, confirming that our findings hold across these architectures as well. The results can be found in Appendix F.3 and F.4. in the updated version of the paper.
>
> Regarding the dataset, we opted for ImageNet as it is the most widely used benchmark in the vision domain, allowing direct comparison with prior works. Specifically, methods like ShearletX [3] also employ subsets of ImageNet for their evaluations.
> As for the sample size, we used 500 images to ensure a balance between computational feasibility and statistical robustness. This sample size aligns with prior studies and significantly exceeds other comparable work, such as [4], which utilized only a subset of ‘only’ 100 samples. This choice ensures a meaningful empirical evaluation without sacrificing experimental thoroughness.
>
> In addition to the vision models and associated mask-based explanation methods tested (pixelMask, waveletX, shearletX), we also want to point out that we further evaluate StartGrad on a second data modality, i.e. time-series where we test our method on an recurrent neural network across two different datasets. Lastly, we provide extensive ablation studies on the vision domain that test the robustness of StartGrad regarding the gradient estimation (see Appendix F.).
>
> We also appreciate your suggestion to include datasets such as Pascal-Part and Monumai. We plan to incorporate these datasets in future work to further evaluate StartGrad framework across different datasets other than the most-commonly used ImageNet dataset.
>
> **Question:  It would be valuable to evaluate whether the proposed initialization method enhances the convergence speed of XAI algorithms.**
>
> Thank you for your thoughtful question regarding the convergence speed. We appreciate the opportunity to clarify this aspect of our work.
>
> For the vision experiments, as shown in the additional figures in Appendix D.3 and F.1.1 (for ResNet18 and VGG16), we observe that StartGrad does not significantly improve convergence speed. Specifically, the performance curves for StartGrad are shifted above those of the All-ones and Uniform initialization methods but plateau around iteration 300, indicating no faster convergence.
>
> However, in the time-series domain, we observe a different trend. StartGrad seems to enhance convergence speed, especially in the deletion game formulation of the ExtremalMask method (see Appendix E.1.1).
>
> As highlighted in the paper, even though StartGrad does not always accelerate convergence across all domains, it facilitates a more efficient path to achieving target metrics. This is particularly evident in the early iterations, where StartGrad leads to faster attainment of target performance. Table 2 demonstrates that, with StartGrad initialization, the number of steps required to reach the target metric is significantly reduced compared to the All-ones and Uniform initialization methods, translating to substantial speedups in practice
>
> We hope this explanation provides clarity on how StartGrad impacts convergence, particularly in terms of early performance gains and efficiency in reaching target metrics. Given the high computational costs of mask-based methods, StartGrad's ability to reduce iteration steps without compromising performance can lead to significant cost savings (in terms or reduced number of iteration steps), making it particularly beneficial for large-scale or time-sensitive applications.
>
> **References**
>
> [1] Stefan Kolek, Robert Windesheim, Hector Andrade Loarca, Gitta Kutyniok, and Ron Levie. Explaining image classifiers with multiscale directional image representation. In Proceedings of the IEEE Conference on Computer Vision and Pattern Recognition (CVPR). IEEE, 2023.
>
> [2] Stefan Kolek, Duc Anh Nguyen, Ron Levie, Joan Bruna, and Gitta Kutyniok. Cartoon explanations
> of image classifiers. In European Conference of Computer Vision (ECCV), 2022.

---

> ### Author Response · Authors · 2024-11-21
> **Response reviewer 1UMB: Comparative analysis, clarification metric choice**
>
> **Comparative analysis**
>
> We appreciate your suggestion to compare StartGrad to other post-hoc explainability techniques, and we want to clarify the specific focus of our work.
>
> The primary goal of this paper is to systematically study mask-initialization strategies for mask-based explanation methods from both a theoretical and empirical perspective. StartGrad is introduced as a novel initialization method specifically designed to enhance mask-based methods. Therefore, our comparisons are focused on commonly-used initialization techniques for these methods, as this aligns directly with our research objectives.
>
> While comparisons with saliency-based techniques such as IG or Grad-CAM are indeed valuable for a broader perspective, StartGrad is not applicable to these methods, which do not rely on mask initialization. That said, extensive comparisons between mask-based methods and alternative explainability approaches have been conducted in prior works, particularly those introducing state-of-the-art mask-based methods [1,2,3].
>
> However, for completeness and for comparison reasons we added the IG and Grad-CAM methods in the revised version of the paper in the Appendix C.2.2 as requested.
>
> **Weakness: Unclear choice of metric**
>
> We thank the reviewer for their insightful comments regarding our choice of evaluation metrics. Here, we provide a more detailed explanation of why we selected the conciseness-preciseness (CP) Pixel and L1 scores instead of the commonly-used insertion and deletion scores.
>
> Insertion and deletion scores are effective evaluation metrics for explanation methods that provide a clear ordering of feature importance. However, unlike other post-hoc explainability methods such as Integrated Gradients, the mask-based methods in our study do not inherently rank the relevance of coefficients and aim to learn a binary, sparse mask by optimizing an objective that approximates $L_{0}$-norm through a  $L_{1}$-norm. Since mask-based explanation methods do not inherently rank feature importance, applying insertion and deletion metrics, which rely on ordered rankings, becomes problematic. This limitation has been discussed in [1], who introduced CP-scores to address such challenges, and we align with their approach to enable meaningful comparisons of StartGrad with alternative initialization schemes.
>
> To illustrate this issue, we computed faithfulness scores (difference between insertion and deletion) for PixelMask, WaveletX, and ShearletX across iterations and found that faithfulness scores consistently decrease during optimization. This counterintuitive behavior highlights the unsuitability of the faithfulness metric for mask-based methods due to their lack of inherently ordering mask values.
>
> In the revised version of our paper, we clarified our rationale by expanding the justification for the CP scores in the main paper, and by adding supplementary explanations and figures in Appendix C.2.2.
>
> **Final words**
>
> We apologize for addressing your comments and questions across multiple responses. Our goal was to provide thorough and detailed answers to your thoughtful and constructive suggestions.
>
> We hope these responses have clarified your questions and concerns. Please feel free to reach out if you have any follow-up or additional questions—we’d be happy to address them.
>
> Best regards,
>
> The Authors
>
> **References**
>
> [1] Stefan Kolek, Robert Windesheim, Hector Andrade Loarca, Gitta Kutyniok, and Ron Levie. Explaining image classifiers with multiscale directional image representation. In Proceedings of the IEEE Conference on Computer Vision and Pattern Recognition (CVPR). IEEE, 2023.
>
> [2] Stefan Kolek, Duc Anh Nguyen, Ron Levie, Joan Bruna, and Gitta Kutyniok. Cartoon explanations of image classifiers. In European Conference of Computer Vision (ECCV), 2022.
>
> [3] Joseph Enguehard. Learning perturbations to explain time series predictions. In Proceedings of
> the 40th International Conference on Machine Learning, pp. 9329–9342. PMLR, 2023.

---

> > ### Comment · Reviewer_1UMB · 2024-11-23
> >
> > Thank you to the authors for providing a detailed response to my comments.
> >
> > That said, I found the rebuttal somewhat difficult to follow. It would have been very helpful if the authors had submitted a revised version of the paper with the modifications clearly highlighted in color. This would make it easier to identify the changes. Additionally, the extensive use of bold text in the rebuttal made it harder to read, as there appear to be almost as many bolded words as non-bolded ones. I recommend using bolding more **sparingly for emphasis.**
> >
> > I also noticed that the authors provided their responses quite late in the rebuttal period, which complicates the discussion process.
> >
> > That being said, I appreciate that the authors took the time to respond. Thank you for including additional experiments and clarifications on the metrics. Many of the explanations were helpful and addressed some of my concerns.
> >
> > However, I remain **unconvinced about the issue of overclaims**. Specifically, Proposition 1 is still unclear to me. Does this equivalence always hold? If so, why do researchers typically optimize
> > I(\hat{y}; \tilde{x})? What exactly do you mean by "equivalent"? Are there bounds associated with this equivalence? If so, why not include the upper and lower bounds in the paper to provide a clearer understanding?
> >
> > I kindly request that the authors **submit a revised version** of the paper with all modifications highlighted in color. This would be greatly beneficial for the review process.
> >
> > I will take also some time to read the other reviews and answers. (I did not have time to do it yet).
> >
> > Thank you for your efforts.

---

> > > ### Author Response · Authors · 2024-11-25
> > > **Response reviewer 1UMB: Clarification Proposition 1**
> > >
> > > Dear reviewer,
> > >
> > > Thank you for your thoughtful comments and for taking the time to carefully review our rebuttal. We sincerely apologize if our initial response was difficult to follow due to the formatting issues, including the extensive use of bold text. This was not our intention, and we regret any inconvenience it caused.
> > >
> > > To address this, we have removed bold text from the rebuttal. Furthermore, we will upload a revised version later today where all modifications are clearly highlighted in red, given that we want to incorporate already the suggestion by reviewer yTDi. We will then inform you immediately. However, we want to already take the time to answer to your questions thoroughly.
> > >
> > > **Clarification Proposition 1**
> > >
> > > We thank you for your thoughtful question and we appreciate the opportunity to clarify it. Below, we aim to address your concerns step-by-step.
> > >
> > > ### Background on RDE and IB
> > >
> > > In the background section 3, we discuss two common objective approaches, i.e. RDE and IB principle, that both have the goal to find a sparse mask $m \in [0, 1]^{d}$ that minimizes the performance drop from the resulting masked input.
> > >
> > > In section 3.1, we first introduce and discuss the RDE objective that aims to find a sparse mask $m \in [0, 1]^{d}$:
> > >
> > > \begin{align}
> > >     \min_{\textbf{m} \in [0, 1]^{d}} E_{\textbf{u} \sim V}\left[\mathcal{D}\left(\Phi_{c}(\mathbf{\tilde{x}}), \Phi_{c}(\mathbf{x}\right))\right] + \lambda \|\mathbf{m}\|_{1},
> > > \end{align}
> > >
> > > where $\mathcal{D}\left(\Phi_{c}(\mathbf{\tilde{x}}), \Phi_{c}(\mathbf{x}\right))$ measures the distortion between the original prediction and the prediction with the masked input using the distortion function $\mathcal{D}: [0, 1]^{c} \times [0, 1]^{c} \rightarrow \mathbb{R}_{+}$.
> > >
> > > In section 3.2 we introduce then the Information Bottleneck principle, that defines the objective function in the following manner:
> > >
> > > \begin{align}
> > >     \min_{\textbf{m} \in [0, 1]^{d}} -I(\hat{y}; \tilde{\mathbf{x}}) + \beta I(\mathbf{x}; \tilde{\mathbf{x}}),
> > > \end{align}
> > >
> > > ### Proposition 1
> > > The purpose of proposition 1 is then to show the correspondence between the RDE and IB principle, i.e. builds a bridge between the two.
> > >
> > > In particular, Proposition 1 states, that if we choose to use the KL-divergence as a choice for the distortion function $\mathcal{D}$ to measure to measure the distortion with the KL-divergence between the original prediction $\Phi_c(\mathbf{x})$ and the masked prediction $\Phi_c(\tilde{\mathbf{x}})$, then we can show (proof in the Appendix) that the following holds:
> > >
> > > \begin{align}
> > >     \min_{\textbf{m} \in [0, 1]^{d}} -I(\hat{y}; \tilde{\mathbf{x}}) = \max_{\textbf{m} \in [0, 1]^{d}} I(\hat{y}; \tilde{\mathbf{x}})
> > >     = \min_{\textbf{m} \in [0, 1]^{d}} E_{\textbf{u} \sim V}\left[\mathcal{D_{KL}}\left(\Phi_{c}(\mathbf{\tilde{x}}), \Phi_{c}(\mathbf{x}\right))\right]
> > > \end{align}
> > >
> > > Therefore, Proposition 1 demonstrates that if the distortion function in the RDE objective is defined as the KL-divergence, the distortion part of the RDE objective maximizes the mutal information between the masked input and the original prediction of the information bottleneck principle. This is what we meant with maximizing mutual information is equivalent to minimizing the expected Kullback–Leibler (KL) divergence $\mathbb{E} \left( D_{\text{KL}} (\Phi_c(\tilde{\mathbf{x}});\Phi_c(\mathbf{x})) \right)$.
> > >
> > > ### Bound with $L_{1}$ Norm
> > >
> > > To further support this equivalence, in remark 1 we include a bound that links the KL divergence and the $L_{1}$ norm using the Pinsker inequality, i.e.:
> > >
> > > \begin{align}
> > > D_{\text{KL}} (\Phi_c(\tilde{\mathbf{x}});\Phi_c(\mathbf{x})) \geq \frac{1}{2\log2} \| \Phi_c(\tilde{\mathbf{x}}) - \Phi_c(\mathbf{x}) \|_1^2
> > > \end{align}
> > >
> > > which further shows that if we take the $L_{1}$ norm as a distortion measure, there is still a correspondence between the mutual information $-I(\hat{y}; \tilde{\mathbf{x}})$ of the IB principle and the distortion part of the RDE framework, albeit as a bound rather than an exact equivalence.
> > >
> > > We hope that this explanation addresses your concerns about Proposition 1 and the claims in our paper.

---

> > > > ### Author Response · Authors · 2024-11-25
> > > > **Response reviewer 1UMB: Clarification Remark 1**
> > > >
> > > > We also want to take the time to further clarify Remark 1 and also clarify our previous answer regarding the sparsity term that is included in both, the RDE objective as well as the IB principle.
> > > >
> > > > ### Clarification Remark 1: Sparsity and $I(\mathbf{x}; \tilde{\mathbf{x}})$
> > > >
> > > > However, both the RDE framework as well as the IB principle, still have their sparsity term, i.e. $\|\mathbf{m}\|_{1}$  and
> > > > $I(\mathbf{x}; \tilde{\mathbf{x}})$ respectively.
> > > >
> > > > In particular, the former sparsity term is a continuous relaxation of the $L_{0}$ norm for optimization purposes within the RDE framework as mentioned in the paper in the background section 3.1.
> > > >
> > > > To illustrate the next point, it is useful to rewrite the mutual information in the following manner:
> > > > \begin{align}
> > > > I(\mathbf{x}; \tilde{\mathbf{x}}) = H(\mathbf{x}) + H(\tilde{\mathbf{x}}) - H(\mathbf{x}, \tilde{\mathbf{x}})
> > > > \end{align}
> > > >
> > > > where $H()$ denotes the entropy. When optimizing for a sparse mask $\textbf{m} \in [0, 1]^{d}$, $H(\mathbf{x})$ remains constant, so we can drop it from the optimization objetive, i.e.
> > > >
> > > > \begin{align}
> > > > I(\mathbf{x}; \tilde{\mathbf{x}}) = H(\tilde{\mathbf{x}}) - H(\mathbf{x}, \tilde{\mathbf{x}})
> > > > \end{align}
> > > >
> > > > where $H(\tilde{\mathbf{x}})$ measures the entropy  of the masked input, and $H(\mathbf{x}, \tilde{\mathbf{x}})$ the joint entropy. If we use the definition of the conditional entropy and rearrange terms, i.e:
> > > >
> > > > \begin{align}
> > > >     H(\mathbf{x}, \tilde{\mathbf{x}}) = H(\tilde{\mathbf{x}}|\mathbf{x}) + H(\mathbf{x})
> > > > \end{align}
> > > >
> > > > Thus, we can again drop $H(\mathbf{x})$ and get finally
> > > >
> > > > \begin{align}
> > > > I(\mathbf{x}; \tilde{\mathbf{x}}) &= H(\tilde{\mathbf{x}}) - H(\tilde{\mathbf{x}}|\mathbf{x}) \approxeq H(\tilde{\mathbf{x}})
> > > > \end{align}
> > > >
> > > > where we made use of the observation that $\tilde{\mathbf{x}}$ is a subset of $\mathbf{x}$, thus the conditional entropy $H(\tilde{\mathbf{x}}|\mathbf{x})$ is typically small [1, 2].
> > > >
> > > > Building on this, increasing input dimensionality (high $L_{0}$ norm, approximated in RDE with high $L_{1}$ norm) typically raises entropy, as each added dimensionality introduces more uncertainty [3, 4]. Thus, the sparsity terms in the RDE framework and IB principle align conceptually, though not exactly as with the distortion term. However, the sparsity connection not part of Proposition 1, but is mentioned in Remark 1.
> > > >
> > > > Lastly, from our observation, the IB objective is commonly employed for in-built explainability, whereas the RDE objective is more used for post-hoc explainability which is the focus of our work.
> > > >
> > > > We hope that this explanation, together with our answer above (clarification Proposition 1) addresses your concerns about Proposition 1 and the claims in our paper. We acknowledge that the explanation might have come short in the initial version of the paper, and we will clarify this further in the revised version which will be updated later today as mentioned above. If you have any further question, please let us know.
> > > >
> > > > Thank you again for your detailed feedback and for helping us improve our work. We really appreciate your suggestions and comments.
> > > >
> > > > Best regards,
> > > >
> > > > The authors
> > > >
> > > > **References**
> > > >
> > > > [1] DJ Strouse and David J Schwab. The deterministic information bottleneck. Neural computation, 29
> > > > (6):1611–1630, 2017.
> > > >
> > > > [2] Andreas Kirsch, Clare Lyle, and Yarin Gal. Unpacking information bottlenecks: Unifying
> > > > information-theoretic objectives in deep learning. arXiv preprint arXiv:2003.12537, 2020.
> > > >
> > > > [3] Seojin Bang, Pengtao Xie, Heewook Lee, Wei Wu, and Eric Xing. Explaining a black-box by using a deep variational information bottleneck approach. In Proceedings of the AAAI conference on artificial intelligence, volume 35, pp. 11396–11404, 2021.
> > > >
> > > > [4] Ruo Yu Tao, Vincent Francois-Lavet, and Joelle Pineau. Novelty search in representational space for sample efficient exploration. Advances in Neural Information Processing Systems, 33:8114–8126, 2020.

---

> ### Author Response · Authors · 2024-11-25
> **Response reviewer 1UMB: Revised version of our paper is uploaded**
>
> **Dear Reviewer 1UMB,**
>
> We are pleased to inform you that we have uploaded a revised version of our paper where all modifications are clearly highlighted in red. This version incorporates all of your proposed suggestions as well as the feedback by the other reviewers. As suggested, we clarified Proposition 1 and included additional steps in the derivation of the first proof, now detailed in Appendix A.
>
> We really appreciate your patience and engagement throughout this process. Your feedback and suggestions have helped us to improve our work for which we are grateful.
>
> Please let us know if you have any further suggestions or questions.
>
> Best regards,
>
> The Authors

---

> > ### Comment · Reviewer_1UMB · 2024-11-27
> >
> > **Dear Authors,**
> >
> > Thank you very much for your detailed response. I appreciate the effort you put into addressing my question. However, I believe I may not have been entirely clear in my initial query.
> >
> > What I find slightly confusing is the statement: "Specifically, we provide formal proofs that StartGrad is provably superior at initialization in balancing the aforementioned tradeoff compared to other mask initialization strategies." My concern is that the proof is based solely on the norm
> > $p$. I wonder, since when is a norm
> > $ p\geq1$ considered a distortion? This was link to main concern regarding the overclaim.
> >
> > Additionally, I am curious about the implications if
> >  $p=0$ or $p=1$ .
> >
> > Finally, regarding Equation (5) and the associated proof, I noticed that there is no mention of
> > $\\circ(\\Delta x)$. Could you clarify why this was omitted? (also in the proof)
> >
> > Thank you again for your time and insights!
> >
> > Best regards,

---

> ### Author Response · Authors · 2024-11-29
> **Response reviewer 1UMB: Clarification norm and distortion and justification for Lp norm**
>
> Dear reviewer 1UMB,
>
> Thank you for clarifying further your concerns. We nevertheless hope that our previous answer was still helpful in clarifying further proposition 1. We sincerely appreciate your time dedicating to our review and will address your concerns in detail:
>
> **Clarification statement/wording**
>
> We revised the wording of the statement that you cited in the revised version of the paper to:
>
> "Specifically, we provide formal proofs that StartGrad is provably
> superior at initialization in balancing the trade-off between distortion and sparsity, **as formalized in the RDE framework**, when compared to other mask initialization strategies."
>
> I hope this makes the wording a bit clearer to the reader.
>
> **Clarification norm and distortion**
>
> We want to clarify that the $L_{p}$ norm is not considered distortion. We define the distortion in the context of our work (and in line with the previous line of research) as quantification of the difference between the original prediction $\Phi_{c}(\mathbf{x})$ we want to explain and the masked prediction (the input that is masked to only retain the most important input features) $\Phi_{c}(\mathbf{\tilde{x}})$.
>
> Hence, we refer to the distortion as the gap between the former two, i.e. $\Phi_{c}(\mathbf{x})$ and $(\mathbf{\tilde{x}})$.
> To quantify the gap between the two, one uses a distortion function, i.e.:
>
> $\mathcal{D} : [0, 1]^c \times [0, 1]^c \longrightarrow \mathbb{R}_+$.
>
> Hence, the $L_{p}$ norm is one particular choice of a suitable distortion function specification (where one of course needs to decide on p). However, one could also alternatively use the KL divergence as a particular choice of the distortion function $\mathcal{D}$. We show in Proposition 1 the connections between choosing the KL divergence and the information bottleneck principle and also that the KL divergence is an upper bound to the $L_{1}$ norm, i.e. there is a relationship between the two.
>
> We hope this clarifies both the distortion, distortion function and $L_{p}$ norm as one particular choice of suitable distortion function.
>
> We also want to address the concern that the proof is based on the $L_{p}$ norm and about the implications of setting p=0 and p=1.
>
> Indeed, in our proof we focus on the choice of $L_{p}$ norm with $p \geq 1$ (which I will comment below) as a choice of for the distortion function $\mathcal{D}$. We do this for the following reasons:
>
> - The RDE framework as introduced in [1] also use the $L_{p}$ norm, which is why we also stick to this choice. As mentioned above, we also changed the wording of the paper slightly, to make it also clearer.
> - Even though one could also use the KL divergence as a distortion measure as pointed above, this choice does not seem to affect the performance empirically, as reported in [2]. Furtheremore, all mask-based explanation methods used in this paper (vision and time-series) use the $L_{2}$ norm which further justifies our choice for focusing on the $L_{p}$ norm.
> - As stated in Proposition 1, the $L_{1}$ norm can be related to to the KL divergence using Pinskers inequality, hence even though we focus on the $L_{p}$ norm with $p \geq 1$, there is still a connection to the KL divergence.
>
> **References**
>
> [1] Jan Macdonald, Stephan W¨aldchen, Sascha Hauch, and Gitta Kutyniok. A rate-distortion framework for explaining neural network decisions, 2019.
>
> [2] Stefan Kolek, Duc Anh Nguyen, Ron Levie, Joan Bruna, and Gitta Kutyniok. Cartoon explanations of image classifiers. In European Conference of Computer Vision (ECCV), 2022.

---

> > ### Comment · Reviewer_1UMB · 2024-11-30
> >
> > Thank you very much for your response!
> >
> > Regarding my previous question about the Taylor expansion, why not applying the **Mean Value Theorem** ? Doing so could ensure the correctness of your proof and eliminate the need for approximations...
> >
> > Could you clarify why you chose not to use it? Additionally, could you verify if adopting this approach would indeed remove the need for approximations throughout?

---

> ### Author Response · Authors · 2024-11-29
> **Respones reviewer 1UMB: Implications p=0 and p=1**
>
> Dear reviewer 1UMB,
>
> #### Implications of p=0 and p=1
>
> Regarding the implication of $p=0$ and $p=1$. This is an important question and we appreciate the opportunity to clarify this in more detail.
>
> Setting p=0 is suitable in situations where one wants to enforce sparsity, as it induces a penalty/loss of 1 for non-zero elements. Hence, this is a desirable choice for the sparsity constraint on the mask as stated in the paper in equation 1.
>
> However, the choice of p=0, i.e. $L_{0}$-norm as a choice for the distortion function $\mathcal{D}$ is not an effective choice for two reasons:
>
> - First and foremost, the $L_{0}$ is not differentiable which is a significant problem, as the mask-based approaches used in our approach are all based on learning a sparse mask using a gradient-based (via Adam) approach. Furthermore, exact $L_{0}$ regularization is computationally expensive and intractable in high dimensions [1]. This is also the reason, why one does not use the $L_{0}$ norm for the sparsity term. Given the fact that the $L_{0}$ has its merits in imposing sparsity, there are works that try to find a continous approximation [3].
> - Apart from non-diffentiability of the $L_{0}$ norm, there is also a more subtle problem with the $L_{0}$ as a choice for the distortion function. If we were to use the $L_{0}$ as a choice for the distortion function (again, it is not differentiable which excludes it from the get-go), it would fail measure the distortion on a granular level, as it would simply penalize the any deviation from the original prediction, i.e. $\Phi_{c}(\mathbf{x})$ that is induced due to the masking with 1, regardless how big the gap in practice really is. To put simply, if $\Phi_{c}(\mathbf{x}) = 0.9$ and we had two suitable candidate masks that induce some distortion, $\Phi_{c}(\mathbf{\tilde{x_{1}}}) = 0.89$ and $\Phi_{c}(\mathbf{\tilde{x_{2}}}) = 0.75$, both options would lead to a distortion loss of 1, which does not capture the fact that the first masked prediction, i.e. $\Phi_{c}(\mathbf{\tilde{x_{1}}}) = 0.89$ is much closer to the original prediction which is 0.9 in this hypothetical case.
>
> For these two reasons, our proofs are restricted to the case of $L_{p}$ with $p \geq 1$.
>
> We want to also clarify the role of $p=1$ and point out some subleties. It is important to note that it p=1 is a suitable choice for the $L_{p}$ norm.
>
> However, again there is subtle meaning of using $p=1$ as opposed to for instance $p=2$. The latter penalizes any given deviation induced due to masking less. For instance, if we take again the same setting as above and use take the candidate 1, i.e. $\Phi_{c}(\mathbf{\tilde{x_{1}}}) = 0.89$, we would get a distortion of 0.01 in case of $p=1$ and a distortion of 0.0001 for $p=2$. This is due to the fact that we operate with softmax probabilities $\in [0,1]$ which is common for image classification tasks.
>
> Interestingly, this observation has subtle implications as reflected in the remark 2 of proposition 3, where we state that StartGrad is superior at initialization compared to the all-ones (in the case that we want to explain a class-specific classification, i.e. c=1) whenever this holds:
>
> \begin{align*}
>     \frac{2^{\frac{1}{p}}}{d} & \leq \lambda
> \end{align*}
>
> In particular, the numerator of the left-hand side of the equation shows that for p=2 StartGrad already becomes superior to all-ones initialization for a choice of lambda $\lambda d \geq \sqrt{2}$ as opposed to the choice for p=1 where we have the condition $\lambda d \geq 2$. In essence, by choosing p=1 we implicitly put more weight on distortion as opposed to $p=2$, as we penalize distortion more severely for a given deviation, i.e. the distortion part has less of an impact in the distortion-sparsity trade-off for $p=2$ as opposed to $p=1$.
>
> We hope this clarifies all of your answers regarding the the $L_{p}$ norm, distortion and distortion function.
>
> Best regards,
>
> The authors
>
> **References**
>
> [1] Yutaro Yamada, Ofir Lindenbaum, Sahand Negahban, and Yuval Kluger. Feature selection using stochastic gates. In ICML, pp. 10648–10659, 2020

---

> ### Author Response · Authors · 2024-11-29
> **Response reviewer 1UMB: Delta and further experiment**
>
> **Clarification $\Delta \mathbf{x}$**
>
> Thank you for asking some clarification question. As stated in the main paper, we define $\Delta \mathbf{x} = \mathbf{\tilde{x}} - \mathbf{x}$ representing distortion (as in, the change compared to the original input) induced due to the mask $\mathbf{m}$. In the proof A.2 in the Appendix, we use the notations interchangably which we agree could cause some confusion which we are sorry for. We will make this clearer in the final version of the paper. However, at this point we cannot change the pdf any further, but we thank the reviewer for pointing this out.
>
> **Additional experiments / insights**
>
> We want to also point out that we have added further clarifications and supporting results in the revised paper, including additional experiments and figures (see Appendix F.6, F.7, F.8). These results illustrate how gradient-informed initialization as done with our novel algorithm StartGrad provides a crucial early advantage by guiding optimization toward regions with minimal distortion. This early benefit proves critical, as uniform initialization, lacking gradient-based guidance, fails to achieve comparable distortion reduction, even by the final iteration. These findings highlight the practical significance of initialization strategies and directly connect initialization to final mask performance.
>
> **Final words**
>
> We apologize for addressing your comments and questions across multiple responses. Our goal was to provide thorough and detailed answers to your question.
>
> We hope these responses have clarified your questions and concerns. Please feel free to reach out if you have any follow-up or additional questions—we’d be happy to address them.
>
> Best regards,
>
> The authors

---

> ### Author Response · Authors · 2024-12-01
> **Response reviewer 1UMB: Mean Value Theorem and Taylor expansion**
>
> Dear reviewer 1UMB,
>
> Thank you for your thoughtful question and the opportunity to clarify our decision to use the Taylor expansion rather than the Mean Value Theorem (MVT) in our proof. Below, we address your question in detail.
>
> ---
> ### The advantage of the Taylor expansion
>
> The Taylor expansion provides a local, first-order approximation of the distortion term $D(\Phi_c(\tilde{x}), \Phi_c(x))$ by expanding $\Phi_c(\tilde{x})$ around $\mathbf{x}$, i.e. the instance that we want to explain via a mask-based explanation method. Specifically, we use the first-order approximation:
>
> $\Phi_c(\tilde{x}) \approx \Phi_c(x) + \nabla_x \Phi_c(x) \cdot (\tilde{x} - x)$
>
> which leads to the approximation:
>
> $D(\Phi_c(\tilde{x}), \Phi_c(x)) \approx \|\nabla_x \Phi_c(x) \cdot (\tilde{x} - x)\|_p$
>
> when we use a $L_{p}$-norm as choice for the distortion function $\mathcal{D}$.
>
> The advantage of this formulation is that it directly links the distortion induced by our mask-based explanation approach to the gradient $\nabla_x \Phi_c(x)$ which is readily available during optimization and central to our algorithm StartGrad. available and which we make us of in our proposed StartGrad algorithm. The Taylor expansion therefore, naturally aligns with our framework's goal of leveraging gradient signals to guide mask initialization.
>
> While Taylor expansion introduces a linear approximation, our extensive experiments across domains (vision, time-series) and architectures (e.g., ResNet18, ViT, GRU) show that this does not compromise StartGrad's performance.
>
> ---
> ### The use of Mean Value Theorem (MVT) and its complications
>
> The mean value theorem (MVT) states that for a continous and differentiable function $\Phi_{c}(\mathbf{x})$, there exists some $\xi \in [\mathbf{x}, \mathbf{\tilde{x}}]$ such that:
>
> $\Phi_c(\tilde{x}) - \Phi_c(x) = \nabla_x \Phi_c(\xi) \cdot (\tilde{x} - x)$
>
> which again, we can leverage to get the following expression:
>
> $D(\Phi_c(\tilde{x}), \Phi_c(x)) = \|\nabla_x \Phi_c(x) \cdot (\tilde{x} - x)\|_p$
>
> where again, we use $L_{p}$-norm as choice for the distortion function $\mathcal{D}$. As we can see, the MVT could be used to represent the distortion term $D(\Phi_c(\tilde{x}), \Phi_c(x))$ without the need of an approximation. While we got rid of the approximation compared to the Taylor expansion from above, some critical obstacles were introduced which we outline below:
>
> 1. **Dependence on the intermediate point $\xi$:**
>    The MVT guarantees the existence of $\xi \in [x, \tilde{x}]$ but does not specify its exact location. Consequently, $\nabla_x \Phi_c(\xi)$ cannot be explicitly computed without additional assumptions. This introduces uncertainty into the distortion term and makes it less practical for our analysis, as we cannot directly leverage $\nabla_x \Phi_c(\xi)$ in our framework for gradient-based mask initialization.
>
> 2. **Practicality in Mask Optimization:**
>    In contrast to the MVT, the Taylor expansion directly uses the gradient $\nabla_x \Phi_c(x)$, which is readily available during optimization. This gradient explicitly guides the design of the gradient-based mask $m_{\text{grad}}$ via our proposed algorithm StartGrad. In particular, in our proof of StartGrad vs. uniform, we make use of the fact that StartGrads algorithm is explicitly constructed to minimize the distortion at initialization, which is guided by the gradient $\nabla \Phi_c({\mathbf{x}})$. However, if we were to use the MVT, we would only know that $\xi$ exists, yet, not its exact location, which would complicate the analysis further.
>
> We want to also point out that in the proof of StartGrad vs. all-ones we do not need to use the first-order Taylor expansion. Hence, the MVT would potentially only be needed in our proof StartGrad vs. uniform.
>
> ---
>
> ### Conclusion
>
> While the MVT avoids approximations, its reliance on the unknown intermediate point $\xi$ introduces significant practical challenges that diminish its applicability within our framework. The Taylor expansion, by contrast, offers a direct and effective means to incorporate gradient information that is readily available during optimization into our analysis and StartGrad's design.
>
> Empirically, despite the approximation introduced by the Taylor expansion, our extensive experiments demonstrate that StartGrad achieves robust and reliable performance across multiple domains and model architectures.
>
> We hope this clarification resolves your concerns fully and demonstrates and strengthen the soundness of our theoretical analysis. Thank you again for your thoughtful question which gave us the opportunity to further justify our choice of the Taylor expansion that was used in our proof and in our paper.
>
> Best regards,
>
> The authors

---

> > ### Author Response · Authors · 2024-12-02
> > **Response reviewer 1UMB: Follow-Up on Reviewer Feedback**
> >
> > Dear reviewer 1UMB,
> >
> > We sincerely hope that our responses have thoroughly addressed all your concerns and questions.
> >
> > As the rebuttal deadline approaches tomorrow, we wanted to kindly follow up to ensure that our clarifications meet your expectations. If you have any remaining questions or need further clarifications, we would be more than happy to assist.
> >
> > Once again, thank you for your thoughtful feedback and the time and effort you have invested in reviewing our work.
> >
> > Best regards,
> >
> > The Authors

---

### Official Review · Reviewer_yTDi · 2024-11-03

**Soundness:** 2
**Presentation:** 3
**Contribution:** 2
**Rating:** 6
**Confidence:** 3

**Summary:**

This paper introduced a new gradient-based mask initialisation technique for mask-based explanation methods called StartGrad. This initialisation method trades-off mask performance against complexity of the resulting masked representation. The authors have completed a number of experiments that demonstrate StartGrad's ability to improve the performance of mask-based methods based on speeding up their optimisation process.

**Strengths:**

1. The theory and proofs in this paper are more than adequate, demonstrating the feasibility of balancing between complexity and information bottlenecks through information theory.

2. The paper has sufficient context, motivation. The authors have a solid understanding of the trade-off in mask-based XAI.

**Weaknesses:**

1.  The **experiments** in this paper are insufficient. the authors have only done experiments with $\lambda$ = 1,2 and 1,10. Since this hyperparameter directly affects the balance of trade-off, it is better to provide sufficient ablation study on lambda.
   It would be useful for the authors to mention the impact of different orders of magnitude of $\lambda$ on network performance. For example, a graph with $log_\lambda$ on the x-axis and performance on the y-axis could be completed to evaluate the impact of lambda.

2. The contribution of this paper does not seem obvious enough. In some metrics, the method proposed by the authors decreases in comparison to baseline in the early iterations, while it is essentially flat in the later period (as shown in Table 3). It is quite doubtful that StartGrad is able to accelerate with improving performance as the authors claimed.
    For example,  in table 3,  StartGrad has less AUP than all ones strategy at 50 iteration steps and less AUR at 100 iteration steps. The 50 steps AUR has even better performence than 100 steps.

**Questions:**

This initialisation method seems to be sensitive to $\lambda$. Does the choice of $\lambda$ greatly affect the performance and speed of StartGrad? And how do the choice of $\lambda_1$ and $\lambda_2$ mentioned in the appendix affect those performances?
    The authors are advised to add sensitivity analysis on these hyperparameters. In addition, for $\lambda_1$ and $\lambda_2$, it is better for the experiment to show the impact of them separately. For example, analysis a trade-off or a similar effect on performance between them.

---

> ### Author Response · Authors · 2024-11-21
> **Response reviewer yTDi: Experiments and lambda hyperparameter**
>
> Dear yTDi,
>
> First of all, we would like to thank you very much for your thorough and insightful review. We sincerely appreciate the time you spent evaluating our work, and will address all your comments, questions, and concerns in detail.
>
> **Weakness: Insufficient experiments and comments related to the role of the lambda hyperparameter**
>
> Thank you for raising this important point regarding the role of the lambda parameter. We agree that $\lambda$ significantly influences the trade-off between sparsity and distortion, and we appreciate the opportunity to clarify this aspect.
>
> The theoretical propositions 2 and 3 in the paper show that StartGrad is superior to both uniform and all-ones initializations at initialization. Specifically, Proposition 2 proves that StartGrad outperforms uniform initialization at initialization for any choice of lambda, while Proposition 3 demonstrates that StartGrad outperforms the all-ones initialization in conditions commonly encountered in XAI applications, such as high-dimensional settings which require sparse explanations.
>
> However, it is important to note that if $\lambda = 0$ - indicating no preference for sparsity - the optimization process becomes trivial. In this case the all-ones mask becomes optimal, rendering the mask-based optimization redundant.
>
> While we have not conducted an extensive ablation study across all lambda values, we focused on the $\lambda$ values recommended and used by the authors of the mask-based explanation methods incorporated in our paper. These choices, validated in prior work, ensure fair comparisons and meaningful evaluation. For example, [1] identified  $\lambda_{1}$ = 1, $\lambda_{2}$ = 1 as optimal for the Extremal model, while [2] recommended $\lambda_{1}$ = 1 and  $\lambda_{2}$ = 2 for ShearletX and $\lambda_{1} = 1$ and $\lambda_{2}$ = 10 for the WaveletX.
>
> The use of these recommended lambda values therefore ensures a consistent and meaningful empirical evaluation of StartGrad when applied to state-of-the-art methods, as choices of the lambda values aligns directly with the original authors' recommendations for their respective methods and the exact setup used when introducing their methods in the first place. Furthermore, using these values for the hyperparameter $\lambda$ aligns our experiments with the original methods’ setups, ensuring consistency to prior work.
>
> That being said, we recognize the importance of further understanding the impact of $\lambda$ on the performance of StartGrad. To further address your concern we are currently running a small set of experiments  to explore the sensitivity of StartGrad to different lambda values, including variations across orders of magnitude. We will update the paper with these results in an updated version of the paper to provide deeper insights. We want to also emphasize that, given the extensive number of model architectures tested (VGG16, ResNet18, Vision Transformer, and Swin Transformer) and mask-based explanation methods (PixelMask, WaveletX, ShearletX), the computational cost of a thorough sensitivity study of $\lambda$ would be prohibitive. Each combination of $\lambda$ parameters would need to be run three times for every initialization method (StartGrad, all-ones, uniform), across all explanation methods (PixelMask, WaveletX, ShearletX, ExtremalMask) and all architectures tested.
>
> Lastly, we want to emphasize that we conducted extensive experiments to evaluate StartGrad across different modalities (vision and time-series domain) for different mask-based state-of-the-art methods (pixelMask, WaveletX, ShearletX, ExtremalMask) for various model architectures (VGG, ResNet, ViT, Swin Transformer, GRU) and further provide extensive ablation studies to test the robustness of the performance of StartGrad. We believe that this demonstrates our thoroughness of our experimental analysis and we hope this addresses your concern of insufficient experiments.
>
> We hope we could clarify with our answer not only your question, but also addressed the first concern you raised associated with the experiments conducted in our paper.
>
> Please let us know if you have any follow-up / additional questions.
>
> Best regards,
>
> The authors
>
> **References**
>
> [1] Joseph Enguehard. Learning perturbations to explain time series predictions. In Proceedings of the 40th International Conference on Machine Learning, pp. 9329–9342. PMLR, 2023.
>
> [2] Stefan Kolek, Duc Anh Nguyen, Ron Levie, Joan Bruna, and Gitta Kutyniok. Cartoon explanations of image classifiers. In European Conference of Computer Vision (ECCV), 2022.

---

> ### Author Response · Authors · 2024-11-21
> **Response reviewer yTDi: Clarity results and contribution of the paper**
>
> **Weakness: Clarity results and contribution of the paper**
>
> Thank you for your thoughtful comment. We would like to clarify the observation in table 3.
>
> At iteration step 50, StartGrad outperforms the baseline initialization method (uniform which was originally used by the authors of ExtremalMask [1]) in 4/4 metrics, and outperforms all-ones initialization in ¾ methods with the exception of AUP where the all-ones initialization reaches a higher value. However, we want to point out that the all-ones initialization reaches a higher AUP value, but a substantially lower AUR value (0.696 compared to StartGrad AUR 0.805) which indicates that the all-ones initialization misses a lot of important features.  Furthermore, we would like to point out that StartGrad leads to notable and substantial performance improvements in AUR, Information and Entropy compared to all-ones and uniform initialization.
>
> At later iteration (100) StartGrad maintain its advantage compared to uniform initialization in 4/4 metrics and also compared to the all-ones initialization in ¾ metrics and being on par with the all-one initialization for the AUR score (within the one standard deviation range).
>
> At even later iteration steps (300 and beyond), we do not see any notable differences in terms of performance across all initialization methods as all methods converge to similar values. This demonstrates however that StartGrad helps achieve better results in the early stages of optimization without compromising final performance for the time-series experiments. To clarify, our results show that StartGrad offers a "warm-start" advantage, enabling a strong performance boost particular during the initial phase of optimization. This makes StartGrad particularly valuable in scenarios where fast and accurate explanations are needed, and running mask-based explanation methods for hundreds of iterations is not feasible.
>
> This early boost is consistent across  datasets (state and switch-feature dataset) and model formulations (preservation and deletion game formulation). These findings align well with the theoretical contributions of our paper and the proofs provided, which show that StartGrad is superior at initialization, particularly in high-dimensional settings.
>
> Also note the results for the vision domain, where we demonstrate that StartGrad can not only boost the performance at early iteration steps, but also at later optimization steps, i.e. using StartGrad can boost the overall performance of mask-based explanation methods. See for instance the results for the  ShearletX model where we see substantial improvements throughout the optimization process which translates to substantial speedups (see Table 2).
>
> We acknowledge that the initial version of the paper may not have sufficiently clarified and discussed the results in Table 3, and we have revised the manuscript to make the findings clearer.
>
> Furthermore, we want to stress that to the best of our knowledge we are the first ones to study the effect of different initialization techniques for mask-based explanation methods from both a theoretical and empirical perspective (next to proposing our novel initialization method StartGrad),
>
> **Final words**
>
> We hope that these responses address and clarify your questions and concerns regarding our paper and strengthen the positioning of our work. Please let us know if you have any follow-up / additional questions.
>
> Best regards,
>
> The authors
>
> **References**
>
> [1] Joseph Enguehard. Learning perturbations to explain time series predictions. In Proceedings of the 40th International Conference on Machine Learning, pp. 9329–9342. PMLR, 2023.

---

> > ### Comment · Reviewer_yTDi · 2024-11-25
> >
> > Thanks to the authors for clearly explaining their results. Considering the cost of time, their method does achieve faster reasoning. Considering that the initialization algorithm does have a limited impact on model performance, the similarity between StartGrad and baseline in terms of performance is acceptable. I noticed that the authors recounted the connection between the results recorded in Table 3 and Table 2, which better explains the effectiveness of their method.
> >
> > In addition, explaining and accelerating via initialization is an interesting perspective. Therefore I decided to raise the score to 5.
> >
> > Hopefully, the authors can complete their experiments on the ablation of lambda in time and update them to the revised paper. The authors mention that too many model frameworks may make it difficult to complete the ablation experiments all together. Therefore I suggest the authors to at least complete and update the results of ablation experiments using ResNet18, since the major experiments of this paper are also completed using ResNet18.
> >
> > Finally, there is another minor suggestion for improving the revised paper. It is better to shorten the caption of tables. For example, the author may move the content about the experimental setup from the caption to the discussion part of the corresponding experiment. This will help to enhance the readability and fluency of the paper.

---

> ### Author Response · Authors · 2024-11-25
> **Response reviewer yTDi: Revised with updated table, ablation study lambda**
>
> Dear reviewer yTDi,
>
> We sincerely thank you for your thoughtful feedback and for raising the score to 5. We greatly appreciate your recognition of our method and our explanations regarding Table 2 and 3.
>
> We just uploaded a revised version of our paper where all modifications are clearly highlighted in red. We want to also take the time to address your suggestions and feedback:
>
> ### Revised captions Table 1 and Table 2
>
> As per your suggestion, we have shortened the table captions to improve fluency and clarity while retaining all critical information for interpreting the results. We hope this change enhances the paper's readability and aligns with your expectations.
>
> ### Ablation studies for hyperparameter $\lambda_{1}$ and $\lambda_{2}$
>
> To address your request, we conducted additional ablation studies on the $\lambda_{1}$ and $\lambda_{2}$ hyperparameters using the ResNet18 classifier for PixelMask and WaveletX. The Figures are inclued in the Appendix F9, F10 and F11.
> Our findings are as follows:
>
> - **PixelMask:**
>   Increasing $\lambda_{1}$ consistently improves both the CP-Pixel and CP-L1 scores (identical in this case, as PixelMask does not apply a latent-space mask) across all initialization methods. Based on these insights, we included a new figure comparing StartGrad with all-ones and uniform initialization for $\lambda_{1}$ for which we still observe a very strong performance of StartGrad compared to the alternative initialization schemes.
>
> - **WaveletX:**
>   - Increasing $\lambda_{1}$ improves the CP-L1 metric but decreases the CP-Pixel metric across all initialization methods.
>   - The chosen hyperparameter values of $\lambda_1 = 1$ and $\lambda_2 = 10$ represent an optimal balance between these metrics, confirming the robustness of our prior selection.
>
> - **General Observation:**
>   The sensitivity of the $\lambda_{1}$ and $\lambda_{2}$ hyperparameters remains consistent across all initialization schemes, further validating our findings.
>
> Additionally, we are currently running ablation experiments for ShearletX (which is the most expensive method) and will include these results before the deadline.
>
> **Final words**:
>
> We hope these revisions and new findings meet your expectations. Thank you again for your constructive suggestions and for helping us improve the paper. We are optimistic that these updates further strengthen our paper and look forward to your feedback.
>
> Best regards,
>
> The Authors

---

> > ### Comment · Reviewer_yTDi · 2024-11-29
> >
> > Thanks to the author for the reply. The authors have addressed all my concerns and the new version of the revision has met my requirements for experimental integrity. I have raised my score to 6.

---

> > > ### Author Response · Authors · 2024-11-29
> > > **Response reviewer yTDi: Thank you**
> > >
> > > Dear reviewer yTDi,
> > >
> > > We are happy to hear that we were able to address all your concerns successfully and that our uploaded version met your requirements. Thank you once again for your valuable feedback and for taking the time to review our work. We deeply appreciate your decision to increase your score and your contribution to improving the quality of our paper.
> > >
> > > Best regards,
> > >
> > > The authors

---

### Official Review · Reviewer_Ev4u · 2024-11-04

**Soundness:** 3
**Presentation:** 3
**Contribution:** 3
**Rating:** 6
**Confidence:** 2

**Summary:**

This paper introduces StartGrad, a novel gradient-based initialization technique specifically designed for mask-based explainability methods in deep learning. While recent research has focused on developing new objective functions for mask-based explanations, the authors identify that initialization strategies have been overlooked despite their importance for optimization performance. The key contributions are (1) StartGrad: A new initialization algorithm that leverages gradient information to provide better starting points for mask optimization, transforming gradient values into initial masks using quantile transformation. (2) Theoretical Framework: A formal analysis showing that StartGrad is provably superior at initialization compared to standard strategies in balancing the fundamental tradeoff between distortion and sparsity. The authors unify existing mask-based methods under an information-theoretic framework and show how StartGrad can enhance their performance while maintaining simplicity of implementation.

**Strengths:**

1. The paper identifies and addresses an overlooked aspect of mask-based XAI methods - initialization strategy. While much work has focused on objective functions, this is the first paper to systematically study initialization.

2. The information-theoretic unification of existing mask-based methods provides a novel theoretical framework for analyzing these approaches.

3. The empirical evaluation is comprehensive (1) Covers different domains (vision, time-series), (2) applicable to multiple state-of-the-art methods (PixelMask, WaveletX, ShearletX, ExtremalMask), (3) The ablation studies thoroughly examine robustness to noisy gradients and alternative implementation choices.

4. The paper is well-structured and clearly written, with a logical flow from motivation to theory to experiments.

**Weaknesses:**

1. Reliance on Simplified Assumptions in Theoretical Analysis: The theoretical foundations of StartGrad depend on assumptions such as local linearity and neighborhood smoothness of the classifier's prediction function. While these assumptions are necessary for proving the benefits of StartGrad, they may not hold in highly non-linear or real-world scenarios, especially in models with complex architectures.

2. Domain Generalizability: The experiments are mainly centered on vision and time-series data, leaving an open question about how well StartGrad performs on other data modalities such as graphs or text.

3. Narrow Experimental Scope in Terms of Model Variants: The study mainly evaluates StartGrad on specific XAI models like PixelMask, WaveletX, and ShearletX, using ResNet18 and VGG16 classifiers. The conclusions would be more compelling if tested across a broader set of models, such as transformers or graph neural networks

4. No analysis of explanation stability across different random seeds.

5.  Missing comparison with recent developments in efficient xai methods

**Questions:**

1. The effectiveness of StartGrad is heavily dependent on the accuracy of gradient signals. Could you clarify if there are specific scenarios (e.g., highly non-linear models or adversarial settings) where gradient inaccuracies significantly degrade performance? (e.g.  understanding if and how StartGrad could be improved or adapted in scenarios where gradients are less reliable could address concerns about robustness)

---

> ### Author Response · Authors · 2024-11-21
> **Response reviewer Ev4u: Linearity assumption, model variants, random seeds**
>
> Dear reviewer Ev4u,
>
> First of all, we would like to thank you very much for your thorough and insightful review. We sincerely appreciate the time you spent evaluating our work, and very much appreciate your positive comments about the writing of our paper, the clarity of our methodology, and the novelty of our approach and the scope of the empirical evaluation. We also sincerely appreciate your thoughtful questions and will address all your comments, questions, and concerns in detail.
>
> **Weaknesses: Reliance on linearity assumption**
>
> Thank you for your insight comment. We agree that this assumption might be a strong assumption and not valid in highly non-linear, complex models. However, we tested the effectiveness of StartGrad across various architectures, including ResNet18, VGG16, as well as for two transformer architectures (Vision Transformer ViT [1] and Swin transformer [2] see our answer below) which we both added in our revised version. Furthermore, we tested StartGrad for a GRU model in a time-series setting. We see a strong performance of StartGrad across architectures and data modalities which gives empirical evidence that StartGrad is effective even for complex, highly-non linear models.
>
> **Domain generalizability**
>
> Thank you for your valuable comment regarding the domain generalizability of StartGrad. We agree that assessing its performance across diverse data modalities is essential for understanding its robustness. To this end, we evaluated StartGrad in two representative and challenging domains: vision and time-series. Vision is a well-established domain for mask-based explanation methods, while time-series data, though less explored, introduces unique challenges for gradient-based attribution methods [3]. By demonstrating StartGrad’s effectiveness in both domains, we highlight its adaptability to generalize across domains.
>
> We fully acknowledge that text and graphs are also important modalities for evaluating XAI methods. These domains present unique structural and sequential challenges, and we see them as promising directions for future research. We intend to extend our evaluation of StartGrad to these areas in future work and appreciate your suggestion in guiding this exploration.
>
> **Narrow experimental scope in terms of model variants**
>
> Thank you for your comment and suggestion regarding extending the scope of the model architectures to further strengthen the paper and empirical evaluation. We incorporated additional experiments, including both a Vision Transformer model as introduced in [1] and Swin transformer model [2]. These results, now included in Appendix F.3 and F.4, show that the overall conclusions of the paper remain consistent.
>
> Regarding your suggestion of including graph neural networks, we recognize their importance in certain application areas. However, they are less commonly used in the vision domain or time-series domain which are the two modalities used in this study. Nonetheless, we appreciate the value of testing StartGrad on graph-based architectures and consider this a promising direction for future work when we apply StartGrad for graph data (see above).
>
> We believe that the breadth of architectures we have already tested (VGG, ResNet, ViT, Swin Transformer and recurrent neural networks)  emphasizes the robustness and generalizability of StartGrad across model variants and data modalities.
>
> **Stability across different random seeds**
>
> Thank you for your thoughtful recommendation regarding the random seeds. In our initial version, we used the same random seeds as the studies that we are referring to in our comparisons to facilitate fair comparison and avoid cherry picking of random seeds. However, we acknowledge that testing the stability across random seeds is important and we therefore are currently running our experiments with additional seeds and will update and reflect this in our manuscript by the end of this week.
>
> **References**
>
> [1] Alexey Dosovitskiy, Lucas Beyer, Alexander Kolesnikov, Dirk Weissenborn, Xiaohua Zhai, Thomas
> Unterthiner, Mostafa Dehghani, Matthias Minderer, Georg Heigold, Sylvain Gelly, Jakob Uszkoreit, and Neil Houlsby. An image is worth 16x16 words: Transformers for image recognition at
> scale. In 9th International Conference on Learning Representations, ICLR 2021, Virtual Event,
> Austria, May 3-7, 2021. OpenReview.net, 2021.
>
> [2] Ze Liu, Han Hu, Yutong Lin, Zhuliang Yao, Zhenda Xie, Yixuan Wei, Jia Ning, Yue Cao, Zheng
> Zhang, Li Dong, Furu Wei, and Baining Guo. Swin transformer v2: Scaling up capacity and reso-
> lution. In Proceedings of the IEEE/CVF Conference on Computer Vision and Pattern Recognition
> (CVPR), pp. 12009–12019, June 2022.
>
> [3] Aya Abdelsalam Ismail, Mohamed Gunady, Hector Corrada Bravo, and Soheil Feizi. Benchmarking deep learning interpretability in time series predictions. In Advances in Neural Information Processing Systems, volume 33, pp. 6441–6452. Curran Associates, Inc., 2020.

---

> ### Author Response · Authors · 2024-11-21
> **Response reviewer Ev4u: XAI comparisons, effectiveness of StartGrad w.r.t. accuracy of gradients**
>
> **Missing comparisons with recent developments**
>
> Thank you for your insightful comment. We appreciate your suggestion to compare our work with recent developments in efficient XAI. To the best of our knowledge, this is the first theoretical and empirical study focusing on mask initialization methods for mask-based explanation approaches, and we are not aware of any work in this specific area within the recent literature, as the role of mask-initialization has been vastly overlooked.
>
> That being said, we fully acknowledge the importance of staying up to date with the latest advancements in the field. If there are relevant recent works that we may have overlooked, we would be grateful for any recommendations or pointers to specific papers, and we would be more than happy to consider them and update our paper accordingly.
>
> **Dependence of StartGrad performance on accuracy of gradient signal.**
>
> Thank you for your thoughtful comment regarding the reliance of StartGrad on gradient signal accuracy. We acknowledge that inaccuracies or noise in the gradient signal can potentially impact performance which is why we conducted ablation studies in our paper.
>
> To address this, we conducted ablation studies (Appendix F.5) testing the robustness of StartGrad under noisy gradient conditions. Our findings show that StartGrad is resilient to noisy gradients, maintaining comparable performance to setups with undistorted gradients.
>
> Additionally, we have expanded our analysis in Appendix F.6 to include scenarios with “uninformative” gradients, where gradient values are shuffled, breaking the link between mask coefficients and gradient information, and an adversarial setting, where the
> correspondence between gradient values and mask initialization values was reversed, such that features with higher gradient values were assigned lower mask values and vice versa.
>
> Results reveal that under uninformative gradient conditions, StartGrad's performance is comparable to the baseline uniform initialization, indicating that StartGrad relies on meaningful gradient signals to offer an advantage. In the adversarial setting, performance degradation highlights the importance of smart initialization, as the algorithm struggles to recover from a poorly initialized mask during optimization. These findings emphasize the critical role of initialization in achieving optimal performance.
>
> As a future direction, we highlight recent findings [1] that suggest filtering out high-frequency artifacts in gradients can improve gradient quality. Incorporating such techniques could further enhance StartGrad’s robustness and represents a promising avenue for follow-up research.
>
> **Final words**
>
> We hope that these responses address and clarify your questions and concerns regarding our paper and strengthen the positioning of our work. Please let us know if you have any follow-up / additional questions.
>
> Best regards,
>
> The authors
>
> **References**
>
> [1] Sabine Muzellec, Thomas FEL, Victor Boutin, Léo Andéol, Rufin VanRullen, and Thomas Serre. Saliency strikes back: How filtering out high frequencies improves white-box explanations. In Forty-first International Conference on Machine Learning, 2024

---

> > ### Comment · Reviewer_Ev4u · 2024-11-26
> >
> > Thank you for addressing my concerns. All of my concerns have been resolved, I have increased my score to 6.

---

> > > ### Author Response · Authors · 2024-11-27
> > > **Response reviewer Ev4u: Thanks**
> > >
> > > Dear reviewer Ev4u,
> > >
> > > We are happy to hear that we were able to address all your concerns successfully. Thank you once again for your valuable feedback and for taking the time to review our work. We deeply appreciate your decision to increase your score and your contribution to improving the quality of our paper.
> > >
> > > Best regards,
> > >
> > > The authors

---

### Author Response · Authors · 2024-11-28
**Updated Manuscript and Responses to Your Valuable Feedback**

Dear Reviewers,

Thank you for your thoughtful reviews and the time you’ve invested in evaluating our work. Your insightful feedback has been instrumental in helping us refine and improve the manuscript.

We are pleased to inform you that we have uploaded a revised version of the paper, addressing all your suggestions and comments. To make it easier for you to assess the changes, we have clearly highlighted all updates in red.

### Key Revisions and Additions
In response to your feedback and suggestions, the new version includes:
- **Additional experiments**: Results on additional model architectures in the vision domain (Vision Transformer, Swin Transformer).
- **Qualitative analyses**: Visual results for mask-based vision models.
- **Justification**: A detailed discussion on our choice of CP scores over faithfulness scores including analysis underlining our reasoning.
- **Comparative analysis**: Evaluation of mask-based explanation methods alongside Integrated Gradients and GradCAM.
- **Hyperparameter studies**: Investigation on the sensitivity of $\lambda_{1}$ and $\lambda_{2}$ for PixelMask, WaveletX and ShearletX.
- **Ablation studies**: Results on random seed sensitivity.
- **Optimization dynamics**: Additional analysis and insights to illustrate why StartGrad improves overall performance
- **Robustness tests**: Additional insights on the gradient signal in StartGrad.
- **Algorithm visualization**: An additional figure illustrating the StartGrad method to aid the understanding of the pseudocode.

We also worked to improve the overall clarity and readability of the manuscript.

We are in the process of addressing all comments individually, but we hope this revised manuscript already reflects our commitment to addressing your concerns to further strengthen our work.

Thank you again for your constructive feedback and for engaging with our work. We hope these improvements meet your expectations and further strengthen the position of our work and demonstrate the significance of our contributions.

Best regards,

The authors

---

### Meta-Review · Area_Chair_yP4K · 2024-12-18

**Metareview:**

This paper introduces StartGrad, a gradient-based mask initialization technique for mask-based post-hoc attribution methods (e.g., ShearletX, WaveletX). StartGrad leverages input feature gradients to initialize masks, enabling faster convergence and improved optimization. The paper includes theoretical analysis demonstrating StartGrad's superiority over traditional initialization methods like uniform or all-ones and presents experiments showcasing its performance benefits.

**Additional Comments On Reviewer Discussion:**

The reviewers appreciate the paper's extensive experiments and practical applicability in the field of mask-based attribution, which demonstrate its value to the field.

---

### Decision · Program_Chairs · 2025-01-22

Accept (Poster)